# LEARNING CURVES FOR GAUSSIAN PROCESS REGRESSION WITH POWER-LAW PRIORS AND TARGETS

**Hui Jin**
UCLA
huijin@ucla.edu

**Pradeep Kr. Banerjee**
MPI MiS
pradeep@mis.mpg.de

**Guido Montúfar**
UCLA & MPI MiS
montufar@math.ucla.edu

## ABSTRACT

We characterize the power-law asymptotics of learning curves for Gaussian process regression (GPR) under the assumption that the eigenspectrum of the prior and the eigenexpansion coefficients of the target function follow a power law. Under similar assumptions, we leverage the equivalence between GPR and kernel ridge regression (KRR) to show the generalization error of KRR. Infinitely wide neural networks can be related to GPR with respect to the neural network GP kernel and the neural tangent kernel, which in several cases is known to have a power-law spectrum. Hence our methods can be applied to study the generalization error of infinitely wide neural networks. We present toy experiments demonstrating the theory.

## 1 INTRODUCTION

Gaussian processes (GPs) provide a flexible and interpretable framework for learning and adaptive inference, and are widely used for constructing prior distributions in non-parametric Bayesian learning. From an application perspective, one crucial question is how fast do GPs learn, i.e., how much training data is needed to achieve a certain level of generalization performance. Theoretically, this is addressed by analyzing so-called "learning curves", which describe the generalization error as a function of the training set size $n$. The rate at which the curve approaches zero determines the difficulty of learning tasks and conveys important information about the asymptotic performance of GP learning algorithms. In this paper, we study the learning curves for Gaussian process regression. Our main result characterizes the asymptotics of the generalization error in cases where the eigenvalues of the GP kernel and the coefficients of the eigenexpansion of the target function have a power-law decay. In the remainder of this introductory section, we review related work and outline our main contributions.

**Gaussian processes** A GP model is a probabilistic model on an infinite-dimensional parameter space (Williams and Rasmussen, 2006; Orbanz and Teh, 2010). In GP regression (GPR), for example, this space can be the set of all continuous functions. Assumptions about the learning problem are encoded by way of a prior distribution over functions, which gets transformed into a posterior distribution given some observed data. The mean of the posterior is then used for prediction. The model uses only a finite subset of the available parameters to explain the data and this subset can grow arbitrarily large as more data are observed. In this sense, GPs are "non-parametric" and contrast with parametric models, where there is a fixed number of parameters. For regression with Gaussian noise, a major appeal of the GP formalism is that the posterior is analytically tractable. GPs are also one important part in learning with kernel machines (Kanagawa et al., 2018) and modeling using GPs has recently gained considerable traction in the neural network community.

**Neural networks and kernel learning** From a GP viewpoint, there exists a well known correspondence between kernel methods and infinite neural networks (NNs) first studied by Neal (1996). Neal showed that the outputs of a randomly initialized one-hidden layer neural network (with appropriate scaling of the variance of the initialization distribution) converges to a GP over functions in the limit of an infinite number of hidden units. Follow-up work extended this correspondence with analytical expressions for the kernel covariance for shallow NNs by Williams (1997), and more recently for deep fully-connected NNs (Lee et al., 2018; de G. Matthews et al., 2018), convolutional NNs with many channels (Novak et al., 2019; Garriga-Alonso et al., 2019), and more general architectures (Yang, 2019). The correspondence enables *exact* Bayesian inference in the associated GP model for

infinite-width NNs on regression tasks and has led to some recent breakthroughs in our understanding of overparameterized NNs (Jacot et al., 2018; Lee et al., 2019; Arora et al., 2019; Belkin et al., 2018; Daniely et al., 2016; Yang and Salman, 2019; Bietti and Mairal, 2019). The most prominent kernels associated with infinite-width NNs are the Neural Network Gaussian Process (NNGP) kernel (Lee et al., 2018; de G. Matthews et al., 2018), and the Neural Tangent Kernel (NTK) (Jacot et al., 2018). Empirical studies have shown that inference with such infinite network kernels is competitive with standard gradient descent-based optimization for fully-connected architectures (Lee et al., 2020).

**Learning curves** A large-scale empirical characterization of the generalization performance of state-of-the-art deep NNs showed that the associated learning curves often follow a power law of the form $n^{-\beta}$ with the exponent $\beta$ ranging between 0.07 and 0.35 depending on the data and the algorithm (Hestness et al., 2017; Spigler et al., 2020). Power-law asymptotics of learning curves have been theoretically studied in early works for the Gibbs learning algorithm (Amari et al., 1992; Amari and Murata, 1993; Haussler et al., 1996) that showed a generalization error scaling with exponent $\beta = 0.5$, 1 or 2 under certain assumptions. More recent results from statistical learning theory characterize the shape of learning curves depending on the properties of the hypothesis class (Bousquet et al., 2021). In the context of GPs, approximations and bounds on learning curves have been investigated in several works (Sollich, 1999; Sollich and Halees, 2002; Sollich, 2001; Opper and Vivarelli, 1999; Opper and Malzahn, 2002; Williams and Vivarelli, 2000; Malzahn and Opper, 2001a;b; Seeger et al., 2008; Van Der Vaart and Van Zanten, 2011; Le Gratiet and Garnier, 2015), with recent extensions to kernel regression from a spectral bias perspective (Bordelon et al., 2020; Canatar et al., 2021). For a review on learning curves in relation to its shape and monotonicity, see Loog et al. (2019); Viering et al. (2019); Viering and Loog (2021). A related but complementary line of work studies the convergence rates and posterior consistency properties of Bayesian non-parametric models (Barron, 1998; Seeger et al., 2008; Van Der Vaart and Van Zanten, 2011).

**Power-law decay of the GP kernel eigenspectrum** The rate of decay of the eigenvalues of the GP kernel conveys important information about its smoothness. Intuitively, if a process is "rough" with more power at high frequencies, then the eigenspectrum decays more slowly. On the other hand, kernels that define smooth processes have a fast-decaying eigenspectrum (Stein, 2012; Williams and Rasmussen, 2006). The precise eigenvalues $(\lambda_p)_{p \geq 1}$ of the operators associated to many kernels and input distributions are not known explicitly, except for a few special cases (Williams and Rasmussen, 2006). Often, however, the asymptotic properties are known. The asymptotic rate of decay of the eigenvalues of stationary kernels for input distributions with bounded support is well understood (Widom, 1963; Ritter et al., 1995). Ronen et al. (2019) showed that for inputs distributed uniformly on a hypersphere, the eigenfunctions of the arc-cosine kernel are spherical harmonics and the eigenvalues follow a power-law decay. The spectral properties of the NTK are integral to the analysis of training convergence and generalization of NNs, and several recent works empirically justify and rely on a power law assumption for the NTK spectrum (Bahri et al., 2021; Canatar et al., 2021; Lee et al., 2020; Nitanda and Suzuki, 2021). Velikanov and Yarotsky (2021) showed that the asymptotics of the NTK of infinitely wide shallow ReLU networks follows a power-law that is determined primarily by the singularities of the kernel and has the form $\lambda_p \propto p^{-\alpha}$ with $\alpha = 1 + \frac{1}{d}$, where $d$ is the input dimension.

**Asymptotics of the generalization error of kernel ridge regression (KRR)** There is a well known equivalence between GPR and KRR with the additive noise in GPR playing the role of regularization in KRR (Kanagawa et al., 2018). Analysis of the decay rates of the excess generalization error of KRR has appeared in several works, e.g, in the noiseless case with constant regularization (Bordelon et al., 2020; Spigler et al., 2020; Jun et al., 2019), and the noisy optimally regularized case (Caponnetto and De Vito, 2007; Steinwart et al., 2009; Fischer and Steinwart, 2020) under the assumption that the kernel eigenspectrum, and the eigenexpansion coefficients of the target function follow a power law. These assumptions, which are often called resp. the *capacity* and *source* conditions are related to the effective dimension of the problem and the difficulty of learning the target function (Caponnetto and De Vito, 2007; Blanchard and Mücke, 2018). Cui et al. (2021) present a unifying picture of the excess error decay rates under the capacity and source conditions in terms of the interplay between noise and regularization illustrating their results with real datasets.

**Contributions** In this work, we characterize the asymptotics of the generalization error of GPR and KRR under the capacity and source conditions. Our main contributions are as follows:

- When the eigenspectrum of the prior decays with rate $\alpha$ and the eigenexpansion coefficients of the target function decay with rate $\beta$, we show that with high probability over the draw of $n$ input samples, the negative log-marginal likelihood behaves as $\Theta(n^{\max\{\frac{1}{\alpha}, \frac{1-2\beta}{\alpha}+1\}})$ (Theorem 7) and the generalization error behaves as $\Theta(n^{\max\{\frac{1}{\alpha}-1, \frac{1-2\beta}{\alpha}\}})$ (Theorem 9). In the special case that the model is correctly specified, i.e., the GP prior is the true one from which the target functions are actually generated, our result implies that the generalization error behaves as $O(n^{\frac{1}{\alpha}-1})$ recovering as a special case a result due to Sollich and Halees (2002) (vide Remark 10).

- Under similar assumptions as in the previous item, we leverage the equivalence between GPR and KRR to show that the excess generalization error of KRR behaves as $\Theta(n^{\max\{\frac{1}{\alpha}-1, \frac{1-2\beta}{\alpha}\}})$ (Theorem 12). In the noiseless case with constant regularization, our result implies that the generalization error behaves as $\Theta(n^{\frac{1-2\beta}{\alpha}})$ recovering as a special case a result due to Bordelon et al. (2020). Specializing to the case of KRR with Gaussian design, we recover as a special case a result due to Cui et al. (2021) (vide Remark 14).

  For the unrealizable case, i.e., when the target function is outside the span of the eigenfunctions with positive eigenvalues, we show that the generalization error converges to a constant.

- We present a few toy experiments demonstrating the theory for GPR with arc-cosine kernel without biases (resp. with biases) which is the conjugate kernel of an infinitely wide shallow network with two inputs and one hidden layer without biases (resp. with biases) (Cho and Saul, 2009; Ronen et al., 2019).

## 2 BAYESIAN LEARNING AND GENERALIZATION ERROR FOR GPs

In GP regression, our goal is to learn a target function $f : \Omega \mapsto \mathbb{R}$ between an input $x \in \Omega$ and output $y \in \mathbb{R}$ based on training samples $D_n = \{(x_i, y_i)\}_{i=1}^n$. We consider an additive noise model $y_i = f(x_i) + \epsilon_i$, where $\epsilon_i \overset{\text{i.i.d.}}{\sim} \mathcal{N}(0, \sigma_{\text{true}}^2)$. If $\rho$ denotes the marginal density of the inputs $x_i$, then the pairs $(x_i, y_i)$ are generated according to the density $q(x, y) = \rho(x)q(y|x)$, where $q(y|x) = \mathcal{N}(y|f(x), \sigma_{\text{true}}^2)$. We assume that there is a prior distribution $\Pi_0$ on $f$ which is defined as a zero-mean GP with continuous and bounded covariance function $k : \Omega \times \Omega \to \mathbb{R}$, i.e., $f \sim \mathcal{GP}(0, k)$. This means that for any finite set $\mathbf{x} = (x_1, ..., x_n)^T$, the random vector $f(\mathbf{x}) = (f(x_1), ..., f(x_n))^T$ follows the multivariate normal distribution $\mathcal{N}(0, K_n)$ with covariance matrix $K_n = (k(x_i, x_j))_{i,j=1}^n \in \mathbb{R}^{n \times n}$. By Bayes' rule, the posterior distribution over $f$ given the training data is given by

$$d\Pi_n(f|D_n) = \frac{1}{Z(D_n)} \prod_{i=1}^n \mathcal{N}(y_i|f(x_i), \sigma_{\text{model}}^2) d\Pi_0(f),$$

where $\Pi_0$ is the prior distribution, $Z(D_n) = \int \prod_{i=1}^n \mathcal{N}(y_i|f(x_i), \sigma_{\text{model}}^2) d\Pi_0(f)$ is the *marginal likelihood* or *model evidence* and $\sigma_{\text{model}}$ is the sample variance used in GPR. In practice, we do not know the exact value of $\sigma_{\text{true}}$ and so our choice of $\sigma_{\text{model}}$ can be different from $\sigma_{\text{true}}$. The GP prior and the Gaussian noise assumption allows for exact Bayesian inference and the posterior distribution over functions is again a GP with mean and covariance function given by

$$\bar{m}(x) = K_{\mathbf{x}x}^T (K_n + \sigma_{\text{model}}^2 I_n)^{-1} \mathbf{y}, x \in \Omega \tag{1}$$

$$\bar{k}(x, x') = k(x, x') - K_{\mathbf{x}x}^T (K_n + \sigma_{\text{model}}^2 I_n)^{-1} K_{\mathbf{x}x'}, x, x' \in \Omega, \tag{2}$$

where $K_{\mathbf{x}x} = (k(x_1, x), ..., k(x_n, x))^T$ and $\mathbf{y} = (y_1, ..., y_n)^T \in \mathbb{R}^n$ (Williams and Rasmussen, 2006, Eqs. 2.23-24).

The performance of GPR depends on how well the posterior approximates $f$ as the number of training samples $n$ tends to infinity. The distance of the posterior to the ground truth can be measured in various ways. We consider two such measures, namely the Bayesian generalization error (Seeger et al., 2008; Haussler and Opper, 1997; Opper and Vivarelli, 1999) and the excess mean squared error (Sollich and Halees, 2002; Le Gratiet and Garnier, 2015; Bordelon et al., 2020; Cui et al., 2021).

**Definition 1** (Bayesian generalization error)**.** *The Bayesian generalization error is defined as the Kullback-Leibler divergence between the true density $q(y|x)$ and the Bayesian predictive density $p_n(y|x, D_n) = \int \mathcal{N}(y|f(x), \sigma_{\text{model}}^2) d\Pi_n(f|D_n)$,*

$$G(D_n) = \int q(x, y) \log \frac{q(y|x)}{p_n(y|x, D_n)} dx dy. \tag{3}$$

A related quantity of interest is the *stochastic complexity* (SC), also known as the *free energy*, which is just the negative log-marginal likelihood. We shall primarily be concerned with a normalized version of the stochastic complexity which is defined as follows:

$$F^0(D_n) = -\log\frac{Z(D_n)}{\prod_{i=1}^n q(y_i|x_i)} = -\log\frac{\int\prod_{i=1}^n \mathcal{N}(y_i|f(x_i),\sigma^2_{\text{model}})d\Pi_0(f)}{\prod_{i=1}^n q(y_i|x_i)}. \tag{4}$$

The generalization error (3) can be expressed in terms of the normalized SC as follows (Watanabe, 2009, Theorem 1.2):

$$G(D_n) = \mathbb{E}_{(x_{n+1},y_{n+1})}F^0(D_{n+1}) - F^0(D_n), \tag{5}$$

where $D_{n+1} = D_n \cup \{(x_{n+1},y_{n+1})\}$ is obtained by augmenting $D_n$ with a test point $(x_{n+1},y_{n+1})$.

If we only wish to measure the performance of the mean of the Bayesian posterior, then we can use the excess mean squared error:

**Definition 2** (Excess mean squared error). *The excess mean squared error is defined as*

$$M(D_n) = \mathbb{E}_{(x_{n+1},y_{n+1})}(\bar{m}(x_{n+1})-y_{n+1})^2 - \sigma^2_{\text{true}} = \mathbb{E}_{x_{n+1}}(\bar{m}(x_{n+1})-f(x_{n+1}))^2. \tag{6}$$

**Proposition 3** (Normalized stochastic complexity for GPR). *Assume that* $\sigma^2_{\text{model}} = \sigma^2_{\text{true}} = \sigma^2$. *The normalized SC* $F^0(D_n)$ *(4) for GPR with prior* $\mathcal{GP}(0,k)$ *is given as*

$$F^0(D_n) = \tfrac{1}{2}\text{logdet}(I_n + \tfrac{K_n}{\sigma^2}) + \tfrac{1}{2\sigma^2}\mathbf{y}^T(I_n+\tfrac{K_n}{\sigma^2})^{-1}\mathbf{y} - \tfrac{1}{2\sigma^2}(\mathbf{y}-f(\mathbf{x}))^T(\mathbf{y}-f(\mathbf{x})), \tag{7}$$

*where* $\boldsymbol{\epsilon} = (\epsilon_1,...,\epsilon_n)^T$. *The expectation of the normalized SC w.r.t. the noise* $\boldsymbol{\epsilon}$ *is given as*

$$\mathbb{E}_{\boldsymbol{\epsilon}}F^0(D_n) = \tfrac{1}{2}\text{logdet}\left(I_n+\tfrac{K_n}{\sigma^2}\right) - \tfrac{1}{2}\text{Tr}\left(I_n - \left(I_n+\tfrac{K_n}{\sigma^2}\right)^{-1}\right) + \tfrac{1}{2\sigma^2}f(\mathbf{x})^T\left(I_n+\tfrac{K_n}{\sigma^2}\right)^{-1}f(\mathbf{x}). \tag{8}$$

This is a basic result and has applications in relation to model selection in GPR (Williams and Rasmussen, 2006). For completeness, we give a proof of Proposition 3 in Appendix B. Seeger et al. (2008, Theorem 1) gave an upper bound on the normalized stochastic complexity for the case when $f$ lies in the reproducing kernel Hilbert space (RKHS) of the GP prior. It is well known, however, that sample paths of GP almost surely fall outside the corresponding RKHS (Van Der Vaart and Van Zanten, 2011) limiting the applicability of the result.

We next derive the asymptotics of $\mathbb{E}_{\boldsymbol{\epsilon}}F^0(D_n)$, the expected generalization error $\mathbb{E}_{\boldsymbol{\epsilon}}G(D_n) = \mathbb{E}_{\boldsymbol{\epsilon}}\mathbb{E}_{(x_{n+1},y_{n+1})}F^0(D_n+1) - \mathbb{E}_{\boldsymbol{\epsilon}}F^0(D_n)$, and the excess mean squared error $\mathbb{E}_{\boldsymbol{\epsilon}}M(D_n)$.

## 3   ASYMPTOTIC ANALYSIS OF GP REGRESSION WITH POWER-LAW PRIORS

We begin by introducing some notations and assumptions. We assume that $f \in L^2(\Omega,\rho)$. By the generalization of Mercer's theorem (Steinwart and Scovel, 2012, Corollary 3.2), the covariance function of the GP prior can be decomposed as $k(x_1, x_2) = \sum_{p=1}^\infty \lambda_p\phi_p(x_1)\phi_p(x_2)$ $\rho$-almost surely, where $(\phi_p(x))_{p\geq 1}$ are the eigenfunctions of the operator $L_k : L^2(\Omega, \rho) \mapsto L^2(\Omega, \rho)$; $(L_kf)(x) = \int_\Omega k(x,s)f(s)d\rho(s)$, and $(\lambda_p)_{p\geq 1}$ are the corresponding positive eigenvalues. We index the sequence of eigenvalues in decreasing order, that is $\lambda_1 \geq \lambda_2 \geq \cdots > 0$. The target function $f(x)$ is decomposed into the orthonormal set $(\phi_p(x))_{p\geq 1}$ and its orthogonal complement $\{\phi_p(x):p\geq 1\}^\perp$ as

$$f(x) = \sum_{p=1}^\infty \mu_p\phi_p(x) + \mu_0\phi_0(x) \in L^2(\Omega,\rho), \tag{9}$$

where $\boldsymbol{\mu} = (\mu_0,\mu_1,...,\mu_p,...)^T$ are the coefficients of the decomposition, and $\phi_0(x)$ satisfies $\|\phi_0(x)\|_2 = 1$ and $\phi_0(x) \in \{\phi_p(x) : p \geq 1\}^\perp$. For given sample inputs $\mathbf{x}$, let $\phi_p(\mathbf{x}) = (\phi_p(x_1),...,\phi_p(x_n))^T$, $\Phi = (\phi_0(\mathbf{x}),\phi_1(\mathbf{x}),...,\phi_p(\mathbf{x}),...)$ and $\Lambda = \text{diag}\{0,\lambda_1,...,\lambda_p,...\}$. Then the covariance matrix $K_n$ can be written as $K_n = \Phi\Lambda\Phi^T$, and the function values on the sample inputs can be written as $f(\mathbf{x}) = \Phi\boldsymbol{\mu}$.

We shall make the following assumptions in order to derive the power-law asymptotics of the normalized stochastic complexity and the generalization error of GPR:

**Assumption 4** (Power law decay of eigenvalues). *The eigenvalues* $(\lambda_p)_{p\geq 1}$ *follow the power law*

$$\underline{C_\lambda}p^{-\alpha} \leq \lambda_p \leq \overline{C_\lambda}p^{-\alpha}, \forall p \geq 1 \tag{10}$$

*where* $\underline{C_\lambda}$, $\overline{C_\lambda}$ *and* $\alpha$ *are three positive constants which satisfy* $0 < \underline{C_\lambda} \leq \overline{C_\lambda}$ *and* $\alpha > 1$.

As mentioned in the introduction, this assumption, called the capacity condition, is fairly standard in kernel learning and is adopted in many recent works (Bordelon et al., 2020; Canatar et al., 2021; Jun et al., 2019; Bietti et al., 2021; Cui et al., 2021). Velikanov and Yarotsky (2021) derived the exact value of the exponent $\alpha$ when the kernel function has a homogeneous singularity on its diagonal, which is the case for instance for the arc-cosine kernel.

**Assumption 5** (Power law decay of coefficients of decomposition)**.** *Let $C_\mu, \underline{C_\mu} > 0$ and $\beta > 1/2$ be positive constants and let $\{p_i\}_{i \geq 1}$ be an increasing integer sequence such that $\sup_{i \geq 1}(p_{i+1} - p_i) < \infty$. The coefficients $(\mu_p)_{p \geq 1}$ of the decomposition (9) of the target function follow the power law*

$$|\mu_p| \leq C_\mu p^{-\beta}, \forall p \geq 1 \quad and \quad |\mu_{p_i}| \geq \underline{C_\mu} p_i^{-\beta}, \forall i \geq 1. \tag{11}$$

Since $f \in L^2(\Omega, \rho)$, we have $\sum_{p=0}^{\infty} \mu_p^2 < \infty$. The condition $\beta > 1/2$ in Assumption 5 ensures that the sum $\sum_{p=0}^{\infty} \mu_p^2$ does not diverge. When the orthonormal basis $(\phi_p(x))_p$ is the Fourier basis or the spherical harmonics basis, the coefficients $(\mu_p)_p$ decay at least as fast as a power law so long as the target function $f(x)$ satisfies certain smoothness conditions (Bietti and Mairal, 2019). Velikanov and Yarotsky (2021) gave examples of some natural classes of functions for which Assumption 5 is satisfied, such as functions that have a bounded support with smooth boundary and are smooth on the interior of this support, and derived the corresponding exponents $\beta$.

**Assumption 6** (Boundedness of eigenfunctions)**.** *The eigenfunctions $(\phi_p(x))_{p \geq 0}$ satisfy*

$$\|\phi_0\|_\infty \leq C_\phi \quad and \quad \|\phi_p\|_\infty \leq C_\phi p^\tau, p \geq 1, \tag{12}$$

*where $C_\phi$ and $\tau$ are two positive constants which satisfy $\tau < \frac{\alpha - 1}{2}$.*

The second condition in (12) appears, for example, in Valdivia (2018, Hypothesis H$_1$) and is less restrictive than the assumption of uniformly bounded eigenfunctions that has appeared in several other works in the GP literature, see, e.g., Braun (2006); Chatterji et al. (2019); Vakili et al. (2021).

Define

$$T_1(D_n) = \tfrac{1}{2} \text{logdet}\left(I_n + \tfrac{\Phi \Lambda \Phi^T}{\sigma^2}\right) - \tfrac{1}{2}\text{Tr}\left(I_n - \left(I_n + \tfrac{\Phi \Lambda \Phi^T}{\sigma^2}\right)^{-1}\right), \tag{13}$$

$$T_2(D_n) = \tfrac{1}{2\sigma^2} f(\mathbf{x})^T \left(I_n + \tfrac{\Phi \Lambda \Phi^T}{\sigma^2}\right)^{-1} f(\mathbf{x}), \tag{14}$$

$$G_1(D_n) = \mathbb{E}_{(x_{n+1}, y_{n+1})}(T_1(D_{n+1}) - T_1(D_n)), \tag{15}$$

$$G_2(D_n) = \mathbb{E}_{(x_{n+1}, y_{n+1})}(T_2(D_{n+1}) - T_2(D_n)). \tag{16}$$

Using (8) and (5), we have $\mathbb{E}_\epsilon F^0(D_n) = T_1(D_n) + T_2(D_n)$ and $\mathbb{E}_\epsilon G(D_n) = G_1(D_n) + G_2(D_n)$. Intuitively, $G_1$ corresponds to the effect of the noise on the generalization error irrespective of the target function $f$, whereas $G_2$ corresponds to the ability of the model to fit the target function. As we will see next in Theorems 9 and 11, if $\alpha$ is large, then the error associated with the noise is smaller. When $f$ is contained in the span of the eigenfunctions $\{\phi_p\}_{p \geq 1}$, $G_2$ decreases with increasing $n$, but if $f$ contains an orthogonal component, then the error remains constant and GP regression is not able to learn the target function.

## 3.1 ASYMPTOTICS OF THE NORMALIZED STOCHASTIC COMPLEXITY

We derive the asymptotics of the normalized SC (8) for the following two cases: $\mu_0 = 0$ and $\mu_0 > 0$. When $\mu_0 = 0$, the target function $f(x)$ lies in the span of all eigenfunctions with positive eigenvalues.

**Theorem 7** (Asymptotics of the normalized SC, $\mu_0 = 0$)**.** *Assume that $\mu_0 = 0$ and $\sigma_{\text{model}}^2 = \sigma_{\text{true}}^2 = \sigma^2 = \Theta(1)$. Under Assumptions 4, 5 and 6, with probability of at least $1 - n^{-q}$ over sample inputs $(x_i)_{i=1}^n$, where $0 \leq q < \min\{\frac{(2\beta-1)(\alpha-1-2\tau)}{4\alpha^2}, \frac{\alpha-1-2\tau}{2\alpha}\}$, the expected normalized SC (8) has the asymptotic behavior:*

$$\mathbb{E}_\epsilon F^0(D_n) = \left[\tfrac{1}{2}\text{logdet}(I + \tfrac{n}{\sigma^2}\Lambda) - \tfrac{1}{2}\text{Tr}\left(I - (I + \tfrac{n}{\sigma^2}\Lambda)^{-1}\right) + \tfrac{n}{2\sigma^2}\boldsymbol{\mu}^T(I + \tfrac{n}{\sigma^2}\Lambda)^{-1}\boldsymbol{\mu}\right](1 + o(1))$$

$$= \Theta(n^{\max\{\frac{1}{\alpha}, \frac{1-2\beta}{\alpha}+1\}}). \tag{17}$$

The complete proof of Theorem 7 is given in Appendix D.1. We give a sketch of the proof below. In the sequel, we use the notations $O$ and $\Theta$ to denote the standard mathematical orders and the notation $\tilde{O}$ to suppress logarithmic factors.

*Proof sketch of Theorem 7.* By (8), (13) and (14) we have $\mathbb{E}_\epsilon F^0(D_n) = T_1(D_n) + T_2(D_n)$. In order to analyze the terms $T_1(D_n)$ and $T_2(D_n)$, we will consider truncated versions of these quantities and bound the corresponding residual errors. Given a truncation parameter $R \in \mathbb{N}$, let $\Phi_R = (\phi_0(\mathbf{x}), \phi_1(\mathbf{x}), ..., \phi_R(\mathbf{x})) \in \mathbb{R}^{n \times R}$ be the truncated matrix of eigenfunctions evaluated at the data points, $\Lambda_R = \text{diag}(0, \lambda_1, ..., \lambda_R) \in \mathbb{R}^{(R+1) \times (R+1)}$ and $\boldsymbol{\mu}_R = (\mu_0, \mu_1, ..., \mu_R) \in \mathbb{R}^{R+1}$. We define the truncated version of $T_1(D_n)$ as follows:

$$T_{1,R}(D_n) = \tfrac{1}{2}\text{logdet}\Big(I_n + \tfrac{\Phi_R \Lambda_R \Phi_R^T}{\sigma^2}\Big) - \tfrac{1}{2}\text{Tr}\Big(I_n - (I_n + \tfrac{\Phi_R \Lambda_R \Phi_R^T}{\sigma^2})^{-1}\Big). \tag{18}$$

Similarly, define $\Phi_{>R} = (\phi_{R+1}(\mathbf{x}), \phi_{R+2}(\mathbf{x}), ..., \phi_p(\mathbf{x}), ...)$, $\Lambda_{>R} = \text{diag}(\lambda_{R+1}, ..., \lambda_p, ...)$, $f_R(x) = \sum_{p=1}^{R} \mu_p \phi_p(x)$, $f_R(\mathbf{x}) = (f_R(x_1), ..., f_R(x_n))^T$, $f_{>R}(x) = f(x) - f_R(x)$, and $f_{>R}(\mathbf{x}) = (f_{>R}(x_1), ..., f_{>R}(x_n))^T$. The truncated version of $T_2(D_n)$ is then defined as

$$T_{2,R}(D_n) = \tfrac{1}{2\sigma^2} f_R(\mathbf{x})^T (I_n + \tfrac{\Phi_R \Lambda_R \Phi_R^T}{\sigma^2})^{-1} f_R(\mathbf{x})^T. \tag{19}$$

The proof consists of three steps:

- **Approximation step:** In this step, we show that the asymptotics of $T_{1,R}$ resp. $T_{2,R}$ dominates that of the residuals, $|T_{1,R}(D_n) - T_1(D_n)|$ resp. $|T_{2,R}(D_n) - T_2(D_n)|$ (see Lemma 32). This builds upon first showing that $\|\Phi_{>R} \Lambda_{>R} \Phi_{>R}^T\|_2 = \tilde{O}(\max\{nR^{-\alpha}, n^{\frac{1}{2}} R^{\frac{1-2\alpha}{2}}, R^{1-\alpha}\})$ (see Lemma 25) and then choosing $R = n^{\frac{1}{\alpha} + \kappa}$ where $0 < \kappa < \frac{\alpha - 1 - 2\tau}{2\alpha^2}$ when we have $\|\Phi_{>R} \Lambda_{>R} \Phi_{>R}^T\|_2 = o(1)$. Intuitively, the choice of the truncation parameter $R$ is governed by the fact that $\lambda_R = \Theta(R^{-\alpha}) = n^{-1+\kappa\alpha} = o(n^{-1})$.

- **Decomposition step:** In this step, we decompose $T_{1,R}$ into a term independent of $\Phi_R$ and a series involving $\Phi_R^T \Phi_R - nI_R$, and likewise for $T_{2,R}$ (see Lemma 34). This builds upon first showing using the Woodbury matrix identity (Williams and Rasmussen, 2006, §A.3) that

$$T_{1,R}(D_n) = \tfrac{1}{2}\text{logdet}(I_R + \tfrac{1}{\sigma^2}\Lambda_R \Phi_R^T \Phi_R) - \tfrac{1}{2}\text{Tr}\Phi_R(\sigma^2 I_R + \Lambda_R \Phi_R^T \Phi_R)^{-1}\Lambda_R \Phi_R^T, \tag{20}$$

$$T_{2,R}(D_n) = \tfrac{1}{2\sigma^2}\boldsymbol{\mu}_R^T \Phi_R^T \Phi_R(\sigma^2 I_R + \Lambda_R \Phi_R^T \Phi_R)^{-1}\boldsymbol{\mu}_R, \tag{21}$$

and then Taylor expanding the matrix inverse $(\sigma^2 I_R + \Lambda_R \Phi_R^T \Phi_R)^{-1}$ in (20) and (21) to show that the $\Phi_R$-independent terms in the decomposition of $T_{1,R}$ and $T_{2,R}$ are, respectively, $\tfrac{1}{2}\text{logdet}(I_R + \tfrac{n}{\sigma^2}\Lambda_R) - \tfrac{1}{2}\text{Tr}\big(I_R - (I_R + \tfrac{n}{\sigma^2}\Lambda_R)^{-1}\big)$, and $\tfrac{n}{2\sigma^2}\boldsymbol{\mu}_R^T (I_R + \tfrac{n}{\sigma^2}\Lambda_R)^{-1}\boldsymbol{\mu}_R$.

- **Concentration step:** Finally, we use concentration inequalities to show that these $\Phi_R$-independent terms dominate the series involving $\Phi_R^T \Phi_R - nI_R$ (see Lemma 35) when we have

$$T_{1,R}(D_n) = \big(\tfrac{1}{2}\text{logdet}(I_R + \tfrac{n}{\sigma^2}\Lambda_R) - \tfrac{1}{2}\text{Tr}\big(I_R - (I_R + \tfrac{n}{\sigma^2}\Lambda_R)^{-1}\big)\big)(1 + o(1)) = \Theta(n^{\frac{1}{\alpha}}),$$

$$T_{2,R}(D_n) = \big(\tfrac{n}{2\sigma^2}\boldsymbol{\mu}_R^T (I_R + \tfrac{n}{\sigma^2}\Lambda_R)^{-1}\boldsymbol{\mu}_R\big)(1 + o(1)) = \begin{cases} \Theta(n^{\max\{0, \frac{1-2\beta}{\alpha}+1\}}), & \alpha \neq 2\beta - 1, \\ \Theta(\log n), & \alpha = 2\beta - 1. \end{cases}$$

The key idea is to consider the matrix $\Lambda_R^{1/2}(I + \tfrac{n}{\sigma^2}\Lambda_R)^{-1/2}\Phi_R^T \Phi_R(I + \tfrac{n}{\sigma^2}\Lambda_R)^{-1/2}\Lambda_R^{1/2}$ and show that it concentrates around $n\Lambda_R(I + \tfrac{n}{\sigma^2})^{-1}$ (see Corollary 22). Note that an ordinary application of the matrix Bernstein inequality to $\Phi_R^T \Phi_R - nI_R$ yields $\|\Phi_R^T \Phi_R - nI\|_2 = O(R\sqrt{n})$, which is not sufficient for our purposes, since this would give $O(R\sqrt{n}) = o(n)$ only when $\alpha > 2$. In contrast, our results are valid for $\alpha > 1$ and cover cases of practical interest, e.g., the NTK of infinitely wide shallow ReLU network (Velikanov and Yarotsky, 2021) and the arc-cosine kernels over high-dimensional hyperspheres (Ronen et al., 2019) that have $\alpha = 1 + O(\frac{1}{d})$, where $d$ is the input dimension. $\qquad\square$

For $\mu_0 > 0$, we note the following result:

**Theorem 8** (Asymptotics of the normalized SC, $\mu_0 > 0$). *Assume $\mu_0 > 0$ and $\sigma_{\text{model}}^2 = \sigma_{\text{true}}^2 = \sigma^2 = \Theta(1)$. Under Assumptions 4, 5 and 6, with probability of at least $1 - n^{-q}$ over sample inputs $(x_i)_{i=1}^n$, where $0 \leq q < \min\{\frac{2\beta-1}{2}, \alpha\} \cdot \min\{\frac{\alpha-1-2\tau}{2\alpha^2}, \frac{2\beta-1}{\alpha}\}$. the expected normalized SC (8) has the asymptotic behavior: $\mathbb{E}_\epsilon F^0(D_n) = \frac{1}{2\sigma^2}\mu_0^2 n + o(n)$.*

The proof of Theorem 8 is given in Appendix D.1 and follows from showing that when $\mu_0 > 0$, $T_{2,R}(D_n) = \big(\frac{n}{2\sigma^2}\boldsymbol{\mu}_R^T (I_R + \frac{n}{\sigma^2}\Lambda_R)^{-1}\boldsymbol{\mu}_R\big)(1 + o(1)) = \frac{1}{2\sigma^2}\mu_0^2 n + o(n)$ (see Lemma 38), which dominates $T_1(D_n)$ and the residual $|T_{2,R}(D_n) - T_2(D_n)|$.

### 3.2 ASYMPTOTICS OF THE BAYESIAN GENERALIZATION ERROR

In this section, we derive the asymptotics of the expected generalization error $\mathbb{E}_\epsilon G(D_n)$ by analyzing the asymptotics of the components $G_1(D_n)$ and $G_2(D_n)$ in resp. (15) and (16) for the following two cases: $\mu_0 = 0$ and $\mu_0 > 0$. First, we consider the case $\mu_0 = 0$.

**Theorem 9** (Asymptotics of the Bayesian generalization error, $\mu_0 = 0$)**.** *Let Assumptions 4, 5, and 6 hold. Assume that $\mu_0 = 0$ and $\sigma_{\text{model}}^2 = \sigma_{\text{true}}^2 = \sigma^2 = \Theta(n^t)$ where $1 - \frac{\alpha}{1+2\tau} < t < 1$. Then with probability of at least $1 - n^{-q}$ over sample inputs $(x_i)_{i=1}^n$ where $0 \leq q < \frac{[\alpha - (1+2\tau)(1-t)](2\beta - 1)}{4\alpha^2}$, the expectation of the Bayesian generalization error (3) w.r.t. the noise $\epsilon$ has the asymptotic behavior:*

$$\mathbb{E}_\epsilon G(D_n) = \frac{1+o(1)}{2\sigma^2}\left( \text{Tr}(I + \tfrac{n}{\sigma^2}\Lambda)^{-1}\Lambda - \|\Lambda^{1/2}(I + \tfrac{n}{\sigma^2}\Lambda)^{-1}\|_F^2 + \|(I + \tfrac{n}{\sigma^2}\Lambda)^{-1}\boldsymbol{\mu}\|_2^2 \right)$$

$$= \frac{1}{\sigma^2}\Theta\left(n^{\max\{\frac{(1-\alpha)(1-t)}{\alpha}, \frac{(1-2\beta)(1-t)}{\alpha}\}}\right). \tag{22}$$

The proof of Theorem 9 is given in Appendix D.2. Intuitively, for a given $t$, the exponent $\frac{(1-\alpha)(1-t)}{\alpha}$ in (22) captures the rate at which the model suppresses the noise, while the exponent $\frac{(1-2\beta)(1-t)}{\alpha}$ captures the rate at which the model learns the target function. A larger $\beta$ implies that the exponent $\frac{(1-2\beta)(1-t)}{\alpha}$ is smaller and it is easier to learn the target. A larger $\alpha$ implies that the exponent $\frac{(1-\alpha)(1-t)}{\alpha}$ is smaller and the error associated with the noise is smaller as well. A larger $\alpha$, however, also implies that the exponent $\frac{(1-2\beta)(1-t)}{\alpha}$ is larger (recall that $\alpha > 1$ and $\beta > 1/2$ by Assumptions 4 and 5, resp.), which means that it is harder to learn the target.

**Remark 10.** *If $f \sim \mathcal{GP}(0, k)$, then using the Karhunen-Loève expansion we have $f(x) = \sum_{p=1}^\infty \sqrt{\lambda_p}\omega_p\phi_p(x)$, where $(\omega_p)_{p=1}^\infty$ are i.i.d. standard Gaussian variables. We can bound $\omega_p$ almost surely as $|\omega_p| \leq C\log p$, where $C = \sup_{p\geq 1}\frac{|\omega_p|}{\log p}$ is a finite constant. Comparing with the expansion of $f(x)$ in (9), we find that $\mu_p = \sqrt{\lambda_p}\omega_p = O(p^{-\alpha/2}\log p) = O(p^{-\alpha/2+\varepsilon})$ where $\varepsilon > 0$ is arbitrarily small. Choosing $\beta = \alpha/2 - \varepsilon$ in (22), we have $\mathbb{E}_\epsilon G(D_n) = O(n^{\frac{1}{\alpha}-1+\frac{2\varepsilon}{\alpha}})$. This rate matches that of an earlier result due to [Sollich and Halees (2002)](#), where it is shown that the asymptotic learning curve (as measured by the expectation of the excess mean squared error, $\mathbb{E}_f M(D_n)$) scales as $n^{\frac{1}{\alpha}-1}$ when the model is correctly specified, i.e., $f$ is a sample from the same Gaussian process $\mathcal{GP}(0,k)$, and the eigenvalues decay as a power law for large $i$, $\lambda_i \sim i^\alpha$.*

For $\mu_0 > 0$, we note the following result:

**Theorem 11** (Asymptotics of the Bayesian generalization error, $\mu_0 > 0$)**.** *Let Assumptions 4, 5, and 6 hold. Assume that $\mu_0 > 0$ and $\sigma_{\text{model}}^2 = \sigma_{\text{true}}^2 = \sigma^2 = \Theta(n^t)$ where $1 - \frac{\alpha}{1+2\tau} < t < 1$. Then with probability of at least $1 - n^{-q}$ over sample inputs $(x_i)_{i=1}^n$, where $0 \leq q < \frac{[\alpha - (1+2\tau)(1-t)](2\beta - 1)}{4\alpha^2}$, the expectation of the Bayesian generalization error (3) w.r.t. the noise $\epsilon$ has the asymptotic behavior: $\mathbb{E}_\epsilon G(D_n) = \frac{1}{2\sigma^2}\mu_0^2 + o(1)$.*

In general, if $\mu_0 > 0$, then the generalization error remains constant when $n \to \infty$. This means that if the target function contains a component in the kernel of the operator $L_k$, then GP regression is not able to learn the target function. The proof of Theorem 11 is given in Appendix D.2.

### 3.3 ASYMPTOTICS OF THE EXCESS MEAN SQUARED ERROR

In this section we derive the asymptotics of the excess mean squared error in Definition 2.

**Theorem 12** (Asymptotics of excess mean squared error)**.** *Let Assumptions 4, 5, and 6 hold. Assume $\sigma_{\text{model}}^2 = \Theta(n^t)$ where $1 - \frac{\alpha}{1+2\tau} < t < 1$. Then with probability of at least $1 - n^{-q}$ over sample inputs $(x_i)_{i=1}^n$, where $0 \leq q < \frac{[\alpha - (1+2\tau)(1-t)](2\beta - 1)}{4\alpha^2}$, the excess mean squared error (6) has the asymptotic:*

$$\mathbb{E}_\epsilon M(D_n) = (1 + o(1))\left[ \frac{\sigma_{\text{true}}^2}{\sigma_{\text{model}}^2}\left( \text{Tr}(I + \tfrac{n}{\sigma_{\text{model}}^2}\Lambda)^{-1}\Lambda - \|\Lambda^{1/2}(I + \tfrac{n}{\sigma_{\text{model}}^2}\Lambda)^{-1}\|_F^2 \right) \right.$$

$$\left. + \|(I + \tfrac{n}{\sigma_{\text{model}}^2}\Lambda)^{-1}\boldsymbol{\mu}\|_2^2 \right] = \Theta\left( \max\{\sigma_{\text{true}}^2 n^{\frac{1-\alpha-t}{\alpha}}, n^{\frac{(1-2\beta)(1-t)}{\alpha}}\} \right)$$

*when $\mu_0 = 0$, and $\mathbb{E}_\epsilon M(D_n) = \mu_0^2 + o(1)$, when $\mu_0 > 0$.*

The proof of Theorem 12 uses similar techniques as Theorem 9 and is given in Appendix D.3.

**Remark 13** (Correspondence with kernel ridge regression)**.** *The kernel ridge regression (KRR) estimator arises as a solution to the optimization problem*

$$\hat{f} = \underset{f \in \mathcal{H}_k}{\operatorname{argmin}} \frac{1}{n} \sum_{i=1}^{n} (f(x_i) - y_i)^2 + \lambda \|f\|_{\mathcal{H}_k}^2, \tag{23}$$

*where the hypothesis space $\mathcal{H}_k$ is chosen to be an RKHS, and $\lambda > 0$ is a regularization parameter. The solution to* (23) *is unique as a function, and is given by $\hat{f}(x) = K_{\mathbf{x}x}^T (K_n + n\lambda I_n)^{-1}\mathbf{y}$, which coincides with the posterior mean function $\bar{m}(x)$ of the GPR* (1) *if $\sigma_{\text{model}}^2 = n\lambda$ (Kanagawa et al., 2018, Proposition 3.6). Thus, the additive Gaussian noise in GPR plays the role of regularization in KRR. Leveraging this well known equivalence between GPR and KRR we observe that Theorem 12 also describes the generalization error of KRR as measured by the excess mean squared error.*

**Remark 14.** *Cui et al. (2021) derived the asymptotics of the expected excess mean-squared error for different regularization strengths and different scales of noise. In particular, for KRR with Gaussian design where $\Lambda_R^{1/2}(\phi_1(x),...,\phi_R(x)))$ is assumed to follow a Gaussian distribution $\mathcal{N}(0, \Lambda_R)$, and regularization $\lambda = n^{t-1}$ where $1 - \alpha \leq t$, Cui et al. (2021, Eq. 10) showed that*

$$\mathbb{E}_{\{x_i\}_{i=1}^n} \mathbb{E}_{\boldsymbol{\epsilon}} M(D_n) = O\Big(\max\{\sigma_{\text{true}}^2 n^{\frac{1-\alpha-t}{\alpha}}, n^{\frac{(1-2\beta)(1-t)}{\alpha}}\}\Big). \tag{24}$$

*Let $\delta = n^{-q}$, where $0 \leq q < \frac{[\alpha - (1+2\tau)(1-t)](2\beta-1)}{4\alpha^2}$. By Markov's inequality, this implies that with probability of at least $1 - \delta$, $\mathbb{E}_{\boldsymbol{\epsilon}} M(D_n) = O(\frac{1}{\delta} \max\{\sigma_{\text{true}}^2 n^{\frac{1-\alpha-t}{\alpha}}, n^{\frac{(1-2\beta)(1-t)}{\alpha}}\}) = O(n^q \max\{\sigma_{\text{true}}^2 n^{\frac{1-\alpha-t}{\alpha}}, n^{\frac{(1-2\beta)(1-t)}{\alpha}}\})$. Theorem 12 improves upon this by showing that with probability of at least $1 - \delta$, we have an optimal bound $\mathbb{E}_{\boldsymbol{\epsilon}} M(D_n) = \Theta(\max\{\sigma_{\text{true}}^2 n^{\frac{1-\alpha-t}{\alpha}}, n^{\frac{(1-2\beta)(1-t)}{\alpha}}\})$. Furthermore, in contrast to the approach by Cui et al. (2021), we have no requirement on the distribution of $\phi_p(x)$, and hence our result is more generally applicable. For example, Theorem 12 can be applied to KRR with the arc-cosine kernel when the Gaussian design assumption is not valid. In the noiseless setting ($\sigma_{\text{true}} = 0$) with constant regularization ($t = 0$), Theorem 12 implies that the mean squared error behaves as $\Theta(n^{\frac{1-2\beta}{\alpha}})$. This recovers a result in Bordelon et al. (2020, §2.2).*

*Our upper bound in Theorem 12 matches with the ones derived in (Steinwart et al., 2009; Fischer and Steinwart, 2020). Steinwart et al. (2009) and Fischer and Steinwart (2020) also derived algorithm independent minmax lower bounds. In contrast to their results, our Theorem 12 gives lower bounds for different regularization strengths $\lambda$.*

## 4 EXPERIMENTS

We illustrate our theory on a few toy experiments. We let the input $x$ be uniformly distributed on a unit circle, i.e., $\Omega = S^1$ and $\rho = \mathcal{U}(S^1)$. The points on $S^1$ can be represented by $x = (\cos\theta, \sin\theta)$ where $\theta \in [-\pi, \pi)$. We use the first order arc-cosine kernel function without bias, $k_{\text{w/o bias}}^{(1)}(x_1, x_2) = \frac{1}{\pi}(\sin\psi + (\pi - \psi)\cos\psi)$, where $\psi = \langle x_1, x_2 \rangle$ is the angle between $x_1$ and $x_2$. Hence Assumption 4 is satisfied with $\alpha = 4$. We consider the target functions in Table 1, which satisfy Assumption 5 with the indicated $\beta$, and $\mu_0$ indicates whether the function lies in the span of eigenfunctions of the kernel. For each target we conduct GPR 20 times and report the mean and standard deviation of the normalized SC and the Bayesian generalization error in Figure 1, which agree with the asymptotics predicted in Theorems 7 and 9. The details of the experiments appear in Appendix A, where we also show more experiments confirming our theory for zero- and second- order arc-cosine kernels, with and without biases.

## 5 CONCLUSION

We described the learning curves for GPR for the case that the kernel and target function follow a power law. This setting is frequently encountered in kernel learning and relates to recent advances on neural networks. Our approach is based on a tight analysis of the concentration of the inner product of empirical eigenfunctions $\Phi^T \Phi$ around $nI$. This allowed us to obtain more general results with more

| | function value | $\beta$ | $\mu_0$ | $\mathbb{E}_{\boldsymbol{\epsilon}}F^0(D_n)$ | $\mathbb{E}_{\boldsymbol{\epsilon}}G(D_n)$ |
|---|---|---|---|---|---|
| $f_1$ | $\cos 2\theta$ | $+\infty$ | $0$ | $\Theta(n^{1/4})$ | $\Theta(n^{-3/4})$ |
| $f_2$ | $\theta^2$ | $2$ | $>0$ | $\Theta(n)$ | $\Theta(1)$ |
| $f_3$ | $(|\theta|-\pi/2)^2$ | $2$ | $0$ | $\Theta(n^{1/4})$ | $\Theta(n^{-3/4})$ |
| $f_4$ | $\begin{cases}\pi/2-\theta, & \theta\in[0,\pi)\\ -\pi/2-\theta, & \theta\in[-\pi,0)\end{cases}$ | $1$ | $0$ | $\Theta(n^{3/4})$ | $\Theta(n^{-1/4})$ |

Table 1: Target functions used in the experiments for the first order arc-cosine kernel without bias $k^{(1)}_{\text{w/o bias}}$, their values of $\beta$ and $\mu_0$, and theoretical rates for the normalized SC and the Bayesian generalization error from our theorems.

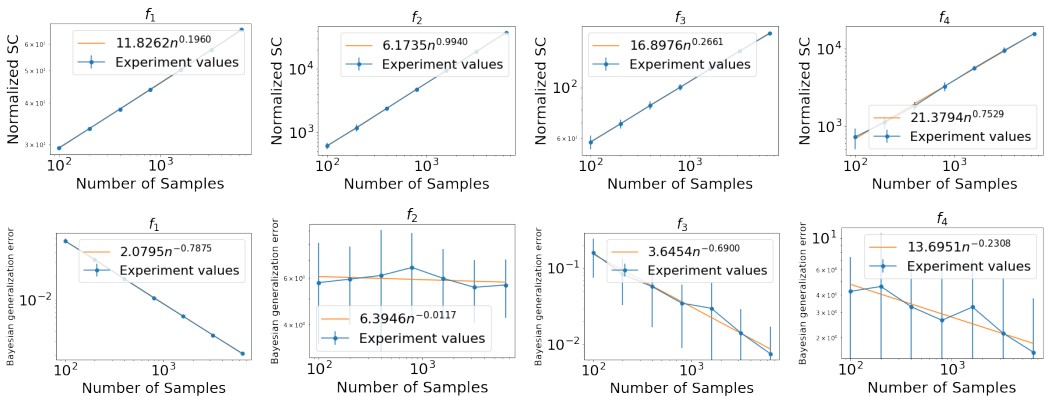

Figure 1: Normalized SC (top) and Bayesian generalization error (bottom) for GPR with the kernel $k^{(1)}_{\text{w/o bias}}$ and the target functions in Table 1. The orange curves show the linear regression fit for the experimental values (in blue) of the log Bayesian generalization error as a function of log $n$.

realistic assumptions than previous works. In particular, we recovered some results on learning curves for GPR and KRR previously obtained under more restricted settings (vide Remarks 10 and 14).

We showed that when $\beta \geq \alpha/2$, meaning that the target function has a compact representation in terms of the eigenfunctions of the kernel, the learning rate is as good as in the correctly specified case. In addition, our result allows us to interpret $\beta$ from a spectral bias perspective. When $\frac{1}{2} < \beta \leq \frac{\alpha}{2}$, the larger the value of $\beta$, the faster the decay of the generalization error. This implies that low-frequency functions are learned faster in terms of the number of training data points.

By leveraging the equivalence between GPR and KRR, we obtained a result on the generalization error of KRR. In the infinite-width limit, training fully-connected deep NNs with gradient descent and infinitesimally small learning rate under least-squared loss is equivalent to solving KRR with respect to the NTK (Jacot et al., 2018; Lee et al., 2019; Domingos, 2020), which in several cases is known to have a power-law spectrum (Velikanov and Yarotsky, 2021). Hence our methods can be applied to study the generalization error of infinitely wide neural networks. In future work, it would be interesting to estimate the values of $\alpha$ and $\beta$ for the NTK and the NNGP kernel of deep fully-connected or convolutional NNs and real data distributions and test our theory in these cases. Similarly, it would be interesting to consider extensions to finite width kernels.

## ACKNOWLEDGMENT

This project has received funding from the European Research Council (ERC) under the EU's Horizon 2020 research and innovation programme (grant agreement nº 757983).

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

# APPENDIX

## A EXPERIMENTS FOR ARC-COSINE KERNELS OF DIFFERENT ORDERS

In our experiment, the input space and input distribution are $\Omega = S^1$ and $\rho = \mathcal{U}(S^1)$, and we use the first order arc-cosine kernel function. (Cho and Saul, 2009) showed that this kernel is the conjugate kernel of an infinitely wide shallow ReLU network with two inputs and no biases in the hidden layer. GP regression with prior $\mathcal{GP}(0,k)$ corresponds to Bayesian training of this network (Lee et al., 2018). Under this setting, the eigenvalues and eigenfunctions are $\lambda_1 = \frac{4}{\pi^2}$, $\lambda_2 = \lambda_3 = \frac{1}{4}$, $\lambda_{2p} = \lambda_{2p+1} = \frac{4}{\pi^2((2p-2)^2-1)^2}$, $p \geq 2$ and $\phi_1(\theta) = 1$, $\phi_2(\theta) = \frac{\sqrt{2}}{2}\cos\theta$, $\phi_3(\theta) = \frac{\sqrt{2}}{2}\sin\theta$, $\phi_{2p}(\theta) = \frac{\sqrt{2}}{2}\cos(2p-2)\theta, \phi_{2p+1}(\theta) = \frac{\sqrt{2}}{2}\sin(2p-2)\theta$, $p \geq 2$. Hence Assumption 4 is satisfied with $\alpha = 4$, and the second part of Assumption 6 is satisfied with $\|\phi_p\| \leq \frac{\sqrt{2}}{2}, p \geq 1$.

The training and test data are generated as follows: We independently sample training inputs $x_1,...,x_n$ and test input $x_{n+1}$ from $\mathcal{U}(S^1)$ and training outputs $y_i$, $i = 1,...,n$ from $\mathcal{N}(f(x_i),\sigma^2)$, where we choose $\sigma = 0.1$. The Bayesian predictive density conditioned on the test point $x_{n+1}$ $\mathcal{N}(\bar{m}(x_{n+1}),\bar{k}(x_{n+1},x_{n+1}))$ is obtained by (1) and (2). We compute the normalized SC by (7) and the Bayesian generalization error by the Kullback-Leibler divergence between $\mathcal{N}(f(x_{n+1}),\sigma^2)$ and $\mathcal{N}(\bar{m}(x_{n+1}),\bar{k}(x_{n+1},x_{n+1}))$.

Next we present experiment results for arc-cosine kernels of different orders and arc-cosine kernels with biases. Consider the first order arc-cosine kernel function with biases,

$$k^{(1)}_{\text{w/ bias}}(x_1,x_2) = \frac{1}{\pi}(\sin\bar{\psi} + (\pi - \bar{\psi})\cos\bar{\psi}), \text{ where } \bar{\psi} = \arccos\left(\frac{1}{2}(\langle x_1,x_2\rangle + 1)\right). \quad (25)$$

Ronen et al. (2019) showed that this kernel is the conjugate kernel of an infinitely wide shallow ReLU network with two inputs and one hidden layer with biases, whose eigenvalues satisfy Assumption 4 with $\alpha = 4$. The eigenfunctions of this kernel are the same as that of the first-order arc-cosine kernel without biases, $k^{(1)}_{\text{w/o bias}}$ in Section 4. We consider the target functions in Table 3, which satisfy Assumption 5 with the indicated $\beta$, and $\mu_0$ indicates whether the function lies in the span of eigenfunctions of the kernel. For each target we conduct GPR 20 times and report the mean and standard deviation of the normalized SC and the Bayesian generalization error in Figure 3, which agree with the asymptotics predicted in Theorems 7 and 9.

Table 2 summarizes all the different kernel functions that we consider in our experiments with pointers to the corresponding tables and figures.

Summarizing the observations from these experiments, we see that the smoothness of the activation function (which is controlled by the order of the arc-cosine kernel) influences the decay rate $\alpha$ of the

| | kernel function | $\alpha$ | activation function | bias | pointer |
|---|---|---|---|---|---|
| $k^{(1)}_{\text{w/o bias}}$ | $\frac{1}{\pi}(\sin\psi+(\pi-\psi)\cos\psi)$ | 4 | $\max\{0,x\}$ | no | Table 1/Figure 1 |
| $k^{(1)}_{\text{w/ bias}}$ | $\frac{1}{\pi}(\sin\bar\psi+(\pi-\bar\psi)\cos\bar\psi)$ | 4 | $\max\{0,x\}$ | yes | Table 3/Figure 3 |
| $k^{(2)}_{\text{w/o bias}}$ | $\frac{1}{\pi}(3\sin\psi\cos\psi+(\pi-\psi)(1+2\cos^2\psi))$ | 6 | $(\max\{0,x\})^2$ | no | Table 4/Figure 4 |
| $k^{(2)}_{\text{w/ bias}}$ | $\frac{1}{\pi}(3\sin\bar\psi\cos\bar\psi+(\pi-\bar\psi)(1+2\cos^2\bar\psi))$ | 6 | $(\max\{0,x\})^2$ | yes | Table 5/Figure 5 |
| $k^{(0)}_{\text{w/o bias}}$ | $\frac{1}{\pi}(\sin\psi+(\pi-\psi)\cos\psi)$ | 2 | $\frac{1}{2}(1+\text{sign}(x))$ | no | Table 6/Figure 6 |
| $k^{(0)}_{\text{w/ bias}}$ | $\frac{1}{\pi}(\sin\bar\psi+(\pi-\psi)\cos\bar\psi)$ | 2 | $\frac{1}{2}(1+\text{sign}(x))$ | yes | Table 7/Figure 7 |

Table 2: The different kernel functions used in our experiments, their values of $\alpha$, the corresponding neural network activation function along with a pointer to the tables showing the target functions used for the kernels and the corresponding figures.

eigenvalues. In general, when the activation function is smoother, the decay rate $\alpha$ is larger. Theorem 9 then implies that smooth activation functions are more capable in suppressing noise but slower in learning the target. We also observe that networks with biases are more capable at learning functions compared to networks without bias. For example, the function $\cos(2\theta)$ cannot be learned by the zero order arc-cosine kernel without biases (see Table 6 and Figure 6), but it can be learned by the zero order arc-cosine kernel with biases (see Table 7 and Figure 7).

| | function value | $\beta$ | $\mu_0$ | $\mathbb{E}_\epsilon F^0(D_n)$ | $\mathbb{E}_\epsilon G(D_n)$ |
|---|---|---|---|---|---|
| $f_1$ | $\cos 2\theta$ | $+\infty$ | 0 | $\Theta(n^{1/4})$ | $\Theta(n^{-3/4})$ |
| $f_2$ | $\theta^2$ | 2 | 0 | $\Theta(n^{1/4})$ | $\Theta(n^{-3/4})$ |
| $f_3$ | $(|\theta|-\pi/2)^2$ | 2 | 0 | $\Theta(n^{1/4})$ | $\Theta(n^{-3/4})$ |
| $f_4$ | $\begin{cases}\pi/2-\theta, & \theta\in[0,\pi)\\ -\pi/2-\theta, & \theta\in[-\pi,0)\end{cases}$ | 1 | 0 | $\Theta(n^{3/4})$ | $\Theta(n^{-1/4})$ |

Table 3: Target functions used in the experiments for the first order arc-cosine kernel with bias, $k^{(1)}_{\text{w/ bias}}$, their values of $\beta$ and $\mu_0$, and theoretical rates for the normalized SC and the Bayesian generalization error from our theorems.

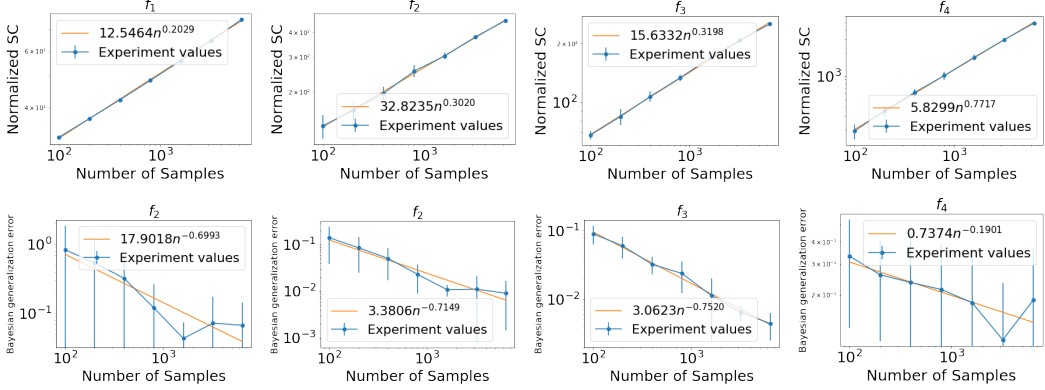

Figure 3: Normalized SC (top) and Bayesian generalization error (bottom) for GPR with kernel $k^{(1)}_{\text{w/ bias}}$ and the target functions in Table 3. The orange curves show the linear regression fit for the experimental values (in blue) of the log Bayesian generalization error as a function of $\log n$.

| | function value | $\beta$ | $\mu_0$ | $\mathbb{E}_\epsilon F^0(D_n)$ | $\mathbb{E}_\epsilon G(D_n)$ |
|---|---|---|---|---|---|
| $f_1$ | $\cos 2\theta$ | $+\infty$ | 0 | $\Theta(n^{1/6})$ | $\Theta(n^{-5/6})$ |
| $f_2$ | $\mathrm{sign}(\theta)$ | 1 | 0 | $\Theta(n^{5/6})$ | $\Theta(n^{-1/6})$ |
| $f_3$ | $\pi/2 - |\theta|$ | 2 | 0 | $\Theta(n^{1/2})$ | $\Theta(n^{-1/2})$ |
| $f_4$ | $\begin{cases} \pi/2 - \theta, & \theta \in [0,\pi) \\ -\pi/2 - \theta, & \theta \in [-\pi, 0) \end{cases}$ | 1 | $>0$ | $\Theta(n)$ | $\Theta(1)$ |

Table 4: Target functions used in the experiments for the second order arc-cosine kernel without bias, $k^{(2)}_{\mathrm{w/o\ bias}}$, their values of $\beta$ and $\mu_0$, and theoretical rates for the normalized SC and the Bayesian generalization error from our theorems.

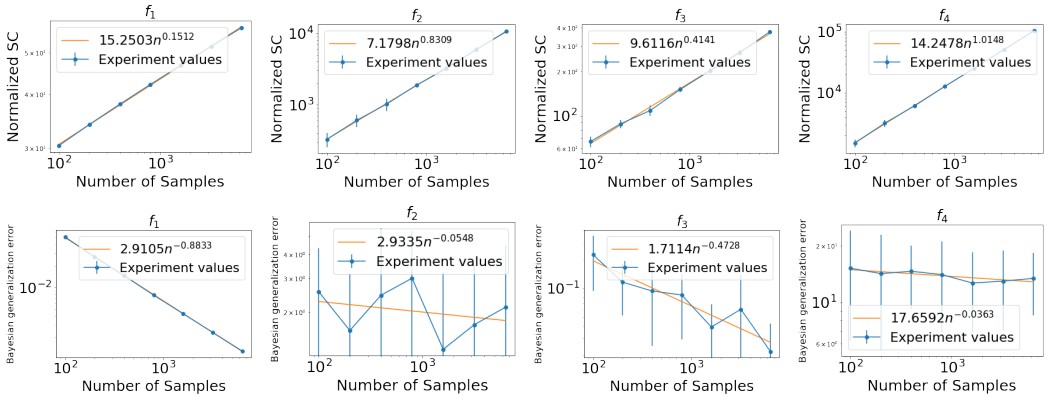

Figure 4: Normalized SC (top) and Bayesian generalization error (bottom) for GPR with kernel $k^{(2)}_{\mathrm{w/o\ bias}}$ and the target functions in Table 4.

| | function value | $\beta$ | $\mu_0$ | $\mathbb{E}_\epsilon F^0(D_n)$ | $\mathbb{E}_\epsilon G(D_n)$ |
|---|---|---|---|---|---|
| $f_1$ | $\cos 2\theta$ | $+\infty$ | 0 | $\Theta(n^{1/6})$ | $\Theta(n^{-5/6})$ |
| $f_2$ | $\theta^2$ | 2 | 0 | $\Theta(n^{1/2})$ | $\Theta(n^{-1/2})$ |
| $f_3$ | $(|\theta| - \pi/2)^2$ | 2 | 0 | $\Theta(n^{1/2})$ | $\Theta(n^{-1/2})$ |
| $f_4$ | $\begin{cases} \pi/2 - \theta, & \theta \in [0,\pi) \\ -\pi/2 - \theta, & \theta \in [-\pi, 0) \end{cases}$ | 1 | 0 | $\Theta(n^{5/6})$ | $\Theta(n^{-1/6})$ |

Table 5: Target functions used in the experiments for the second order arc-cosine kernel with bias, $k^{(2)}_{\mathrm{w/\ bias}}$, their values of $\beta$ and $\mu_0$, and theoretical rates for the normalized SC and the Bayesian generalization error from our theorems.

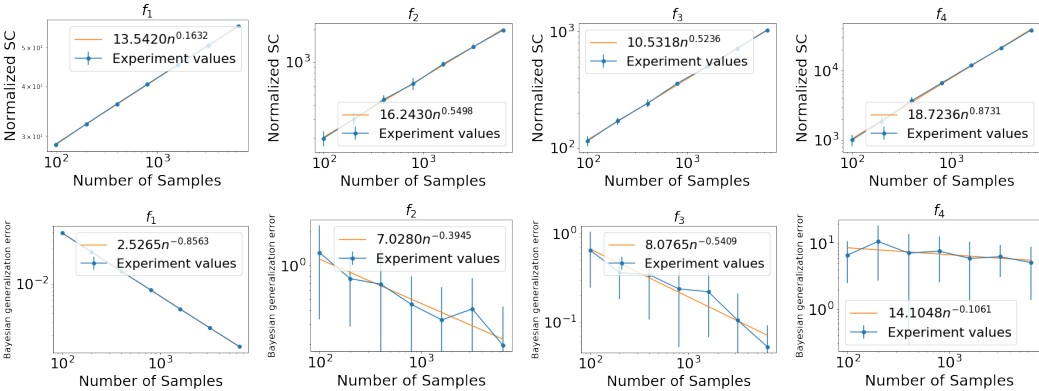

Figure 5: Normalized SC (top) and Bayesian generalization error (bottom) for GPR with kernel $k^{(2)}_{\mathrm{w/\ bias}}$ and the target functions in Table 5.

|  | function value | $\beta$ | $\mu_0$ | $\mathbb{E}_\epsilon F^0(D_n)$ | $\mathbb{E}_\epsilon G(D_n)$ |
|---|---|---|---|---|---|
| $f_1$ | $\cos 2\theta$ | $+\infty$ | $>0$ | $\Theta(n)$ | $\Theta(1)$ |
| $f_2$ | $\mathrm{sign}(\theta)$ | $1$ | $0$ | $\Theta(n^{1/2})$ | $\Theta(n^{-1/2})$ |
| $f_3$ | $\pi/2-|\theta|$ | $2$ | $0$ | $\Theta(n^{1/2})$ | $\Theta(n^{-1/2})$ |
| $f_4$ | $\begin{cases} \pi/2-\theta, & \theta\in[0,\pi) \\ -\pi/2-\theta, & \theta\in[-\pi,0) \end{cases}$ | $1$ | $>0$ | $\Theta(n)$ | $\Theta(1)$ |

Table 6: Target functions used in the experiments for the zero order arc-cosine kernel without bias, $k^{(0)}_{\mathrm{w/o\ bias}}$, their values of $\beta$ and $\mu_0$, and theoretical rates for the normalized SC and the Bayesian generalization error from our theorems.

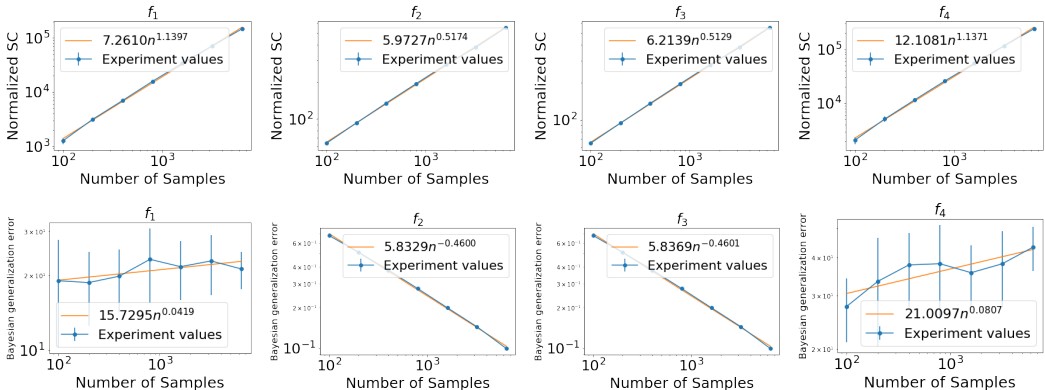

Figure 6: Normalized SC (top) and Bayesian generalization error (bottom) for GPR with kernel $k^{(0)}_{\mathrm{w/o\ bias}}$ and the target functions in Table 6.

|  | function value | $\beta$ | $\mu_0$ | $\mathbb{E}_\epsilon F^0(D_n)$ | $\mathbb{E}_\epsilon G(D_n)$ |
|---|---|---|---|---|---|
| $f_1$ | $\cos 2\theta$ | $+\infty$ | $0$ | $\Theta(n^{1/2})$ | $\Theta(n^{-1/2})$ |
| $f_2$ | $\theta^2$ | $2$ | $0$ | $\Theta(n^{1/2})$ | $\Theta(n^{-1/2})$ |
| $f_3$ | $(|\theta|-\pi/2)^2$ | $2$ | $0$ | $\Theta(n^{1/2})$ | $\Theta(n^{-1/2})$ |
| $f_4$ | $\begin{cases} \pi/2-\theta, & \theta\in[0,\pi) \\ -\pi/2-\theta, & \theta\in[-\pi,0) \end{cases}$ | $1$ | $0$ | $\Theta(n^{1/2})$ | $\Theta(n^{-1/2})$ |

Table 7: Target functions used in the experiments for the zero order arc-cosine kernel with bias, $k^{(0)}_{\mathrm{w/\ bias}}$, their values of $\beta$ and $\mu_0$, and theoretical rates for the normalized SC and the Bayesian generalization error from our theorems.

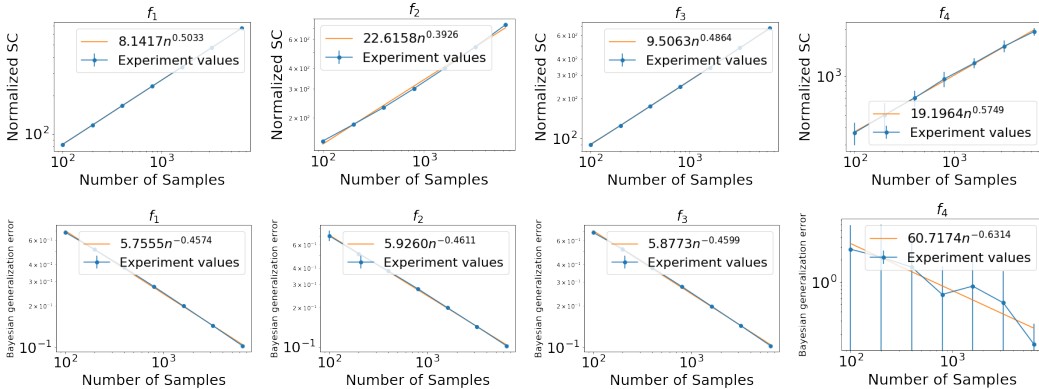

Figure 7: Normalized SC (top) and Bayesian generalization error (bottom) for GPR with kernel $k^{(0)}_{\mathrm{w/\ bias}}$ and the target functions in Table 7.

## B    PROOFS RELATED TO THE MARGINAL LIKELIHOOD

*Proof of Proposition 3.* Let $\bar{\mathbf{y}} = (\bar{y}_1,...,\bar{y}_n)^T$ be the outputs of the GP regression model on training inputs $\mathbf{x}$. Under the GP prior, the prior distribution of $\bar{\mathbf{y}}$ is $\mathcal{N}(0,K_n)$. Then the evidence of the model is given as follows:

$$
\begin{aligned}
Z_n &= \int_{\mathbb{R}^n} \left( \prod_{i=1}^n \frac{1}{\sqrt{2\pi}\sigma} e^{-\frac{(\bar{y}_i - y_i)^2}{2\sigma^2}} \right) \frac{1}{(2\pi)^{n/2}\det(K_n)^{1/2}} e^{-\frac{1}{2}\bar{\mathbf{y}}^T K_n^{-1}\bar{\mathbf{y}}} \mathrm{d}\bar{\mathbf{y}} \\
&= \frac{1}{(2\pi)^n \sigma^n \det(K_n)^{1/2}} \int_{\mathbb{R}^n} e^{-\frac{1}{2}\bar{\mathbf{y}}^T (K_n^{-1} + \frac{1}{\sigma^2}I)\bar{\mathbf{y}} + \frac{1}{\sigma^2}\bar{\mathbf{y}}^T \mathbf{y} - \frac{1}{2\sigma^2}\mathbf{y}^T \mathbf{y}} \mathrm{d}\bar{\mathbf{y}}.
\end{aligned}
\tag{26}
$$

Letting $\tilde{K}_n^{-1} = K_n^{-1} + \frac{1}{\sigma^2}I$ and $\mu = \frac{1}{\sigma^2}\tilde{K}_n\mathbf{y}$, we have

$$
\begin{aligned}
Z_n &= \frac{1}{(2\pi)^n \sigma^n \det(K_n)^{1/2}} \int_{\mathbb{R}^n} e^{-\frac{1}{2}(\bar{\mathbf{y}} - \mu)^T \tilde{K}_n^{-1}(\bar{\mathbf{y}} - \mu) - \frac{1}{2\sigma^2}\mathbf{y}^T \mathbf{y} + \frac{1}{2}\mu^T \tilde{K}_n^{-1}\mu} \mathrm{d}\bar{\mathbf{y}} \\
&= \frac{1}{(2\pi)^n \sigma^n \det(K_n)^{1/2}} (2\pi)^{n/2}\det(\tilde{K}_n)^{1/2} e^{-\frac{1}{2\sigma^2}\mathbf{y}^T \mathbf{y} + \frac{1}{2}\mu^T \tilde{K}_n^{-1}\mu} \\
&= \frac{\det(\tilde{K}_n)^{1/2}}{(2\pi)^{n/2}\sigma^n \det(K_n)^{1/2}} e^{-\frac{1}{2\sigma^2}\mathbf{y}^T \mathbf{y} + \frac{1}{2}\mu^T \tilde{K}_n^{-1}\mu}.
\end{aligned}
\tag{27}
$$

The normalized evidence is

$$
\begin{aligned}
Z_n^0 &= \frac{Z_n}{(2\pi)^{-n/2}\sigma^{-n} e^{-\frac{1}{2\sigma^2}(\mathbf{y} - f(\mathbf{x}))^T (\mathbf{y} - f(\mathbf{x}))}} \\
&= \frac{\det(\tilde{K}_n)^{1/2}}{\det(K_n)^{1/2}} e^{-\frac{1}{2\sigma^2}\mathbf{y}^T \mathbf{y} + \frac{1}{2}\mu^T \tilde{K}_n^{-1}\mu + \frac{1}{2\sigma^2}(\mathbf{y} - f(\mathbf{x}))^T (\mathbf{y} - f(\mathbf{x}))}.
\end{aligned}
\tag{28}
$$

So the normalized stochastic complexity is

$$
\begin{aligned}
F^0(D_n) &= -\log Z_n^0 \\
&= -\frac{1}{2}\mathrm{logdet}(\tilde{K}_n)^{1/2} + \frac{1}{2}\mathrm{logdet}(K_n)^{1/2} + \frac{1}{2\sigma^2}\mathbf{y}^T \mathbf{y} - \frac{1}{2}\mu^T \tilde{K}_n^{-1}\mu - \frac{1}{2\sigma^2}(\mathbf{y} - f(\mathbf{x}))^T (\mathbf{y} - f(\mathbf{x})) \\
&= -\frac{1}{2}\mathrm{logdet}(K_n^{-1} + \frac{1}{\sigma^2}I)^{-1} + \frac{1}{2}\mathrm{logdet}(K_n) + \frac{1}{2\sigma^2}\mathbf{y}^T \mathbf{y} - \frac{1}{2\sigma^4}\mathbf{y}^T (K_n^{-1} + \frac{1}{\sigma^2}I)^{-1}\mathbf{y} \\
&\quad - \frac{1}{2\sigma^2}(\mathbf{y} - f(\mathbf{x}))^T (\mathbf{y} - f(\mathbf{x})) \\
&= \frac{1}{2}\mathrm{logdet}(I + \frac{K_n}{\sigma^2}) + \frac{1}{2\sigma^2}\mathbf{y}^T (I + \frac{K_n}{\sigma^2})^{-1}\mathbf{y} - \frac{1}{2\sigma^2}(\mathbf{y} - f(\mathbf{x}))^T (\mathbf{y} - f(\mathbf{x})). \\
&= \frac{1}{2}\mathrm{logdet}(I + \frac{K_n}{\sigma^2}) + \frac{1}{2\sigma^2} f(\mathbf{x})^T (I + \frac{K_n}{\sigma^2})^{-1} f(\mathbf{x}) + \frac{1}{2\sigma^2}\epsilon^T (I + \frac{K_n}{\sigma^2})^{-1}\epsilon - \frac{1}{2\sigma^2}\epsilon^T \epsilon \\
&\quad + \frac{1}{2\sigma^2}\epsilon^T (I + \frac{K_n}{\sigma^2})^{-1} f(\mathbf{x})
\end{aligned}
$$

.

$$
\tag{29}
$$

After taking the expectation over noises $\epsilon$, we get

$$
\mathbb{E}_\epsilon F^0(D_n) = \frac{1}{2}\mathrm{logdet}(I + \frac{K_n}{\sigma^2}) + \frac{1}{2\sigma^2} f(\mathbf{x})^T (I + \frac{K_n}{\sigma^2})^{-1} f(\mathbf{x}) - \frac{1}{2}\mathrm{Tr}(I - (I + \frac{K_n}{\sigma^2})^{-1}).
\tag{30}
$$

This concludes the proof.                                                                               $\square$

## C    HELPER LEMMAS

**Lemma 15.** *Assume that $m \to \infty$ as $n \to \infty$. Given constants $a_1, a_2, s_1, s_2 > 0$, if $s_1 > 1$ and $s_2 s_3 > s_1 - 1$, we have that*

$$
\sum_{i=1}^R \frac{a_1 i^{-s_1}}{(1 + a_2 m i^{-s_2})^{s_3}} = \Theta(m^{\frac{1-s_1}{s_2}}).
\tag{31}
$$

*If $s_1 > 1$ and $s_2 s_3 = s_1 - 1$, we have that*

$$\sum_{i=1}^{R} \frac{a_1 i^{-s_1}}{(1 + a_2 m i^{-s_2})^{s_3}} = \Theta(m^{-s_3} \log m). \tag{32}$$

*If $s_1 > 1$ and $s_2 s_3 < s_1 - 1$, we have that*

$$\sum_{i=1}^{R} \frac{a_1 i^{-s_1}}{(1 + a_2 m i^{-s_2})^{s_3}} = \Theta(m^{-s_3}). \tag{33}$$

*Overall, if $s_1 > 1$ and $m \to \infty$,*

$$\sum_{i=1}^{R} \frac{a_1 i^{-s_1}}{(1 + a_2 m i^{-s_2})^{s_3}} = \begin{cases} \Theta(m^{\max\{-s_3, \frac{1-s_1}{s_2}\}}), & s_2 s_3 \neq s_1 - 1, \\ \Theta(m^{\frac{1-s_1}{s_2}} \log m), & s_2 s_3 = s_1 - 1. \end{cases} \tag{34}$$

*Proof of Lemma 15.* First, when $s_1 > 1$ and $s_2 s_3 > s_1 - 1$, we have that

$$\sum_{i=1}^{R} \frac{a_1 i^{-s_1}}{(1 + a_2 m i^{-s_2})^{s_3}} \leq \frac{a_1}{(1 + a_2 m)^{s_3}} + \int_{[1, +\infty]} \frac{a_1 x^{-s_1}}{(1 + a_2 m x^{-s_2})^{s_3}} \mathrm{d}x$$

$$= \frac{a_1}{(1 + a_2 m)^{s_3}} + m^{\frac{1-s_1}{s_2}} \int_{[1, +\infty]} \frac{a_1 \left(\frac{x}{m^{1/s_2}}\right)^{-s_1}}{(1 + a_2 \left(\frac{x}{m^{1/s_2}}\right)^{-s_2})^{s_3}} \mathrm{d}\frac{x}{m^{1/s_2}}$$

$$= \frac{a_1}{(1 + a_2 m)^{s_3}} + m^{\frac{1-s_1}{s_2}} \int_{[1/m^{1/s_2}, +\infty]} \frac{a_1 x^{-s_1}}{(1 + a_2 x^{-s_2})^{s_3}} \mathrm{d}x$$

$$= \Theta(m^{\frac{1-s_1}{s_2}}).$$

On the other hand, we have

$$\sum_{i=1}^{R} \frac{a_1 i^{-s_1}}{(1 + a_2 m i^{-s_2})^{s_3}} \geq \int_{[1, R+1]} \frac{a_1 x^{-s_1}}{(1 + a_2 m x^{-s_2})^{s_3}} \mathrm{d}x$$

$$= m^{\frac{1-s_1}{s_2}} \int_{[1, R+1]} \frac{a_1 \left(\frac{x}{m^{1/s_2}}\right)^{-s_1}}{(1 + a_2 \left(\frac{x}{m^{1/s_2}}\right)^{-s_2})^{s_3}} \mathrm{d}\frac{x}{m^{1/s_2}}$$

$$= m^{\frac{1-s_1}{s_2}} \int_{[1/m^{1/s_2}, (R+1)/m^{1/s_2}]} \frac{a_1 x^{-s_1}}{(1 + a_2 x^{-s_2})^{s_3}} \mathrm{d}x$$

$$= \Theta(m^{\frac{1-s_1}{s_2}}).$$

Second, when $s_1 > 1$ and $s_2 s_3 = s_1 - 1$, we have that

$$\sum_{i=1}^{R} \frac{a_1 i^{-s_1}}{(1 + a_2 m i^{-s_2})^{s_3}} \leq \frac{a_1}{(1 + a_2 m)^{s_3}} + m^{\frac{1-s_1}{s_2}} \int_{[1/m^{1/s_2}, +\infty]} \frac{a_1 x^{-s_1}}{(1 + a_2 x^{-s_2})^{s_3}} \mathrm{d}x$$

$$\leq \frac{a_1}{(1 + a_2 m)^{s_3}} + m^{\frac{1-s_1}{s_2}} O(\log m^{(1/s_2)})$$

$$= \Theta(m^{\frac{1-s_1}{s_2}} \log n).$$

On the other hand, we have

$$\sum_{i=1}^{R} \frac{a_1 i^{-s_1}}{(1 + a_2 m i^{-s_2})^{s_3}} \geq \int_{[1, R+1]} \frac{a_1 x^{-s_1}}{(1 + a_2 m x^{-s_2})^{s_3}} \mathrm{d}x$$

$$= m^{\frac{1-s_1}{s_2}} \int_{[1, R+1]} \frac{a_1 \left(\frac{x}{m^{1/s_2}}\right)^{-s_1}}{(1 + a_2 \left(\frac{x}{m^{1/s_2}}\right)^{-s_2})^{s_3}} \mathrm{d}\frac{x}{m^{1/s_2}}$$

$$= m^{\frac{1-s_1}{s_2}} \int_{[1/m^{1/s_2}, (R+1)/m^{1/s_2}]} \frac{a_1 x^{-s_1}}{(1 + a_2 x^{-s_2})^{s_3}} \mathrm{d}x$$

$$= \Theta(m^{\frac{1-s_1}{s_2}} \log n).$$

Third, when $s_1 > 1$ and $s_2 s_3 < s_1 - 1$, we have that

$$\sum_{i=1}^{R} \frac{a_1 i^{-s_1}}{(1+a_2 m i^{-s_2})^{s_3}} \leq \frac{a_1}{(1+a_2 m)^{s_3}} + m^{\frac{1-s_1}{s_2}} \int_{[1/m^{1/s_2}, +\infty]} \frac{a_1 x^{-s_1}}{(1+a_2 x^{-s_2})^{s_3}} \mathrm{d}x$$

$$\leq \frac{a_1}{(1+a_2 m)^{s_3}} + m^{\frac{1-s_1}{s_2}} \Theta(m^{(-1/s_2)(1-s_1+s_2 s_3)})$$

$$= \Theta(m^{-s_3}).$$

On the other hand, we have

$$\sum_{i=1}^{R} \frac{a_1 i^{-s_1}}{(1+a_2 m i^{-s_2})^{s_3}} \leq \frac{a_1}{(1+a_2 m)^{s_3}} + m^{\frac{1-s_1}{s_2}} \int_{[2/m^{1/s_2}, (R+1)/m^{1/s_2}]} \frac{a_1 x^{-s_1}}{(1+a_2 x^{-s_2})^{s_3}} \mathrm{d}x$$

$$\leq \frac{a_1}{(1+a_2 m)^{s_3}} + m^{\frac{1-s_1}{s_2}} \Theta(m^{(-1/s_2)(1-s_1+s_2 s_3)})$$

$$= \Theta(m^{-s_3}).$$

Overall, if $s_1 > 1$,

$$\sum_{i=1}^{R} \frac{a_1 i^{-s_1}}{(1+a_2 m i^{-s_2})^{s_3}} = \begin{cases} \Theta(m^{\max\{-s_3, \frac{1-s_1}{s_2}\}}), & s_2 s_3 \neq s_1 - 1, \\ \Theta(m^{-s_3} \log n), & s_2 s_3 = s_1 - 1. \end{cases} \tag{35}$$

$\square$

**Lemma 16.** *Assume that $R = m^{\frac{1}{s_2}+\kappa}$ for $\kappa > 0$. Given constants $a_1, a_2, s_1, s_2 > 0$, if $s_1 \leq 1$, we have that*

$$\sum_{i=1}^{R} \frac{a_1 i^{-s_1}}{(1+a_2 m i^{-s_2})^{s_3}} = \tilde{O}(\max\{m^{-s_3}, R^{1-s_1}\}). \tag{36}$$

*Proof of Lemma 16.* First, when $s_1 \leq 1$ and $s_2 s_3 > s_1 - 1$, we have that

$$\sum_{i=1}^{R} \frac{a_1 i^{-s_1}}{(1+a_2 m i^{-s_2})^{s_3}} \leq \frac{a_1}{(1+a_2 m)^{s_3}} + \int_{[1,R]} \frac{a_1 x^{-s_1}}{(1+a_2 m x^{-s_2})^{s_3}} \mathrm{d}x$$

$$= \frac{a_1}{(1+a_2 m)^{s_3}} + m^{\frac{1-s_1}{s_2}} \int_{[1,R]} \frac{a_1 (\frac{x}{m^{1/s_2}})^{-s_1}}{(1+a_2(\frac{x}{m^{1/s_2}})^{-s_2})^{s_3}} \mathrm{d} \frac{x}{m^{1/s_2}}$$

$$= \frac{a_1}{(1+a_2 m)^{s_3}} + m^{\frac{1-s_1}{s_2}} \int_{[1/m^{1/s_2}, R/m^{1/s_2}]} \frac{a_1 x^{-s_1}}{(1+a_2 x^{-s_2})^{s3}} \mathrm{d}x$$

$$= \frac{a_1}{(1+a_2 m)^{s_3}} + \tilde{O}(m^{\frac{1-s_1}{s_2}} (\frac{R}{m^{1/s_2}})^{1-s_1})$$

$$= \tilde{O}(\max\{m^{-s_3}, R^{1-s_1}\}).$$

Second, when $s_1 \leq 1$ and $s_2 s_3 \leq s_1 - 1$, we have that

$$\sum_{i=1}^{R} \frac{a_1 i^{-s_1}}{(1+a_2 m i^{-s_2})^{s_3}} \leq \frac{a_1}{(1+a_2 m)^{s_3}} + m^{\frac{1-s_1}{s_2}} \int_{[1/m^{1/s_2}, R/m^{1/s_2}]} \frac{a_1 x^{-s_1}}{(1+a_2 x^{-s_2})^{s_3}} \mathrm{d}x$$

$$\leq \frac{a_1}{(1+a_2 m)^{s_3}} + m^{\frac{1-s_1}{s_2}} \tilde{O}(m^{(-1/s_2)(1-s_1+s_2 s_3)} + (\frac{R}{m^{1/s_2}})^{1-s_1})$$

$$= \tilde{O}(\max\{m^{-s_3}, R^{1-s_1}\}).$$

Overall, if $s_1 \leq 1$,

$$\sum_{i=1}^{R} \frac{a_1 i^{-s_1}}{(1+a_2 m i^{-s_2})^{s_3}} = \tilde{O}(\max\{m^{-s_3}, R^{1-s_1}\}). \tag{37}$$

$\square$

**Lemma 17.** *Assume that $f \in L^2(\Omega, \rho)$. Consider the random vector $f(\mathbf{x}) = (f(x_1), ..., f(x_n))^T$, where $x_1, ..., x_n$ are drawn i.i.d from $\rho$. Then with probability of at least $1 - \delta_1$, we have*

$$\|f(\mathbf{x})\|_2^2 = \sum_{i=1}^{n} f^2(x_i) = \tilde{O}\Big( (\tfrac{1}{\delta_1} + 1) n \|f\|_2^2 \Big),$$

*where $\|f\|_2^2 = \int_{x \in \Omega} f^2(x) \mathrm{d}\rho(x)$.*

*Proof of Lemma 17.* Given a positive number $C \geq \|f\|_2^2$, applying Markov's inequality we have

$$\mathbb{P}(f^2(X) > C) \leq \frac{1}{C} \|f\|_2^2.$$

Let $A$ be the event that for all sample inputs $(x_i)_{i=1}^n$, $f^2(x_i) \leq C$. Then

$$\mathbb{P}(A) \geq 1 - n\mathbb{P}(f^2(X) > C) \geq 1 - \frac{1}{C} n \|f\|_2^2. \tag{38}$$

Define $\bar{f}^2(x) = \min\{f^2(x), C\}$. Then $\mathbb{E}\bar{f}^2(X) \leq \mathbb{E}f^2(X) = \|f\|_2^2$. So $|\bar{f}^2(X) - \mathbb{E}\bar{f}^2(X)| \leq \max\{C, \|f\|_2^2\} = C$ Since $0 \leq \bar{f}^2(x) \leq C$, we have

$$\mathbb{E}(\bar{f}^4(X)) \leq C\mathbb{E}(\bar{f}^2(X)) \leq C\|f\|_2^2. \tag{39}$$

So we have

$$\mathbb{E}|\bar{f}^2(X) - \mathbb{E}\bar{f}^2(X)|^2 \leq \mathbb{E}(\bar{f}^4(X)) \leq C\|f\|_2^2. \tag{40}$$

Applying Bernstein's inequality, we have

$$\mathbb{P}(\sum_{i=1}^{n} \bar{f}^2(x_i) > t + n\mathbb{E}\bar{f}^2(X)) \leq \exp\left( - \frac{t^2}{2(n\mathbb{E}|\bar{f}^2(X) - \mathbb{E}\bar{f}^2(X)|^2) + \frac{Ct}{3})} \right)$$

$$\leq \exp\left( - \frac{t^2}{2(nC\|f\|_2^2 + \frac{Ct}{3})} \right)$$

$$\leq \exp\left( - \frac{t^2}{4\max\{nC\|f\|_2^2, \frac{Ct}{3}\}} \right).$$

Hence, with probability of at least $1 - \delta_1/2$ we have

$$\sum_{i=1}^{n} \bar{f}^2(x_i) \leq \max\left\{ \sqrt{4C\log\frac{2}{\delta_1} n\|f\|_2^2}, \frac{4C}{3}\log\frac{2}{\delta_1} \right\} + n\mathbb{E}\bar{f}^2(X)$$

$$\leq \max\left\{ \sqrt{4C\log\frac{2}{\delta_1} n\|f\|_2^2}, \frac{4C}{3}\log\frac{2}{\delta_1} \right\} + n\|f\|_2^2. \tag{41}$$

When event $A$ happens, $f^2(x_i) = \bar{f}^2(x_i)$ for all sample inputs. According to (38) and (41), with probability at least $1 - \frac{1}{C} n\|f\|_2^2 - \delta_1/2$, we have

$$\sum_{i=1}^{n} f^2(x_i) = \sum_{i=1}^{n} \bar{f}^2(x_i) \leq \max\left\{ \sqrt{4C\log\frac{2}{\delta_1} n\|f\|_2^2}, \frac{4C}{3}\log\frac{2}{\delta_1} \right\} + n\|f\|_2^2.$$

Choosing $C = \frac{2}{\delta_1} n\|f\|_2^2$, with probability of at least $1 - \delta_1$ we have

$$\sum_{i=1}^{n} f^2(x_i) = \sum_{i=1}^{n} \bar{f}^2(x_i) \leq \max\left\{ \sqrt{\frac{8}{\delta_1}\log\frac{2}{\delta_1} n^2\|f\|_2^4}, \frac{8}{3\delta_1} n\|f\|_2^2\log\frac{2}{\delta_1} \right\} + n\|f\|_2^2 = \tilde{O}\Big( (\tfrac{1}{\delta_1} + 1) n\|f\|_2^2 \Big).$$

$\square$

**Lemma 18.** *Assume that $f \in L^2(\Omega, \rho)$. Consider the random vector $f(\mathbf{x}) = (f(x_1), ..., f(x_n))^T$, where $x_1, ..., x_n$ are drawn i.i.d from $\rho$. Assume that $\|f\|_\infty = \sup_{x \in \Omega} f(x) \leq C$. With probability of at least $1 - \delta_1$, we have*

$$\|f(\mathbf{x})\|_2^2 = \tilde{O}\Big( \sqrt{C^2 n\|f\|_2^2} + C^2 \Big) + n\|f\|_2^2,$$

*where $\|f\|_2^2 = \int_{x \in \Omega} f^2(x)\mathrm{d}\rho(x)$.*

*Proof of Lemma 18.* We have $|f^2(X) - \mathbb{E}f^2(X)| \leq \max\{C^2, \|f\|_2^2\} = C^2$ Since $0 \leq f^2(x) \leq C$, we have

$$\mathbb{E}(f^4(X)) \leq C^2 \mathbb{E}(f^2(X)) \leq C^2 \|f\|_2^2. \tag{42}$$

So we have

$$\mathbb{E}|f^2(X) - \mathbb{E}f^2(X)|^2 \leq \mathbb{E}(f^4(X)) \leq C^2 \|f\|_2^2. \tag{43}$$

Applying Bernstein's inequality, we have

$$\mathbb{P}(\sum_{i=1}^n f^2(x_i) > t + n\mathbb{E}f^2(X)) \leq \exp\left(-\frac{t^2}{2(n\mathbb{E}|f^2(X) - \mathbb{E}f^2(X)|^2) + \frac{C^2 t}{3})}\right)$$

$$\leq \exp\left(-\frac{t^2}{2(nC^2\|f\|_2^2 + \frac{C^2 t}{3})}\right)$$

$$\leq \exp\left(-\frac{t^2}{4\max\{nC^2\|f\|_2^2, \frac{C^2 t}{3}\}}\right).$$

Hence, with probability of at least $1 - \delta_1$ we have

$$\sum_{i=1}^n f^2(x_i) \leq \max\left\{\sqrt{4C^2 \log\frac{1}{\delta_1} n\|f\|_2^2}, \frac{4C^2}{3}\log\frac{1}{\delta_1}\right\} + n\mathbb{E}f^2(X)$$

$$\leq \tilde{O}\left(\max\left\{\sqrt{C^2 n\|f\|_2^2}, C^2\right\}\right) + n\|f\|_2^2 \tag{44}$$

$$\leq \tilde{O}\left(\sqrt{C^2 n\|f\|_2^2} + C^2\right) + n\|f\|_2^2.$$

$\square$

For the proofs in the reminder of this section, the definitions of the relevant quantities are given in Section 3.

**Corollary 19.** *With probability of at least $1 - \delta_1$, we have*

$$\|f_{>R}(\mathbf{x})\|_2^2 = \tilde{O}\left((\tfrac{1}{\delta_1} + 1)nR^{1-2\beta}\right).$$

*Proof of Corollary 19.* The $L_2$ norm of $f_{>R}(x)$ is given by $\|f_{>R}\|_2^2 = \sum_{p=R+1}^\infty \mu_p^2 \leq \frac{C_\mu}{2\beta - 1}R^{1-2\beta}$. Applying Lemma 17 we get the result. $\square$

**Corollary 20.** *For any $\nu \in \mathbb{R}^R$, with probability of at least $1 - \delta_1$ we have*

$$\|\Phi_R \nu\|_2^2 = \tilde{O}\left((\tfrac{1}{\delta_1} + 1)n\|\nu\|_2^2\right).$$

*Proof of Corollary 20.* Let $g(x) = \sum_{p=1}^R \nu_p \phi_p(x)$. Then $\Phi_R \nu = g(\mathbf{x})$. The $L_2$ norm of $g(x)$ is given by $\|g\|_2^2 = \sum_{p=1}^R \nu_p^2 = \|\nu\|_2^2$. Applying Lemma 17 we get the result. $\square$

Next we consider the quantity, $\Phi_R^T \Phi_R - nI$. The key tool that we use is the matrix Bernstein inequality that describes the upper tail of a sum of independent zero-mean random matrices.

**Lemma 21.** *Let $D = \text{diag}\{d_1, \ldots, d_R\}$, $d_1, \ldots, d_R > 0$ and $d_{\max} = \max\{d_1, \ldots, d_R\}$. Let $M = \max\{\sum_{p=0}^R d_p^2 \|\phi_p\|_\infty^2, d_{\max}^2\}$. Then with probability of at least $1 - \delta$, we have*

$$\|D(\Phi_R^T \Phi_R - nI)D\|_2 \leq \max\left\{\sqrt{nd_{\max}^2 M \log\frac{R}{\delta}}, M\log\frac{R}{\delta})\right\}. \tag{45}$$

*Proof of Lemma 21.* Let $Y_j = (\phi_1(x_j), \dots, \phi_R(x_j))^T$ and $Z_j = DY_j$. It is easy to verify that $\mathbb{E}(Z_j Z_j^T) = D^2$. Then the left hand side of (45) is $\sum_{j=1}^n [Z_j Z_j^T - \mathbb{E}(Z_j Z_j^T)]$. We note that

$$\|Z_j Z_j^T - \mathbb{E}(Z_j Z_j^T)\|_2 \leq \max\{\|Z_j Z_j^T\|_2, \|\mathbb{E}(Z_j Z_j^T)\|_2\} \leq \max\{\|Z_j\|_2^2, d_{\max}^2\}.$$

For $\|Z_j\|_2^2$, we have

$$\|Z_j\|_2^2 = \sum_{p=0}^R d_p^2 \phi_p^2(x_j) \leq \sum_{p=0}^R d_p^2 \|\phi_p\|_\infty^2, \tag{46}$$

we have

$$\|Z_j Z_j^T - \mathbb{E}(Z_j Z_j^T)\|_2 \leq \max\{\textstyle\sum_{p=0}^R d_p^2 \|\phi_p\|_\infty^2, d_{\max}^2\}.$$

On the other hand,

$$\mathbb{E}[(Z_j Z_j^T - \mathbb{E}(Z_j Z_j^T))^2] = \mathbb{E}[\|Z_j\|_2^2 Z_j Z_j^T] - (\mathbb{E}(Z_j Z_j^T))^2.$$

Since

$$\mathbb{E}[\|Z_j\|_2^2 Z_j Z_j^T] \preccurlyeq \mathbb{E}[\sum_{p=0}^R d_p^2 \|\phi_p\|_\infty^2 Z_j Z_j^T], \quad \text{(by (46))}$$

$$= \sum_{p=0}^R d_p^2 \|\phi_p\|_\infty^2 \mathbb{E}[Z_j Z_j^T],$$

we have

$$\|\mathbb{E}[(Z_j Z_j^T - \mathbb{E}(Z_j Z_j^T))^2]\|_2 \leq \max\{\textstyle\sum_{p=0}^R d_p^2 \|\phi_p\|_\infty^2 \|\mathbb{E}[Z_j Z_j^T]\|_2, d_{\max}^4\}$$

$$\leq \max\{\textstyle\sum_{p=0}^R d_p^2 \|\phi_p\|_\infty^2 d_{\max}^2, d_{\max}^4\}$$

$$\leq d_{\max}^2 \max\{\textstyle\sum_{p=0}^R d_p^2 \|\phi_p\|_\infty^2, d_{\max}^2\}.$$

Using the matrix Bernstein inequality ([Tropp, 2012](), Theorem 6.1), we have

$$\mathbb{P}(\|\sum_{j=1}^n [Z_j Z_j^T - \mathbb{E}(Z_j Z_j^T)]\|_2 > t)$$

$$\leq R \exp\left(\frac{-t^2}{2(n\|\mathbb{E}[(Z_j Z_j^T - \mathbb{E}(Z_j Z_j^T))^2]\|_2 + \frac{t \max_j \|Z_j Z_j^T - \mathbb{E}(Z_j Z_j^T)\|_2}{3})}\right)$$

$$\leq R \exp\left(\frac{-t^2}{2(n d_{\max}^2 \max\{\sum_{p=0}^R d_p^2 \|\phi_p\|_\infty^2, d_{\max}^2\} + \frac{t \max\{\sum_{p=0}^R d_p^2 \|\phi_p\|_\infty^2, d_{\max}^2\}}{3})}\right)$$

$$= R \exp\left(\frac{-t^2}{O(\max\{n d_{\max}^2 \max\{\sum_{p=0}^R d_p^2 \|\phi_p\|_\infty^2, d_{\max}^2\}, t \max\{\sum_{p=0}^R d_p^2 \|\phi_p\|_\infty^2, d_{\max}^2\}\})}\right).$$

Then with probability of at least $1 - \delta$, we have

$$\|\sum_{j=1}^n [Z_j Z_j^T - \mathbb{E}(Z_j Z_j^T)]\|_2$$

$$\leq \max\left\{\sqrt{n d_{\max}^2 \max\{\textstyle\sum_{p=0}^R d_p^2 \|\phi_p\|_\infty^2, d_{\max}^2\} \log \frac{R}{\delta}}, \max\{\textstyle\sum_{p=0}^R d_p^2 \|\phi_p\|_\infty^2, d_{\max}^2\} \log \frac{R}{\delta}\right\}.$$

$\square$

**Corollary 22.** *Suppose that the eigenvalues $(\lambda_p)_{p \geq 1}$ satisfy Assumption 4, and the eigenfunctions satisfy Assumption 6. Assume $\sigma^2 = \Theta(n^t)$ where $1 - \frac{\alpha}{1+2\tau} < t < 1$ Let $\gamma$ be a positive number such that $\frac{1+\alpha+2\tau-(1+2\tau+2\alpha)t}{2\alpha(1-t)} < \gamma \leq 1$. Then with probability of at least $1 - \delta$, we have*

$$\|\frac{1}{\sigma^2}(I + \frac{n}{\sigma^2}\Lambda_R)^{-\gamma/2}\Lambda_R^{\gamma/2}(\Phi_R^T \Phi_R - nI)\Lambda_R^{\gamma/2}(I + \frac{n}{\sigma^2}\Lambda_R)^{-\gamma/2}\|_2$$

$$\leq O\left(n^{\frac{1+\alpha+2\tau-(1+2\tau+2\alpha)t}{2\alpha}-\gamma(1-t)}\sqrt{\log \frac{R}{\delta}}\right). \tag{47}$$

*Proof of Corollary 22.* Use the same notation as in Lemma 21. Let $D = (I + \frac{n}{\sigma^2}\Lambda_R)^{-\gamma/2}\Lambda_R^{\gamma/2}$. Then $d_{\max}^2 \leq \frac{\sigma^{2\gamma}}{n^\gamma}$ and $\sum_{p=0}^R d_p^2\|\phi_p\|_\infty^2 \leq \sum_{p=0}^R C_\phi^2 \frac{\lambda_p^\gamma p^{2\tau}}{(1+\frac{n}{\sigma^2}\lambda_p)^\gamma} = O((\frac{n}{\sigma^2})^{\frac{1-\gamma\alpha+2\tau}{\alpha}})$, where the first inequality follows from Assumptions 4 and 6 and the last equality from Lemma 15. Then $M = \max\{\sum_{p=0}^R d_p^2\|\phi_p\|_\infty^2, d_{\max}^2\} = O((\frac{n}{\sigma^2})^{\frac{1-\gamma\alpha+2\tau}{\alpha}})$. Applying Lemma 21, we have

$$\|\frac{1}{\sigma^2}(I + \frac{n}{\sigma^2}\Lambda_R)^{-\gamma/2}\Lambda_R^{\gamma/2}(\Phi_R^T\Phi_R - nI)\Lambda_R^{\gamma/2}(I + \frac{n}{\sigma^2}\Lambda_R)^{-\gamma/2}\|_2$$

$$\leq \frac{1}{\sigma^2}\max\left\{\sqrt{n\frac{\sigma^{2\gamma}}{n^\gamma}O((\frac{n}{\sigma^2})^{\frac{1-\gamma\alpha+2\tau}{\alpha}})\log\frac{R}{\delta}}, O((\frac{n}{\sigma^2})^{\frac{1-\gamma\alpha+2\tau}{\alpha}})\log\frac{R}{\delta}\right\}$$

$$= O(\frac{1}{\sigma^2}(\frac{n}{\sigma^2})^{\frac{1-2\gamma\alpha+2\tau}{2\alpha}}n^{\frac{1}{2}}) = O(\sqrt{\log\frac{R}{\delta}}n^{\frac{(1-2\gamma\alpha+2\tau)(1-t)}{2\alpha}+\frac{1}{2}-t})$$

$$= O\left(\sqrt{\log\frac{R}{\delta}}n^{\frac{1+\alpha+2\tau}{2\alpha}-\frac{(1+2\tau+2\alpha)t}{2\alpha}-\gamma(1-t)}\right).$$

(48)

$\square$

**Corollary 23.** *Suppose that the eigenvalues $(\lambda_p)_{p\geq 1}$ satisfy Assumption 4, and the eigenfunctions satisfy Assumption 6. Let $\tilde{\Lambda}_{1,R} = \text{diag}\{1,\lambda_1,...,\lambda_R\}$. Assume $\sigma^2 = \Theta(n^t)$ where $t < 1$ Let $\gamma$ be a positive number such that $\frac{1+2\tau}{\alpha} < \gamma \leq 1$. Then with probability of at least $1-\delta$, we have*

$$\|(I + \frac{n}{\sigma^2}\Lambda_R)^{-\gamma/2}\tilde{\Lambda}_{1,R}^{\gamma/2}(\Phi_R^T\Phi_R - nI)\tilde{\Lambda}_{1,R}^{\gamma/2}(I + \frac{n}{\sigma^2}\Lambda_R)^{-\gamma/2}\|_2 \leq O\left(\sqrt{\log\frac{R}{\delta}}n^{\frac{1}{2}}\right). \quad (49)$$

*Proof of Corollary 23.* Use the same notation as in Lemma 21. Let $D = (I + \frac{n}{\sigma^2}\Lambda_R)^{-\gamma/2}\tilde{\Lambda}_{1,R}^{\gamma/2}$. Then $d_{\max}^2 \leq 1$ and $\sum_{p=0}^R d_p^2\|\phi_p\|_\infty^2 \leq C_\phi^2 + \sum_{p=1}^R C_\phi^2\frac{\lambda_p^\gamma p^{2\tau}}{(1+\frac{n}{\sigma^2}\lambda_p)^\gamma} = C_\phi^2 + O(n^{\frac{(1-\gamma\alpha+2\tau)(1-t)}{\alpha}}) = O(1)$ where the first inequality follows from Assumptions 4 and 6 and the second equality from Lemma 15. Then $M = \max\{\sum_{p=0}^R d_p^2\|\phi_p\|_\infty^2, d_{\max}^2\} = O(1)$. Applying Lemma 21, we have

$$\|(I + \frac{n}{\sigma^2}\Lambda_R)^{-\gamma/2}\Lambda_R^{\gamma/2}(\Phi_R^T\Phi_R - nI)\Lambda_R^{\gamma/2}(I + \frac{n}{\sigma^2}\Lambda_R)^{-\gamma/2}\|_2$$

$$\leq \max\left\{\sqrt{\log\frac{R}{\delta}nO(1)}, \log\frac{R}{\delta}O(1)\right\}$$

$$= O\left(\sqrt{\log\frac{R}{\delta}}n^{\frac{1}{2}}\right).$$

(50)

$\square$

**Corollary 24.** *Suppose that the eigenvalues $(\lambda_p)_{p\geq 1}$ satisfy Assumption 4, and the eigenfunctions satisfy Assumption 6. Let $\Phi_{R+1:S} = (\phi_{R+1}(\mathbf{x}),...,\phi_S(\mathbf{x}))$, and $\Lambda_{R+1:S} = (\lambda_{R+1},...,\lambda_S)$. Then with probability of at least $1-\delta$, we have*

$$\|\Lambda_{R+1:S}^{1/2}(\Phi_{R+1:S}^T\Phi_{R+1:S} - nI)\Lambda_{R+1:S}^{1/2}\|_2 \leq O\left(\log\frac{S-R}{\delta}\max\{n^{\frac{1}{2}}R^{\frac{1-2\alpha+2\tau}{2}}, R^{1-\alpha+2\tau}\}\right). \quad (51)$$

*Proof of Corollary 24.* Use the same notation as in Lemma 21. Let $D = \Lambda_{R+1:S}^{1/2}$. Then $d_{\max}^2 \leq \overline{C_\lambda}R^{-\alpha} = O(R^{-\alpha})$ and $\sum_{p=R+1}^S C_\phi^2 d_p^2 p^{2\tau} \leq \sum_{p=R+1}^S C_\phi^2\overline{C_\lambda}p^{-\alpha}p^{2\tau} = O(R^{1-\alpha+2\tau})$, where the first inequality follows from Assumptions 4 and 6. Then $M = \max\{\sum_{p=R+1}^S C_\phi^2 d_p^2 p^{2\tau}, d_{\max}^2\} = O(R^{1-\alpha+2\tau})$. Applying Lemma 21, we have

$$\|(I + \frac{n}{\sigma^2}\Lambda_R)^{-\gamma/2}\Lambda_R^{\gamma/2}(\Phi_R^T\Phi_R - nI)\Lambda_R^{\gamma/2}(I + \frac{n}{\sigma^2}\Lambda_R)^{-\gamma/2}\|_2$$

$$\leq \max\left\{\sqrt{\log\frac{S-R}{\delta}nO(R^{-\alpha})O(R^{1-\alpha+2\tau})}, \log\frac{S-R}{\delta}O(R^{1-\alpha+2\tau}))\right\}$$

$$= O\left(\log\frac{S-R}{\delta}\max\{n^{\frac{1}{2}}R^{\frac{1-2\alpha+2\tau}{2}}, R^{1-\alpha+2\tau}\}\right).$$

(52)

$\square$

**Lemma 25.** *Under the assumptions of Corollary 24, with probability of at least $1-\delta$, we have*

$$\|\Phi_{>R}\Lambda_{>R}\Phi_{>R}^T\|_2=\tilde{O}(\max\{nR^{-\alpha},n^{\frac{1}{2}}R^{\frac{1-2\alpha+2\tau}{2}},R^{1-\alpha+2\tau}\}).$$

*Proof of Lemma 25.* For $S\in\mathbb{N}$, we have

$$\|\Phi_{>S}\Lambda_{>S}\Phi_{>S}^T\|_2\le\sum_{p=S+1}^{\infty}\|\Lambda_p\phi_p(\mathbf{x})\phi_p(\mathbf{x})^T\|_2$$

$$=\sum_{p=S+1}^{\infty}\lambda_p\|\phi_p(\mathbf{x})\|_2^2$$

$$\le\sum_{p=S+1}^{\infty}\lambda_p nC_\phi^2 p^{2\tau}$$

$$=O(nS^{1-\alpha+2\tau}).$$

Let $S=R^{\frac{\alpha}{\alpha-1-2\tau}}$. Then we get $\|\Phi_{>S}\Lambda_{>S}\Phi_{>S}^T\|_2=O(nR^{-\alpha})$.

Let $\Phi_{R+1:S}=(\phi_{R+1}(\mathbf{x}),...,\phi_S(\mathbf{x}))$, $\Lambda_{R+1:S}=(\lambda_{R+1},...,\lambda_S)$. We then have

$$\|\Phi_{>R}\Lambda_{>R}\Phi_{>R}^T\|_2\le\|\Phi_{>S}\Lambda_{>S}\Phi_{>S}^T\|_2+\|\Phi_{R+1:S}\Lambda_{R+1:S}\Phi_{R+1:S}^T\|_2$$

$$\le O(nR^{-\alpha})+\|\Lambda_{R+1:S}^{1/2}\Phi_{R+1:S}^T\Phi_{R+1:S}\Lambda_{R+1:S}^{1/2}\|_2$$

$$\le O(nR^{-\alpha})+n\|\Lambda_{R+1:S}\|_2+\|\Lambda_{R+1:S}^{1/2}(\Phi_{R+1:S}^T\Phi_{R+1:S}-nI)\Lambda_{R+1:S}^{1/2}\|_2$$

$$\le O(nR^{-\alpha})+O(nR^{-\alpha})+O(\log\frac{R^{\frac{\alpha}{\alpha-1}}-R}{\delta}\max\{n^{\frac{1}{2}}R^{\frac{1-2\alpha+2\tau}{2}},R^{1-\alpha+2\tau}\})$$

$$=\tilde{O}(\max\{nR^{-\alpha},n^{\frac{1}{2}}R^{\frac{1-2\alpha+2\tau}{2}},R^{1-\alpha+2\tau}\}),$$

where in the fourth inequality we use Corollary 24. $\qquad\square$

**Corollary 26.** *Assume that $\sigma^2=\Theta(1)$. If $R=n^{\frac{1}{\alpha}+\kappa}$ where $0<\kappa<\frac{\alpha-1-2\tau}{\alpha(1+2\tau)}$, then with probability of at least $1-\delta$, we have*

$$\|(I+\frac{\Phi_R\Lambda_R\Phi_R^T}{\sigma^2})^{-1}\frac{\Phi_{>R}\Lambda_{>R}\Phi_{>R}^T}{\sigma^2}\|_2\le\|\frac{\Phi_{>R}\Lambda_{>R}\Phi_{>R}^T}{\sigma^2}\|_2=\tilde{O}(n^{-\kappa\alpha})=o(1).$$

*Proof of Corollary 26.* By Lemma 25 and the assumption $R=n^{\frac{1}{\alpha}+\kappa}$, we have

$$\|(I+\frac{\Phi_R\Lambda_R\Phi_R^T}{\sigma^2})^{-1}\frac{\Phi_{>R}\Lambda_{>R}\Phi_{>R}^T}{\sigma^2}\|_2\le\|\frac{\Phi_{>R}\Lambda_{>R}\Phi_{>R}^T}{\sigma^2}\|_2$$

$$\le\tilde{O}(\max\{nR^{-\alpha},n^{\frac{1}{2}}R^{\frac{1-2\alpha+2\tau}{2}},R^{1-\alpha+2\tau}\})$$

$$=\tilde{O}(n^{-\kappa\alpha}).$$

$\qquad\square$

**Lemma 27.** *Assume that $\|\frac{1}{\sigma^2}(I+\frac{n}{\sigma^2}\Lambda_R)^{-\gamma/2}\Lambda_R^{\gamma/2}(\Phi_R^T\Phi_R-nI)\Lambda_R^{\gamma/2}(I+\frac{n}{\sigma^2}\Lambda_R)^{-\gamma/2}\|_2<1$ where $\frac{1+2\tau}{\alpha}<\gamma\le1$. We then have*

$$(I+\frac{1}{\sigma^2}\Lambda_R\Phi_R^T\Phi_R)^{-1}$$

$$=(I+\frac{n}{\sigma^2}\Lambda_R)^{-1}+\sum_{j=1}^{\infty}(-1)^j\big(\frac{1}{\sigma^2}(I+\frac{n}{\sigma^2}\Lambda_R)^{-1}\Lambda_R(\Phi_R^T\Phi_R-nI)\big)^j(I+\frac{n}{\sigma^2}\Lambda_R)^{-1}.$$

*Proof of Lemma 27.* First note that

$$\|\frac{1}{\sigma^2}(I+\frac{n}{\sigma^2}\Lambda_R)^{-1/2}\Lambda_R^{1/2}(\Phi_R^T\Phi_R-nI)\Lambda_R^{1/2}(I+\frac{n}{\sigma^2}\Lambda_R)^{-1/2}\|_2$$

$$<\|\frac{1}{\sigma^2}(I+\frac{n}{\sigma^2}\Lambda_R)^{-\gamma/2}\Lambda_R^{\gamma/2}(\Phi_R^T\Phi_R-nI)\Lambda_R^{\gamma/2}(I+\frac{n}{\sigma^2}\Lambda_R)^{-\gamma/2}\|_2<1.$$

Let $\tilde{\Lambda}_{\epsilon,R} = \mathrm{diag}\{\epsilon,\lambda_1,...,\lambda_R\}$. Since $\Lambda_R = \mathrm{diag}\{0,\lambda_1,...,\lambda_R\}$, we have that when $\epsilon$ is sufficiently small, $\|\frac{1}{\sigma^2}(I+\frac{n}{\sigma^2}\tilde{\Lambda}_{\epsilon,R})^{-1/2}\tilde{\Lambda}_{\epsilon,R}^{1/2}(\Phi_R^T\Phi_R - nI)\tilde{\Lambda}_{\epsilon,R}^{1/2}(I+\frac{n}{\sigma^2}\tilde{\Lambda}_{\epsilon,R})^{-1/2}\|_2 < 1$. Since all diagonal entries of $\tilde{\Lambda}_{\epsilon,R}$ are positive, we have

$$(I+\tfrac{1}{\sigma^2}\tilde{\Lambda}_{\epsilon,R}\Phi_R^T\Phi_R)^{-1}$$

$$=(I+\tfrac{n}{\sigma^2}\tilde{\Lambda}_{\epsilon,R}+\tfrac{1}{\sigma^2}\tilde{\Lambda}_{\epsilon,R}(\Phi_R^T\Phi_R-nI))^{-1}$$

$$=\tilde{\Lambda}_{\epsilon,R}^{1/2}(I+\tfrac{n}{\sigma^2}\tilde{\Lambda}_{\epsilon,R})^{-1/2}\Big[I+\tfrac{1}{\sigma^2}(I+\tfrac{n}{\sigma^2}\tilde{\Lambda}_{\epsilon,R})^{-1/2}\tilde{\Lambda}_{\epsilon,R}^{1/2}(\Phi_R^T\Phi_R-nI)\tilde{\Lambda}_{\epsilon,R}^{1/2}(I+\tfrac{n}{\sigma^2}\tilde{\Lambda}_{\epsilon,R})^{-1/2}\Big]^{-1}$$

$$(I+\tfrac{n}{\sigma^2}\tilde{\Lambda}_{\epsilon,R})^{-1/2}\tilde{\Lambda}_{\epsilon,R}^{-1/2}$$

$$=(I+\tfrac{n}{\sigma^2}\tilde{\Lambda}_{\epsilon,R})^{-1}$$

$$+\sum_{j=1}^{\infty}\Big[(-1)^j\tilde{\Lambda}_{\epsilon,R}^{1/2}(I+\tfrac{n}{\sigma^2}\tilde{\Lambda}_{\epsilon,R})^{-1/2}\Big(\tfrac{1}{\sigma^2}(I+\tfrac{n}{\sigma^2}\tilde{\Lambda}_{\epsilon,R})^{-1/2}\tilde{\Lambda}_{\epsilon,R}^{1/2}(\Phi_R^T\Phi_R-nI)\tilde{\Lambda}_{\epsilon,R}^{1/2}(I+\tfrac{n}{\sigma^2}\tilde{\Lambda}_{\epsilon,R})^{-1/2}\Big)^j$$

$$(I+\tfrac{n}{\sigma^2}\tilde{\Lambda}_{\epsilon,R})^{-1/2}\tilde{\Lambda}_{\epsilon,R}^{-1/2}\Big]$$

$$=(I+\tfrac{n}{\sigma^2}\tilde{\Lambda}_{\epsilon,R})^{-1}+\sum_{j=1}^{\infty}(-1)^j\Big(\tfrac{1}{\sigma^2}(I+\tfrac{n}{\sigma^2}\tilde{\Lambda}_{\epsilon,R})^{-1}\tilde{\Lambda}_{\epsilon,R}(\Phi_R^T\Phi_R-nI)\Big)^j(I+\tfrac{n}{\sigma^2}\tilde{\Lambda}_{\epsilon,R})^{-1}.$$

Letting $\epsilon \to 0$, we get

$$(I+\tfrac{1}{\sigma^2}\Lambda_R\Phi_R^T\Phi_R)^{-1}$$

$$=(I+\tfrac{n}{\sigma^2}\Lambda_R)^{-1}+\sum_{j=1}^{\infty}(-1)^j\Big(\frac{1}{\sigma^2}(I+\tfrac{n}{\sigma^2}\Lambda_R)^{-1}\Lambda_R(\Phi_R^T\Phi_R-nI)\Big)^j(I+\tfrac{n}{\sigma^2}\Lambda_R)^{-1}.$$

This concludes the proof. $\qquad\square$

**Lemma 28.** *If* $\|(I+\frac{\Phi_R\Lambda_R\Phi_R^T}{\sigma^2})^{-1}\frac{\Phi_{>R}\Lambda_{>R}\Phi_{>R}^T}{\sigma^2}\|_2 < 1$, *then we have*

$$(I+\tfrac{\Phi\Lambda\Phi^T}{\sigma^2})^{-1}-(I+\tfrac{\Phi_R\Lambda_R\Phi_R^T}{\sigma^2})^{-1}=\sum_{j=1}^{\infty}(-1)^j\Big((I+\tfrac{\Phi_R\Lambda_R\Phi_R^T}{\sigma^2})^{-1}\tfrac{\Phi_{>R}\Lambda_{>R}\Phi_{>R}^T}{\sigma^2}\Big)^j(I+\tfrac{\Phi_R\Lambda_R\Phi_R^T}{\sigma^2})^{-1}.$$

$$(53)$$

*In particular, assume that* $\sigma^2=\Theta(1)$. *Let* $R=n^{\frac{1}{\alpha}+\kappa}$ *where* $0<\kappa<\frac{\alpha-1-2\tau}{\alpha(1+2\tau)}$. *Then with probability of at least* $1-\delta$, *for sufficiently large* $n$, *we have* $\|(I+\frac{\Phi_R\Lambda_R\Phi_R^T}{\sigma^2})^{-1}\frac{\Phi_{>R}\Lambda_{>R}\Phi_{>R}^T}{\sigma^2}\|_2 < 1$ *and* (53) *holds.*

*Proof of Lemma 28.* Define $\Phi_{>R}=(\phi_{R+1}(\mathbf{x}),\phi_{R+2}(\mathbf{x}),...)$, $\Lambda_{>R}=\mathrm{diag}(\lambda_{R+1},\lambda_{R+2},...)$. Then we have

$$(I+\tfrac{\Phi\Lambda\Phi^T}{\sigma^2})^{-1}-(I+\tfrac{\Phi_R\Lambda_R\Phi_R^T}{\sigma^2})^{-1}$$

$$=(I+\tfrac{\Phi_R\Lambda_R\Phi_R^T}{\sigma^2}+\tfrac{\Phi_{>R}\Lambda_{>R}\Phi_{>R}^T}{\sigma^2})^{-1}-(I+\tfrac{\Phi_R\Lambda_R\Phi_R^T}{\sigma^2})^{-1}$$

$$=\Big(\big(I+(I+\tfrac{\Phi_R\Lambda_R\Phi_R^T}{\sigma^2})^{-1}\tfrac{\Phi_{>R}\Lambda_{>R}\Phi_{>R}^T}{\sigma^2}\big)^{-1}-I\Big)(I+\tfrac{\Phi_R\Lambda_R\Phi_R^T}{\sigma^2})^{-1}.$$

By Corollary 26, for sufficiently large $n$, $\|(I+\frac{\Phi_R\Lambda_R\Phi_R^T}{\sigma^2})^{-1}\frac{\Phi_{>R}\Lambda_{>R}\Phi_{>R}^T}{\sigma^2}\|_2 < 1$ with probability of at least $1-\delta$. Hence

$$(I+\tfrac{\Phi\Lambda\Phi^T}{\sigma^2})^{-1}-(I+\tfrac{\Phi_R\Lambda_R\Phi_R^T}{\sigma^2})^{-1}$$

$$=\Big(\big(I+(I+\tfrac{\Phi_R\Lambda_R\Phi_R^T}{\sigma^2})^{-1}\tfrac{\Phi_{>R}\Lambda_{>R}\Phi_{>R}^T}{\sigma^2}\big)^{-1}-I\Big)(I+\tfrac{\Phi_R\Lambda_R\Phi_R^T}{\sigma^2})^{-1}$$

$$=\sum_{j=1}^{\infty}(-1)^j\Big((I+\tfrac{\Phi_R\Lambda_R\Phi_R^T}{\sigma^2})^{-1}\tfrac{\Phi_{>R}\Lambda_{>R}\Phi_{>R}^T}{\sigma^2}\Big)^j(I+\tfrac{\Phi_R\Lambda_R\Phi_R^T}{\sigma^2})^{-1}.$$

$$\square$$

**Lemma 29.** *Assume that $\mu_0 = 0$ and $\sigma^2 = \Theta(n^t)$ where $1 - \frac{\alpha}{1+2\tau} < t < 1$. Let $R = n^{(\frac{1}{\alpha}+\kappa)(1-t)}$ where $0 < \kappa < \frac{\alpha - 1 - 2\tau + (1+2\tau)t}{\alpha^2(1-t)}$. Then when $n$ is sufficiently large, with probability of at least $1 - 2\delta$ we have*

$$\|(I + \tfrac{1}{\sigma^2}\Phi_R\Lambda_R\Phi_R^T)^{-1}f_R(\mathbf{x})\|_2 = \tilde{O}\left(\sqrt{(\tfrac{1}{\delta}+1)n} \cdot n^{\max\{-(1-t), \frac{(1-2\beta)(1-t)}{2\alpha}\}}\right). \tag{54}$$

*Proof of Lemma 29.* Let $\Lambda_{1:R} = \text{diag}\{\lambda_1, \dots, \lambda_R\}$, $\Phi_{1:R} = (\phi_1(\mathbf{x}), \phi_1(\mathbf{x}), \dots, \phi_R(\mathbf{x}))$ and $\boldsymbol{\mu}_{1:R} = (\mu_1, \dots, \mu_R)$. Since $\mu_0 = 0$, we have $(I + \frac{1}{\sigma^2}\Phi_R\Lambda_R\Phi_R^T)^{-1}f_R(\mathbf{x}) = (I + \frac{1}{\sigma^2}\Phi_{1:R}\Lambda_{1:R}\Phi_{1:R}^T)^{-1}\Phi_{1:R}\boldsymbol{\mu}_{1:R}$. Using the Woodbury matrix identity, we have that

$$(I + \tfrac{1}{\sigma^2}\Phi_{1:R}\Lambda_{1:R}\Phi_{1:R}^T)^{-1}\Phi_{1:R}\boldsymbol{\mu}_{1:R} = \left[I - \Phi_{1:R}(\sigma^2 I + \Lambda_{1:R}\Phi_{1:R}^T\Phi_{1:R})^{-1}\Lambda_{1:R}\Phi_{1:R}^T\right]\Phi_{1:R}\boldsymbol{\mu}_{1:R}$$
$$= \Phi_{1:R}\boldsymbol{\mu}_{1:R} - \Phi_{1:R}(\sigma^2 I + \Lambda_{1:R}\Phi_{1:R}^T\Phi_{1:R})^{-1}\Lambda_{1:R}\Phi_{1:R}^T\Phi_{1:R}\boldsymbol{\mu}_{1:R}$$
$$= \Phi_{1:R}(I + \tfrac{1}{\sigma^2}\Lambda_{1:R}\Phi_{1:R}^T\Phi_{1:R})^{-1}\boldsymbol{\mu}_{1:R}. \tag{55}$$

Let $A = (I + \frac{n}{\sigma^2}\Lambda_{1:R})^{-1/2}\Lambda_{1:R}^{1/2}(\Phi_{1:R}^T\Phi_{1:R} - nI)\Lambda_{1:R}^{1/2}(I + \frac{n}{\sigma^2}\Lambda_{1:R})^{-1/2}$. By Corollary 22, with probability of at least $1 - \delta$, we have $\|\frac{1}{\sigma^2}A\|_2 = \sqrt{\log\frac{R}{\delta}}n^{\frac{1-\alpha+2\tau}{2\alpha} - \frac{(1+2\tau)t}{2\alpha}}$. When $n$ is sufficiently large, $\|\frac{1}{\sigma^2}A\|_2 = o(1)$ is less than 1 because $1 - \frac{\alpha}{1+2\tau} < t < 1$. By Lemma 27, we have

$$(I + \tfrac{1}{\sigma^2}\Lambda_{1:R}\Phi_{1:R}^T\Phi_{1:R})^{-1}$$
$$= (I + \tfrac{n}{\sigma^2}\Lambda_{1:R})^{-1} + \sum_{j=1}^{\infty}(-1)^j\left(\tfrac{1}{\sigma^2}(I + \tfrac{n}{\sigma^2}\Lambda_{1:R})^{-1}\Lambda_{1:R}(\Phi_{1:R}^T\Phi_{1:R} - nI)\right)^j(I + \tfrac{n}{\sigma^2}\Lambda_{1:R})^{-1}.$$

We then have

$$\|(I + \tfrac{1}{\sigma^2}\Lambda_{1:R}\Phi_{1:R}^T\Phi_{1:R})^{-1}\boldsymbol{\mu}_{1:R}\|_2$$
$$= \left\|\left((I + \tfrac{n}{\sigma^2}\Lambda_{1:R})^{-1} + \sum_{j=1}^{\infty}(-1)^j\left(\tfrac{1}{\sigma^2}(I + \tfrac{n}{\sigma^2}\Lambda_{1:R})^{-1}\Lambda_{1:R}(\Phi_{1:R}^T\Phi_{1:R} - nI)\right)^j(I + \tfrac{n}{\sigma^2}\Lambda_{1:R})^{-1}\right)\boldsymbol{\mu}_{1:R}\right\|_2$$
$$\leq \left(\|(I + \tfrac{n}{\sigma^2}\Lambda_{1:R})^{-1}\boldsymbol{\mu}_{1:R}\|_2 + \sum_{j=1}^{\infty}\left\|\left(\tfrac{1}{\sigma^2}(I + \tfrac{n}{\sigma^2}\Lambda_{1:R})^{-1}\Lambda_{1:R}(\Phi_{1:R}^T\Phi_{1:R} - nI)\right)^j(I + \tfrac{n}{\sigma^2}\Lambda_{1:R})^{-1}\boldsymbol{\mu}_{1:R}\right\|_2\right). \tag{56}$$

By Lemma 15 and Assumption 5, assuming that $\sup_{i\geq 1}p_{i+1} - p_i = h$, we have

$$\|(I + \tfrac{n}{\sigma^2}\Lambda_{1:R})^{-1}\boldsymbol{\mu}_{1:R}\|_2 \leq \sqrt{\sum_{p=1}^{R}\frac{C_\mu^2 p^{-2\beta}}{(1 + nC_\lambda p^{-\alpha}/\sigma^2)^2}} = \Theta(n^{\max\{-(1-t), \frac{(1-2\beta)(1-t)}{2\alpha}\}}\log^{k/2}n),$$

$$\|(I + \tfrac{n}{\sigma^2}\Lambda_{1:R})^{-1}\boldsymbol{\mu}_{1:R}\|_2 \geq \sqrt{\sum_{i=1}^{\lfloor\frac{R}{h}\rfloor}\frac{C_\mu^2 i^{-2\beta}}{(1 + \frac{n}{\sigma^2}C_\lambda(hi)^{-\alpha})^2}} = \Theta(n^{\max\{-(1-t), \frac{(1-2\beta)(1-t)}{2\alpha}\}}\log^{k/2}n)$$

where $k = \begin{cases} 0, & 2\alpha \neq 2\beta - 1 \\ 1, & 2\alpha = 2\beta - 1 \end{cases}$. Overall we have

$$\|(I + \tfrac{n}{\sigma^2}\Lambda_{1:R})^{-1}\boldsymbol{\mu}_{1:R}\|_2 = \Theta(n^{(1-t)\max\{-1, \frac{1-2\beta}{2\alpha}\}}\log^{k/2}n). \tag{57}$$

Using the fact that $\|\frac{1}{\sigma^2}A\|_2 = \sqrt{\log\frac{R}{\delta}}n^{\frac{1-\alpha+2\tau}{2\alpha} - \frac{(1+2\tau)t}{2\alpha}}$ and $\|(I + \frac{n}{\sigma^2}\Lambda_{1:R})^{-1}\Lambda_{1:R}\|_2 \leq n^{-1}$, we have

$$\left\|\left(\tfrac{1}{\sigma^2}(I + \tfrac{n}{\sigma^2}\Lambda_{1:R})^{-1}\Lambda_{1:R}(\Phi_{1:R}^T\Phi_{1:R} - nI)\right)^j(I + \tfrac{n}{\sigma^2}\Lambda_{1:R})^{-1}\boldsymbol{\mu}_{1:R}\right\|_2$$
$$= \left\|(I + \tfrac{n}{\sigma^2}\Lambda_{1:R})^{-\frac{1}{2}}\Lambda_{1:R}^{\frac{1}{2}}(\tfrac{1}{\sigma^2}A)^j(I + \tfrac{n}{\sigma^2}\Lambda_{1:R})^{-\frac{1}{2}}\Lambda_{1:R}^{\frac{1}{2}}\boldsymbol{\mu}_{1:R}\right\|_2 \tag{58}$$
$$\leq \tilde{O}(n^{-\frac{1-t}{2}})\|\tfrac{1}{\sigma^2}A\|_2^j\|(I + \tfrac{n}{\sigma^2}\Lambda_{1:R})^{-\frac{1}{2}}\Lambda_{1:R}^{-\frac{1}{2}}\boldsymbol{\mu}_{1:R}\|_2$$

By Lemma 16 and the assumption $R = n^{(\frac{1}{\alpha}+\kappa)(1-t)}$,

$$
\begin{aligned}
\|(I+\tfrac{n}{\sigma^2}\Lambda_{1:R})^{-\frac{1}{2}}\Lambda_{1:R}^{-\frac{1}{2}}\boldsymbol{\mu}_{1:R}\|_2 &\le \sqrt{\sum_{p=1}^{R} \frac{(C_\lambda p^{-\alpha})^{-1}C_\mu^2 p^{-2\beta}}{(1+n\underline{C_\lambda}p^{-\alpha}/\sigma^2)^1}} \\
&= \tilde{O}(\max\{n^{-(1-t)/2}, R^{1/2-\beta+\alpha/2}\}) \\
&= \tilde{O}(\max\{n^{-(1-t)/2}, n^{(\frac{1}{2}+\frac{1-2\beta}{2\alpha}+\kappa(1/2-\beta+\alpha/2))(1-t)}\})
\end{aligned}
\tag{59}
$$

We then have

$$
\begin{aligned}
&\left\|\left(\tfrac{1}{\sigma^2}(I+\tfrac{n}{\sigma^2}\Lambda_{1:R})^{-1}\Lambda_{1:R}(\Phi_{1:R}^T\Phi_{1:R}-nI)\right)^j(I+\tfrac{n}{\sigma^2}\Lambda_{1:R})^{-1}\boldsymbol{\mu}_{1:R}\right\|_2 \\
&= \|\tfrac{1}{\sigma^2}A\|_2^j \tilde{O}(\max\{n^{-(1-t)}, n^{(\frac{1-2\beta}{2\alpha}+\kappa(1/2-\beta+\alpha/2))(1-t)}\})
\end{aligned}
\tag{60}
$$

By (56), (57) and (60), we have

$$
\begin{aligned}
&\|(I+\tfrac{1}{\sigma^2}\Lambda_{1:R}\Phi_{1:R}^T\Phi_{1:R})^{-1}\boldsymbol{\mu}_{1:R}\|_2 \\
&= \Theta(n^{(1-t)\max\{-1,\frac{1-2\beta}{2\alpha}\}}\log^{k/2}n) + \sum_{j=1}^{\infty}\|\tfrac{1}{\sigma^2}A\|_2^j \tilde{O}(\max\{n^{-(1-t)}, n^{(1-t)\frac{1-2\beta}{2\alpha}+\kappa(1-t)(1/2-\beta+\alpha/2)}\}) \\
&= \Theta(n^{(1-t)\max\{-1,\frac{1-2\beta}{2\alpha}\}}\log^{k/2}n) + \tilde{O}(n^{\frac{1-\alpha+2\tau}{2\alpha}-\frac{(1+2\tau)t}{2\alpha}})\tilde{O}(\max\{n^{-(1-t)}, n^{(1-t)\frac{1-2\beta}{2\alpha}+\kappa(1-t)(1/2-\beta+\alpha/2)}\}).
\end{aligned}
\tag{61}
$$

By assumption $\kappa < \frac{\alpha-1-2\tau+(1+2\tau)t}{\alpha^2(1-t)}$, we have that

$$
\kappa(1-t)(1/2-\beta+\alpha/2)+\frac{1-\alpha+2\tau}{2\alpha}-\frac{(1+2\tau)t}{2\alpha} < \kappa\alpha(1-t)/2+\frac{1-\alpha+2\tau}{2\alpha}-\frac{(1+2\tau)t}{2\alpha} < 0.
$$

Using (61), we then get

$$
\begin{aligned}
\|(I+\tfrac{1}{\sigma^2}\Lambda_{1:R}\Phi_{1:R}^T\Phi_{1:R})^{-1}\boldsymbol{\mu}_{1:R}\|_2 &= \Theta(n^{(1-t)\max\{-1,\frac{1-2\beta}{2\alpha}\}}\log^{k/2}n) \\
&= \frac{1+o(1)}{\sigma^2}\|(I+\frac{n}{\sigma^2}\Lambda_{1:R})^{-1}\boldsymbol{\mu}_{1:R}\|_2.
\end{aligned}
\tag{62}
$$

By Corollary 20, with probability of at least $1-\delta$, we have

$$
\begin{aligned}
\|\Phi_{1:R}(I+\tfrac{1}{\sigma^2}\Lambda_{1:R}\Phi_{1:R}^T\Phi_{1:R})^{-1}\boldsymbol{\mu}_{1:R}\|_2 &= \tilde{O}(\sqrt{(\tfrac{1}{\delta}+1)n}\|(I+\tfrac{1}{\sigma^2}\Lambda_{1:R}\Phi_{1:R}^T\Phi_{1:R})^{-1}\boldsymbol{\mu}_{1:R}\|_2) \\
&= \tilde{O}(\sqrt{(\tfrac{1}{\delta}+1)n}\cdot n^{(1-t)\max\{-1,\frac{1-2\beta}{2\alpha}\}}).
\end{aligned}
\tag{63}
$$

From (55), we get $\|(I+\tfrac{1}{\sigma^2}\Phi_{1:R}\Lambda_{1:R}\Phi_{1:R}^T)^{-1}\Phi_{1:R}\boldsymbol{\mu}_{1:R}\|_2 = \tilde{O}(\sqrt{(\tfrac{1}{\delta}+1)n}\cdot n^{(1-t)\max\{-1,\frac{1-2\beta}{2\alpha}\}})$. This concludes the proof. $\qquad\square$

**Lemma 30.** *Assume that $\mu_0 > 0$ and $\sigma^2 = \Theta(n^t)$ where $1-\frac{\alpha}{1+2\tau} < t < 1$. Let $R = n^{\frac{1}{\alpha}+\kappa}$ where $0 < \kappa < \frac{\alpha-1-2\tau+(1+2\tau)t}{\alpha^2}$. Then when $n$ is sufficiently large, with probability of at least $1-2\delta$, we have*

$$
\|(I+\tfrac{1}{\sigma^2}\Phi_R\Lambda_R\Phi_R^T)^{-1}f_R(\mathbf{x})\|_2 = \tilde{O}\left(\sqrt{(\tfrac{1}{\delta}+1)n}\right).
\tag{64}
$$

*Proof of Lemma 30.* Using the Woodbury matrix identity, we have that

$$
\begin{aligned}
(I+\tfrac{1}{\sigma^2}\Phi_R\Lambda_R\Phi_R^T)^{-1}f_R(\mathbf{x}) &= \left[I-\Phi_R(\sigma^2 I+\Lambda_R\Phi_R^T\Phi_R)^{-1}\Lambda_R\Phi_R^T\right]\Phi_R\boldsymbol{\mu}_R \\
&= \Phi_R\boldsymbol{\mu}_R - \Phi_R(\sigma^2 I+\Lambda_R\Phi_R^T\Phi_R)^{-1}\Lambda_R\Phi_R^T\Phi_R\boldsymbol{\mu}_R \\
&= \Phi_R(I+\tfrac{1}{\sigma^2}\Lambda_R\Phi_R^T\Phi_R)^{-1}\boldsymbol{\mu}_R.
\end{aligned}
\tag{65}
$$

Let $\boldsymbol{\mu}_{R,1} = (\mu_0,0,...,0)$ and $\boldsymbol{\mu}_{R,2} = (0,\mu_1,...,\mu_R)$. Then $\boldsymbol{\mu}_R = \boldsymbol{\mu}_{R,1}+\boldsymbol{\mu}_{R,2}$. Then we have

$$
\|(I+\tfrac{1}{\sigma^2}\Lambda_R\Phi_R^T\Phi_R)^{-1}\boldsymbol{\mu}_R\|_2 = \|(I+\tfrac{1}{\sigma^2}\Lambda_R\Phi_R^T\Phi_R)^{-1}\boldsymbol{\mu}_{R,1}\|_2 + \|(I+\tfrac{1}{\sigma^2}\Lambda_R\Phi_R^T\Phi_R)^{-1}\boldsymbol{\mu}_{R,2}\|_2.
\tag{66}
$$

According to (62) in the proof of Lemma 29, we have $\|(I + \frac{1}{\sigma^2}\Lambda_R\Phi_R^T\Phi_R)^{-1}\boldsymbol{\mu}_{R,2}\|_2 = \tilde{O}(n^{\max\{-(1-t),\frac{(1-t)(1-2\beta)}{2\alpha}\}})$. Next we estimate $\|(I+\frac{1}{\sigma^2}\Lambda_R\Phi_R^T\Phi_R)^{-1}\boldsymbol{\mu}_{R,1}\|_2$.

Let

$$A=(I+\frac{n}{\sigma^2}\Lambda_{1:R})^{-\gamma/2}\Lambda_{1:R}^{\gamma/2}(\Phi_{1:R}^T\Phi_{1:R}-nI)\Lambda_{1:R}^{\gamma/2}(I+\frac{n}{\sigma^2}\Lambda_{1:R})^{-\gamma/2}$$

where $\frac{1}{1-t}(\frac{1+\alpha+2\tau}{2\alpha}-\frac{(1+2\tau+2\alpha)t}{2\alpha})<\gamma<1$. Since $1-\frac{\alpha}{1+2\tau}<t<1$, $\frac{1}{1-t}(\frac{1+\alpha+2\tau}{2\alpha}-\frac{(1+2\tau+2\alpha)t}{2\alpha})<1$ so the range for $\gamma$ is well-defined. By Corollary 22, with probability of at least $1-\delta$, we have $\|\frac{1}{\sigma^2}A\|_2=\tilde{O}(\sqrt{\log\frac{R}{\delta}}n^{\frac{1+\alpha+2\tau}{2\alpha}-\frac{(1+2\tau+2\alpha)t}{2\alpha}-\gamma(1-t)})=o(1)$. When $n$ is sufficiently large, $\|\frac{1}{\sigma^2}A\|_2$ is less than 1 because $1-\frac{\alpha}{1+2\tau}<t<1$. By Lemma 27, we have

$$(I+\tfrac{1}{\sigma^2}\Lambda_R\Phi_R^T\Phi_R)^{-1}$$
$$=(I+\tfrac{n}{\sigma^2}\Lambda_R)^{-1}+\sum_{j=1}^\infty(-1)^j\big(\tfrac{1}{\sigma^2}(I+\tfrac{n}{\sigma^2}\Lambda_R)^{-1}\Lambda_R(\Phi_R^T\Phi_R-nI)\big)^j(I+\tfrac{n}{\sigma^2}\Lambda_R)^{-1}.$$

We then have

$$\|(I+\tfrac{1}{\sigma^2}\Lambda_R\Phi_R^T\Phi_R)^{-1}\boldsymbol{\mu}_{R,1}\|_2$$
$$=\left\|\left((I+\tfrac{n}{\sigma^2}\Lambda_R)^{-1}+\sum_{j=1}^\infty(-1)^j\big(\tfrac{1}{\sigma^2}(I+\tfrac{n}{\sigma^2}\Lambda_R)^{-1}\Lambda_R(\Phi_R^T\Phi_R-nI)\big)^j(I+\tfrac{n}{\sigma^2}\Lambda_R)^{-1}\right)\boldsymbol{\mu}_{R,1}\right\|_2$$
$$\leq\left(\|(I+\tfrac{n}{\sigma^2}\Lambda_R)^{-1}\boldsymbol{\mu}_{R,1}\|_2+\sum_{j=1}^\infty\left\|\big(\tfrac{1}{\sigma^2}(I+\tfrac{n}{\sigma^2}\Lambda_R)^{-1}\Lambda_R(\Phi_R^T\Phi_R-nI)\big)^j(I+\tfrac{n}{\sigma^2}\Lambda_R)^{-1}\boldsymbol{\mu}_{R,1}\right\|_2\right).$$
$$(67)$$

By Lemma 15,

$$\|(I+\frac{n}{\sigma^2}\Lambda_R)^{-1}\boldsymbol{\mu}_{R,1}\|_2\leq\sqrt{\mu_0^2+\sum_{p=1}^R\frac{C_\mu^2p^{-2\beta}}{(1+n\underline{C_\lambda}p^{-\alpha}/\sigma^2)^2}}=O(1). \tag{68}$$

Let $\tilde{\Lambda}_{1,R}=\text{diag}\{1,\lambda_1,...,\lambda_R\}$ and $I_{0,R}=(0,1,...,1)$. Then $\Lambda_R=\tilde{\Lambda}_{1,R}I_{0,R}$. Let $B=(I+\frac{n}{\sigma^2}\Lambda_R)^{-\gamma/2}\tilde{\Lambda}_{1,R}^{\gamma/2}(\Phi_R^T\Phi_R-nI)\tilde{\Lambda}_{1,R}^{\gamma/2}(I+\frac{n}{\sigma^2}\Lambda_R)^{-\gamma/2}$. According to Corollary 23, we have $\|B\|_2=O(\sqrt{\log\frac{R}{\delta}}n^{\frac{1}{2}})$. Using the fact that $\|\frac{1}{\sigma^2}A\|_2=\tilde{O}(\sqrt{\log\frac{R}{\delta}}n^{\frac{1+\alpha+2\tau}{2\alpha}-\frac{(1+2\tau+2\alpha)t}{2\alpha}-\gamma(1-t)})$, we have

$$\left\|\big(\tfrac{1}{\sigma^2}(I+\tfrac{n}{\sigma^2}\Lambda_R)^{-1}\Lambda_R(\Phi_R^T\Phi_R-nI)\big)^j(I+\tfrac{n}{\sigma^2}\Lambda_R)^{-1}\boldsymbol{\mu}_{R,1}\right\|_2$$
$$=\frac{1}{\sigma^{2j}}\left\|(I+\tfrac{n}{\sigma^2}\Lambda_R)^{-1+\frac{\gamma}{2}}\Lambda_R^{1-\frac{\gamma}{2}}\left(A(I+\tfrac{n}{\sigma^2}\Lambda_R)^{-1+\gamma}\Lambda_R^{1-\gamma}\right)^{j-1}B(I+\tfrac{n}{\sigma^2}\Lambda_R)^{-1+\frac{\gamma}{2}}\boldsymbol{\mu}_{R,1}\right\|_2$$
$$\leq\frac{1}{\sigma^2}(n^{(-1+\frac{\gamma}{2}+(-1+\gamma)(j-1))(1-t)}\tilde{O}(\sqrt{\log\tfrac{R}{\delta}}n^{(j-1)(\frac{1+\alpha+2\tau}{2\alpha}-\frac{(1+2\tau+2\alpha)t}{2\alpha}-\gamma(1-t))})\sqrt{\log\tfrac{R}{\delta}}n^{\frac{1}{2}}\|\boldsymbol{\mu}_{R,1}\|_2$$
$$\leq n^{(-1+\frac{\gamma}{2})(1-t)+\frac{1}{2}-t}\tilde{O}(n^{\frac{[1-\alpha+2\tau-(1+2\tau)t](j-1)}{2\alpha}})\sqrt{\log\tfrac{R}{\delta}}\|\boldsymbol{\mu}_{R,1}\|_2$$
$$=\tilde{O}(n^{-\frac{1}{2}+\frac{\gamma}{2}(1-t)+\frac{[1-\alpha+2\tau-(1+2\tau)t](j-1)}{2\alpha}}).$$
$$(69)$$

Since $\frac{1}{1-t}(\frac{1+\alpha+2\tau}{2\alpha}-\frac{(1+2\tau+2\alpha)t}{2\alpha})<\gamma<1$ and $-\frac{1}{2}+\frac{1}{1-t}(\frac{1+\alpha+2\tau}{2\alpha}-\frac{(1+2\tau+2\alpha)t}{2\alpha})\frac{1-t}{2}<0$, we can let $\gamma$ be a little bit larger than $\frac{1}{1-t}(\frac{1+\alpha+2\tau}{2\alpha}-\frac{(1+2\tau+2\alpha)t}{2\alpha})$ and make $-\frac{1}{2}+\frac{\gamma}{2}(1-t)<0$ holds. By (67), (68), (69), we have

$$\|(I+\tfrac{1}{\sigma^2}\Lambda_R\Phi_R^T\Phi_R)^{-1}\boldsymbol{\mu}_{R,1}\|_2$$
$$\leq O(1)+\sum_{j=1}^\infty\tilde{O}(n^{-\frac{1}{2}+\frac{\gamma}{2}(1-t)+\frac{[1-\alpha+2\tau-(1+2\tau)t](j-1)}{2\alpha}})$$
$$\leq O(1)+o(1)=O(1).$$
$$(70)$$

According to (66), we have $\|(I + \frac{1}{\sigma^2}\Lambda_R\Phi_R^T\Phi_R)^{-1}\boldsymbol{\mu}_R\|_2 = \tilde{O}(n^{\max\{-(1-t), \frac{(1-t)(1-2\beta)}{2\alpha}\}}) + O(1) = O(1)$. By Corollary 20, with probability of at least $1 - \delta$, we have

$$\|\Phi_R(I + \tfrac{1}{\sigma^2}\Lambda_R\Phi_R^T\Phi_R)^{-1}\boldsymbol{\mu}_R\|_2 = \tilde{O}(\sqrt{(\tfrac{1}{\delta}+1)n}\|(I + \tfrac{1}{\sigma^2}\Lambda_R\Phi_R^T\Phi_R)^{-1}\boldsymbol{\mu}_R\|_2)$$

$$= \tilde{O}\left(\sqrt{(\tfrac{1}{\delta}+1)n}\right).$$

From (65), we get $\|(I + \frac{1}{\sigma^2}\Phi_R\Lambda_R\Phi_R^T)^{-1}f_R(\mathbf{x})\|_2 = \tilde{O}\left(\sqrt{(\frac{1}{\delta}+1)n}\right)$. This concludes the proof. $\square$

**Lemma 31.** *Assume that $\sigma^2 = \Theta(1)$. Let $R = n^{\frac{1}{\alpha}+\kappa}$ where $0 < \kappa < \frac{\alpha-1-2\tau}{\alpha^2}$. Assume that $\mu_0 = 0$. Then when $n$ is sufficiently large, with probability of at least $1 - 3\delta$ we have*

$$\|(I + \tfrac{\Phi\Lambda\Phi^T}{\sigma^2})^{-1}f_R(\mathbf{x})\|_2 = \tilde{O}(\sqrt{(\tfrac{1}{\delta}+1)n}\cdot n^{\max\{-1, \frac{1-2\beta}{2\alpha}\}}). \tag{71}$$

*Assume that $\mu_0 > 0$. Then when $n$ is sufficiently large, with probability of at least $1 - 3\delta$ we have*

$$\|(I + \tfrac{\Phi\Lambda\Phi^T}{\sigma^2})^{-1}f_R(\mathbf{x})\|_2 = \tilde{O}(\sqrt{(\tfrac{1}{\delta}+1)n}). \tag{72}$$

*Proof of Lemma 31.* We have

$$(I + \tfrac{\Phi\Lambda\Phi^T}{\sigma^2})^{-1}f_R(\mathbf{x})$$
$$= (I + \tfrac{\Phi_R\Lambda_R\Phi_R^T}{\sigma^2})^{-1}f_R(\mathbf{x}) + \left((I + \tfrac{\Phi\Lambda\Phi^T}{\sigma^2})^{-1} - (I + \tfrac{\Phi_R\Lambda_R\Phi_R^T}{\sigma^2})^{-1}\right)f_R(\mathbf{x}). \tag{73}$$

When $\mu_0 = 0$, by Lemma 29, with probability of at least $1 - 2\delta$, we have

$$\|(I + \tfrac{1}{\sigma^2}\Phi_R\Lambda_R\Phi_R^T)^{-1}f_R(\mathbf{x})\|_2 = \tilde{O}(\sqrt{(\tfrac{1}{\delta}+1)n}\cdot n^{\max\{-1, \frac{1-2\beta}{2\alpha}\}}).$$

Since $\frac{\alpha-1-2\tau}{\alpha^2} < \frac{\alpha-1-2\tau}{\alpha(1+2\tau)}$, we apply Lemma 28 and Corollary 26 and get that with probability of at least $1 - \delta$, the second term in the right hand side of (73) is estimated as follows:

$$\|\left((I + \tfrac{\Phi\Lambda\Phi^T}{\sigma^2})^{-1} - (I + \tfrac{\Phi_R\Lambda_R\Phi_R^T}{\sigma^2})^{-1}\right)f_R(\mathbf{x})\|_2$$

$$= \|\sum_{j=1}^{\infty}(-1)^j\left((I + \tfrac{\Phi_R\Lambda_R\Phi_R^T}{\sigma^2})^{-1}\tfrac{\Phi_{>R}\Lambda_{>R}\Phi_{>R}^T}{\sigma^2}\right)^j(I + \tfrac{\Phi_R\Lambda_R\Phi_R^T}{\sigma^2})^{-1}f_R(\mathbf{x})\|_2$$

$$= \sum_{j=1}^{\infty}\left\|\left((I + \tfrac{\Phi_R\Lambda_R\Phi_R^T}{\sigma^2})^{-1}\tfrac{\Phi_{>R}\Lambda_{>R}\Phi_{>R}^T}{\sigma^2}\right)\right\|_2^j\|(I + \tfrac{\Phi_R\Lambda_R\Phi_R^T}{\sigma^2})^{-1}f_R(\mathbf{x})\|_2$$

$$= \sum_{j=1}^{\infty}\tilde{O}(n^{-j\kappa\alpha})\tilde{O}(\sqrt{(\tfrac{1}{\delta}+1)n}\cdot n^{\max\{-1, \frac{1-2\beta}{2\alpha}\}})$$

$$= o(\sqrt{(\tfrac{1}{\delta}+1)n}\cdot n^{\max\{-1, \frac{1-2\beta}{2\alpha}\}}).$$

Overall, from (73), we have that with probability $1 - 3\delta$,

$$\|(I + \tfrac{\Phi\Lambda\Phi^T}{\sigma^2})^{-1}f_R(\mathbf{x})\|_2 = \tilde{O}(\sqrt{(\tfrac{1}{\delta}+1)n}\cdot n^{\max\{-1, \frac{1-2\beta}{2\alpha}\}}).$$

When $\mu_0 > 0$, using the same approach and Lemma 30, we can prove that $\|(I + \frac{\Phi\Lambda\Phi^T}{\sigma^2})^{-1}f_R(\mathbf{x})\|_2 = \tilde{O}(\sqrt{(\frac{1}{\delta}+1)n})$. This concludes the proof. $\square$

# D    PROOF OF THE MAIN RESULTS

## D.1    PROOFS RELATED TO THE ASYMPTOTICS OF THE NORMALIZED STOCHASTIC COMPLEXITY

**Lemma 32.** *Under Assumptions 4, 5 and 6, with probability of at least $1 - 2\delta$ we have, we have*

$$|T_{1,R}(D_n) - T_1(D_n)| = \tilde{O}\left(\tfrac{1}{\sigma^2}(nR^{1-\alpha} + n^{1/2}R^{1-\alpha+\tau} + R^{1-\alpha+2\tau})\right) \tag{74}$$

If $R = n^{\frac{1}{\alpha}+\kappa}$ where $\kappa > 0$, we have $|T_{1,R}(D_n) - T_1(D_n)| = o\left(\frac{1}{\sigma^2} n^{\frac{1}{\alpha}}\right)$. If we further assume that $0 < \kappa < \frac{\alpha-1-2\tau}{\alpha^2}$, $\mu_0 = 0$ and $\sigma^2 = \Theta(1)$, then for sufficiently large $n$ with probability of at least $1-4\delta$ we have

$$|T_{2,R}(D_n) - T_2(D_n)| = \tilde{O}\left((\tfrac{1}{\delta}+1)n^{\max\{(\frac{1}{\alpha}+\kappa)\frac{1-2\beta}{2}, 1+\frac{1-2\beta}{\alpha}+\frac{(1-2\beta)\kappa}{2}, -1-\kappa\alpha, 1+\frac{1-2\beta}{\alpha}-\kappa\alpha\}}\right). \quad (75)$$

*Proof of Lemma 32.* Define $\Phi_{>R} = (\phi_{R+1}(\mathbf{x}), \phi_{R+2}(\mathbf{x}), \dots, \phi_p(\mathbf{x}), \dots)$, and $\Lambda_{>R} = \text{diag}(\lambda_{R+1}, \dots, \lambda_p, \dots)$. We then have

$$|T_1(D_n) - T_{1,R}(D_n)| = \left|\frac{1}{2}\text{logdet}(I+\frac{1}{\sigma^2}\Phi\Lambda\Phi^T) - \frac{1}{2}\text{logdet}(I+\frac{1}{\sigma^2}\Phi_R\Lambda_R\Phi_R^T)\right|$$
$$+ \frac{1}{2}\left|\text{Tr}(I+\frac{\Phi\Lambda\Phi^T}{\sigma^2})^{-1} - \text{Tr}(I+\frac{\Phi_R\Lambda_R\Phi_R^T}{\sigma^2})^{-1}\right|. \quad (76)$$

As for the first term in the right hand side of (76), we have

$$\left|\frac{1}{2}\text{logdet}(I+\frac{1}{\sigma^2}\Phi\Lambda\Phi^T) - \frac{1}{2}\text{logdet}(I+\frac{1}{\sigma^2}\Phi_R\Lambda_R\Phi_R^T)\right|$$
$$= \left|\frac{1}{2}\text{logdet}\left((I+\frac{1}{\sigma^2}\Phi_R\Lambda_R\Phi_R^T)^{-1}(I+\frac{1}{\sigma^2}\Phi_R\Lambda_R\Phi_R^T+\frac{1}{\sigma^2}\Phi_{>R}\Lambda_{>R}\Phi_{>R}^T)\right)\right|$$
$$= \left|\frac{1}{2}\text{logdet}\left(I+\frac{1}{\sigma^2}(I+\frac{1}{\sigma^2}\Phi_R\Lambda_R\Phi_R^T)^{-1}\Phi_{>R}\Lambda_{>R}\Phi_{>R}^T\right)\right| \quad (77)$$
$$= \frac{1}{2}\left|\text{Trlog}\left(I+\frac{1}{\sigma^2}(I+\frac{1}{\sigma^2}\Phi_R\Lambda_R\Phi_R^T)^{-1}\Phi_{>R}\Lambda_{>R}\Phi_{>R}^T\right)\right|.$$

Given a concave function $h$ and a matrix $B \in \mathbb{R}^{n\times n}$ whose eigenvalues $\zeta_1, \dots, \zeta_n$ are all positive, we have that

$$\text{Tr}h(B) = \sum_{p=1}^n h(\zeta_i) \leq nh(\tfrac{1}{n}\sum_{p=1}^n \zeta_i) \leq nh(\tfrac{1}{n}\text{Tr}B), \quad (78)$$

where we used Jensen's inequality. Using $h(x) = \log(1+x)$ in (78), with probability $1-\delta$, we have

$$\left|\tfrac{1}{2}\text{logdet}(I+\tfrac{1}{\sigma^2}\Phi\Lambda\Phi^T) - \tfrac{1}{2}\text{logdet}(I+\tfrac{1}{\sigma^2}\Phi_R\Lambda_R\Phi_R^T)\right|$$
$$\leq \tfrac{n}{2}\log(1+\tfrac{1}{n}\text{Tr}(\tfrac{1}{\sigma^2}(I+\tfrac{\Phi_R\Lambda_R\Phi_R^T}{\sigma^2})^{-1}\Phi_{>R}\Lambda_{>R}\Phi_{>R}^T))$$
$$\leq \tfrac{n}{2}\log(1+\tfrac{1}{n\sigma^2}\|(I+\tfrac{\Phi_R\Lambda_R\Phi_R^T}{\sigma^2})^{-1}\|_2\text{Tr}(\Phi_{>R}\Lambda_{>R}\Phi_{>R}^T))$$
$$\leq \tfrac{n}{2}\log(1+\tfrac{1}{n\sigma^2}\sum_{p=R+1}^\infty \lambda_p\|\phi_p(\mathbf{x})\|_2^2) \leq \tfrac{1}{2\sigma^2}\sum_{p=R+1}^\infty \lambda_p\|\phi_p(\mathbf{x})\|_2^2 \quad (79)$$
$$= \tfrac{1}{2\sigma^2}\sum_{p=R+1}^\infty \lambda_p\left(C_\phi^2\tilde{O}\left(\sqrt{p^{2\tau}n\|\phi_p\|_2^2}+p^{2\tau}\right)+n\|\phi_p\|_2^2\right)$$
$$= \tilde{O}(\tfrac{1}{\sigma^2}n\sum_{p=R+1}^\infty \lambda_p+n^{1/2}\sum_{p=R+1}^\infty \lambda_p p^\tau+\sum_{p=R+1}^\infty \lambda_p p^{2\tau})$$
$$= \tilde{O}\left(\tfrac{1}{\sigma^2}(nR^{1-\alpha}+n^{1/2}R^{1-\alpha+\tau}+R^{1-\alpha+2\tau})\right) = o\left(\tfrac{1}{\sigma^2}n^{\frac{1}{\alpha}}\right),$$

where in the second inequality we use the fact that $\text{Tr}AB \leq \|A\|_2\text{Tr}B$ when $A$ and $B$ are symmetric positive definite matrices, and in the last inequality we use Lemma 18.

As for the second term in the right hand side of (76), let $A = (I+\frac{\Phi_R\Lambda_R\Phi_R^T}{\sigma^2})^{-1/2}$. Then we have

$$\tfrac{1}{2}\left|\text{Tr}(I+\tfrac{\Phi\Lambda\Phi^T}{\sigma^2})^{-1} - \text{Tr}(I+\tfrac{\Phi_R\Lambda_R\Phi_R^T}{\sigma^2})^{-1}\right|$$
$$= \tfrac{1}{2}\left|\text{Tr}A\left[I-(I+A(\tfrac{\Phi_{>R}\Lambda_{>R}\Phi_{>R}^T}{\sigma^2})A)^{-1}\right]A\right|$$
$$\leq \tfrac{1}{2}\text{Tr}\left[I-(I+A(\tfrac{\Phi_{>R}\Lambda_{>R}\Phi_{>R}^T}{\sigma^2})A)^{-1}\right]$$
$$\leq \tfrac{n}{2}(1-(1+\tfrac{1}{n}\text{Tr}A(\tfrac{\Phi_{>R}\Lambda_{>R}\Phi_{>R}^T}{\sigma^2})A)^{-1}) \leq \tfrac{n}{2}(1-(1+\tfrac{1}{n}\text{Tr}(\tfrac{\Phi_{>R}\Lambda_{>R}\Phi_{>R}^T}{\sigma^2}))^{-1})$$
$$\leq \tfrac{n}{2}(1-(1+\tfrac{1}{n\sigma^2}\sum_{p=R+1}^\infty \lambda_p\|\phi_p(\mathbf{x})\|_2^2)^{-1}) \leq \tfrac{1}{2\sigma^2}\sum_{p=R+1}^\infty \lambda_p\|\phi_p(\mathbf{x})\|_2^2$$
$$= \tilde{O}\left(\tfrac{1}{\sigma^2}(nR^{1-\alpha}+n^{1/2}R^{1-\alpha+\tau}+R^{1-\alpha+2\tau})\right) = o\left(\tfrac{1}{\sigma^2}n^{\frac{1}{\alpha}}\right),$$

where in the first inequality we use the fact that $\|A\|_2 < 1$ and $\operatorname{Tr} ABA \le \|A\|_2^2 \operatorname{Tr} B$ when $A$ and $B$ are symmetric positive definite matrices, in the second inequality we use $h(x) = 1 - 1/(1+x)$ in (78) and in the last equality we use the last few steps of (79). This concludes the proof of the first statement.

As for $|T_2(D_n) - T_{2,R}(D_n)|$, we have

$$
\begin{aligned}
|T_2(D_n) - T_{2,R}(D_n)| = & \left| f(\mathbf{x})^T (I + \tfrac{\Phi \Lambda \Phi^T}{\sigma^2})^{-1} f(\mathbf{x}) - f_R(\mathbf{x})^T (I + \tfrac{\Phi \Lambda \Phi^T}{\sigma^2})^{-1} f_R(\mathbf{x}) \right| \\
& + \left| f_R(\mathbf{x})^T (I + \tfrac{\Phi \Lambda \Phi^T}{\sigma^2})^{-1} f_R(\mathbf{x}) - f_R(\mathbf{x})^T (I + \tfrac{\Phi_R \Lambda_R \Phi_R^T}{\sigma^2})^{-1} f_R(\mathbf{x}) \right|.
\end{aligned}
\tag{80}
$$

For the first term on the right-hand side of (80), we have

$$
\begin{aligned}
& \left| f(\mathbf{x})^T (I + \tfrac{\Phi \Lambda \Phi^T}{\sigma^2})^{-1} f(\mathbf{x}) - f_R(\mathbf{x})^T (I + \tfrac{\Phi \Lambda \Phi^T}{\sigma^2})^{-1} f_R(\mathbf{x}) \right| \\
& \le 2 \left| f_{>R}(\mathbf{x})^T (I + \tfrac{\Phi \Lambda \Phi^T}{\sigma^2})^{-1} f_R(\mathbf{x}) \right| + \left| f_{>R}(\mathbf{x})^T (I + \tfrac{\Phi \Lambda \Phi^T}{\sigma^2})^{-1} f_{>R}(\mathbf{x}) \right| \\
& \le 2 \| f_{>R}(\mathbf{x}) \|_2 \| (I + \tfrac{\Phi \Lambda \Phi^T}{\sigma^2})^{-1} f_R(\mathbf{x}) \|_2 + \| f_{>R}(\mathbf{x}) \|_2 \| (I + \tfrac{\Phi \Lambda \Phi^T}{\sigma^2})^{-1} \|_2 \| f_{>R}(\mathbf{x}) \|_2 \\
& \le 2 \| f_{>R}(\mathbf{x}) \|_2 \| (I + \tfrac{\Phi \Lambda \Phi^T}{\sigma^2})^{-1} f_R(\mathbf{x}) \|_2 + \| f_{>R}(\mathbf{x}) \|_2^2 .
\end{aligned}
$$

Applying Corollary 19 and Lemma 31, with probability of at least $1 - 4\delta$, we have

$$
\begin{aligned}
& \left| f(\mathbf{x})^T (I + \tfrac{\Phi \Lambda \Phi^T}{\sigma^2})^{-1} f(\mathbf{x}) - f_R(\mathbf{x})^T (I + \tfrac{\Phi \Lambda \Phi^T}{\sigma^2})^{-1} f_R(\mathbf{x}) \right| \\
& \le 2 \tilde{O} \left( \sqrt{(\tfrac{1}{\delta}+1) n R^{1-2\beta}} \right) \tilde{O}(\sqrt{(\tfrac{1}{\delta}+1)n} \cdot n^{\max\{-1, \frac{1-2\beta}{2\alpha}\}}) + \tilde{O}((\tfrac{1}{\delta}+1) n R^{1-2\beta}) \\
& = 2 \tilde{O} \left( (\tfrac{1}{\delta}+1) n^{1 + (\frac{1}{\alpha}+\kappa) \frac{1-2\beta}{2} + \max\{-1, \frac{1-2\beta}{2\alpha}\}} \right) + \tilde{O}((\tfrac{1}{\delta}+1) n^{1 + (\frac{1}{\alpha}+\kappa)(1-2\beta)}) \\
& = 2 \tilde{O} \left( (\tfrac{1}{\delta}+1) n^{1 + (\frac{1}{\alpha}+\kappa) \frac{1-2\beta}{2} + \max\{-1, \frac{1-2\beta}{2\alpha}\}} \right),
\end{aligned}
$$

where the last equality holds because $(\frac{1}{\alpha}+\kappa) \frac{1-2\beta}{2} < \frac{1-2\beta}{2\alpha}$ when $\kappa > 0$.

As for the second term on the right-hand side of (80), according to Lemma 28, Corollary 26 and Lemma 29, we have

$$
\begin{aligned}
& \left| f_R(\mathbf{x})^T (I + \tfrac{\Phi \Lambda \Phi^T}{\sigma^2})^{-1} f_R(\mathbf{x}) - f_R(\mathbf{x})^T (I + \tfrac{\Phi_R \Lambda_R \Phi_R^T}{\sigma^2})^{-1} f_R(\mathbf{x}) \right| \\
& = \left| \sum_{j=1}^{\infty} (-1)^j f_R(\mathbf{x})^T \left( (I + \tfrac{\Phi_R \Lambda_R \Phi_R^T}{\sigma^2})^{-1} \tfrac{\Phi_{>R} \Lambda_{>R} \Phi_{>R}^T}{\sigma^2} \right)^j (I + \tfrac{\Phi_R \Lambda_R \Phi_R^T}{\sigma^2})^{-1} f_R(\mathbf{x}) \right| \\
& \le \sum_{j=1}^{\infty} \| (I + \tfrac{\Phi_R \Lambda_R \Phi_R^T}{\sigma^2})^{-1} \|_2^{j-1} \cdot \| \tfrac{\Phi_{>R} \Lambda_{>R} \Phi_{>R}^T}{\sigma^2} \|_2^j \cdot \| (I + \tfrac{\Phi_R \Lambda_R \Phi_R^T}{\sigma^2})^{-1} f_R(\mathbf{x}) \|_2^2 \\
& = \sum_{j=1}^{\infty} \tilde{O}(n^{-j\kappa\alpha}) \tilde{O}((\tfrac{1}{\delta}+1) n^{1 + \max\{-2, \frac{1-2\beta}{\alpha}\}}) \\
& = \tilde{O}((\tfrac{1}{\delta}+1) n^{1 + \max\{-2, \frac{1-2\beta}{\alpha}\} - \kappa\alpha}).
\end{aligned}
\tag{81}
$$

By (80), we have

$$
\begin{aligned}
|T_2(D_n) - T_{2,R}(D_n)| = & \tilde{O} \left( (\tfrac{1}{\delta}+1) n^{1 + (\frac{1}{\alpha}+\kappa) \frac{1-2\beta}{2} + \max\{-1, \frac{1-2\beta}{2\alpha}\}} \right) + \tilde{O} \left( (\tfrac{1}{\delta}+1) n^{1 + \max\{-2, \frac{1-2\beta}{\alpha}\} - \kappa\alpha} \right) \\
= & \tilde{O} \left( (\tfrac{1}{\delta}+1) n^{\max\{(\frac{1}{\alpha}+\kappa) \frac{1-2\beta}{2}, 1 + \frac{1-2\beta}{\alpha} + \frac{(1-2\beta)\kappa}{2}, -1-\kappa\alpha, 1 + \frac{1-2\beta}{\alpha} - \kappa\alpha\}} \right).
\end{aligned}
$$

This concludes the proof of the second statement. $\qquad \square$

In Lemma 32, we gave a bound for $|T_{2,R}(D_n) - T_2(D_n)|$ when $n^{\frac{1}{\alpha}} < R < n^{\frac{1}{\alpha} + \frac{\alpha-1-2\tau}{\alpha^2}}$. For $R > n$, we note the following lemma:

**Lemma 33.** *Let $R = n^C$ and $\sigma^2 = n^t$. Assume that $C \geq 1$ and $C(1 - \alpha + 2\tau) - t < 0$. Under Assumptions 4, 5 and 6, for sufficiently large $n$ and with probability of at least $1 - 3\delta$ we have*

$$|T_{2,R}(D_n) - T_2(D_n)| = \tilde{O}\left((\tfrac{1}{\delta} + 1)\tfrac{1}{\sigma^2}nR^{\max\{1/2 - \beta, 1 - \alpha + 2\tau\}}\right). \tag{82}$$

*Proof of Lemma 33.* Define $\Phi_{>R} = (\phi_{R+1}(\mathbf{x}), \phi_{R+2}(\mathbf{x}), \ldots, \phi_p(\mathbf{x}), \ldots)$, and $\Lambda_{>R} = \mathrm{diag}(\lambda_{R+1}, \ldots, \lambda_p, \ldots)$. Then we have

$$
\begin{aligned}
|T_2(D_n) - T_{2,R}(D_n)| = &\left| f(\mathbf{x})^T(I + \frac{\Phi\Lambda\Phi^T}{\sigma^2})^{-1}f(\mathbf{x}) - f_R(\mathbf{x})^T(I + \frac{\Phi\Lambda\Phi^T}{\sigma^2})^{-1}f_R(\mathbf{x}) \right| \\
&+ \left| f_R(\mathbf{x})^T(I + \frac{\Phi\Lambda\Phi^T}{\sigma^2})^{-1}f_R(\mathbf{x}) - f_R(\mathbf{x})^T(I + \frac{\Phi_R\Lambda_R\Phi_R^T}{\sigma^2})^{-1}f_R(\mathbf{x}) \right|.
\end{aligned}
\tag{83}
$$

For the first term on the right-hand side of (83), with probability $1 - 3\delta$ we have

$$
\begin{aligned}
&\left| f(\mathbf{x})^T(I + \frac{\Phi\Lambda\Phi^T}{\sigma^2})^{-1}f(\mathbf{x}) - f_R(\mathbf{x})^T(I + \frac{\Phi\Lambda\Phi^T}{\sigma^2})^{-1}f_R(\mathbf{x}) \right| \\
\leq &2\left| f_{>R}(\mathbf{x})^T(I + \frac{\Phi\Lambda\Phi^T}{\sigma^2})^{-1}f_R(\mathbf{x}) \right| + \left| f_{>R}(\mathbf{x})^T(I + \frac{\Phi\Lambda\Phi^T}{\sigma^2})^{-1}f_{>R}(\mathbf{x}) \right| \\
\leq &2\|f_{>R}(\mathbf{x})\|_2\|(I + \frac{\Phi\Lambda\Phi^T}{\sigma^2})^{-1}\|_2\|f_R(\mathbf{x})\|_2 + \|f_{>R}(\mathbf{x})\|_2\|(I + \frac{\Phi\Lambda\Phi^T}{\sigma^2})^{-1}\|_2\|f_{>R}(\mathbf{x})\|_2 \\
\leq &2\|f_{>R}(\mathbf{x})\|_2\|f_R(\mathbf{x})\|_2 + \|f_{>R}(\mathbf{x})\|_2^2 \\
\leq &2\tilde{O}\left(\sqrt{(\tfrac{1}{\delta} + 1)nR^{1-2\beta}}\right)\tilde{O}(\sqrt{(\tfrac{1}{\delta} + 1)n} \cdot \|f\|_2) + \tilde{O}((\tfrac{1}{\delta} + 1)nR^{1-2\beta}) \\
= &\tilde{O}\left((\tfrac{1}{\delta} + 1)nR^{1/2 - \beta}\right),
\end{aligned}
$$

where we used Corollary 19 and Lemma 17 for the last inequality.

The assumption $C(1 - \alpha + 2\tau) - t < 0$ means that $\frac{R^{1 - \alpha + 2\tau}}{\sigma^2} = o(1)$. For the second term on the right-hand side of (83), by Lemmas 28 and 25, we have

$$
\begin{aligned}
&\left| f_R(\mathbf{x})^T(I + \frac{\Phi\Lambda\Phi^T}{\sigma^2})^{-1}f_R(\mathbf{x}) - f_R(\mathbf{x})^T(I + \frac{\Phi_R\Lambda_R\Phi_R^T}{\sigma^2})^{-1}f_R(\mathbf{x}) \right| \\
= &\left| \sum_{j=1}^{\infty}(-1)^j f_R(\mathbf{x})^T\left((I + \frac{\Phi_R\Lambda_R\Phi_R^T}{\sigma^2})^{-1}\frac{\Phi_{>R}\Lambda_{>R}\Phi_{>R}^T}{\sigma^2}\right)^j(I + \frac{\Phi_R\Lambda_R\Phi_R^T}{\sigma^2})^{-1}f_R(\mathbf{x}) \right| \\
\leq &\sum_{j=1}^{\infty}\|(I + \frac{\Phi_R\Lambda_R\Phi_R^T}{\sigma^2})^{-1}\|_2^{j+1} \cdot \|\frac{\Phi_{>R}\Lambda_{>R}\Phi_{>R}^T}{\sigma^2}\|_2^j \cdot \|f_R(\mathbf{x})\|_2^2 \\
= &\sum_{j=1}^{\infty}\tilde{O}(\frac{1}{\sigma^2}R^{j(1-\alpha+2\tau)})\tilde{O}((\tfrac{1}{\delta} + 1)n\|f\|_2^2) \\
= &\tilde{O}((\tfrac{1}{\delta} + 1)\frac{1}{\sigma^2}nR^{1-\alpha+2\tau}).
\end{aligned}
\tag{84}
$$

Using (83), we have

$$
\begin{aligned}
|T_2(D_n) - T_{2,R}(D_n)| &= \tilde{O}\left((\tfrac{1}{\delta} + 1)nR^{1/2 - \beta}\right) + \tilde{O}((\tfrac{1}{\delta} + 1)n\frac{1}{\sigma^2}R^{1-\alpha+2\tau}) \\
&= \tilde{O}\left((\tfrac{1}{\delta} + 1)n\frac{1}{\sigma^2}R^{\max\{1/2 - \beta, 1 - \alpha + 2\tau\}}\right).
\end{aligned}
$$

$\square$

Next we consider the asympototics of $T_{1,R}(D_n)$ and $T_{2,R}(D_n)$.

**Lemma 34.** *Let $A = (I + \frac{n}{\sigma^2}\Lambda_R)^{-\gamma/2}\Lambda_R^{\gamma/2}(\Phi_R^T\Phi_R - nI)\Lambda_R^{\gamma/2}(I + \frac{n}{\sigma^2}\Lambda_R)^{-\gamma/2}$. Assume that $\|A\|_2 < 1$ where $\frac{1+2\tau}{\alpha} < \gamma \le 1$. Then we have*

$$T_{2,R}(D_n) = \frac{n}{2\sigma^2}\boldsymbol{\mu}_R^T(I + \frac{n}{\sigma^2}\Lambda_R)^{-1}\boldsymbol{\mu}_R + \frac{1}{2}\sum_{j=1}^{\infty}(-1)^{j+1}E_j,$$

*where*

$$E_j = \boldsymbol{\mu}_R^T\frac{1}{\sigma^2}(I + \frac{n}{\sigma^2}\Lambda_R)^{-1}(\Phi_R^T\Phi_R - nI)\left(\frac{1}{\sigma^2}(I + \frac{n}{\sigma^2}\Lambda_R)^{-1}\Lambda_R(\Phi_R^T\Phi_R - nI)\right)^{j-1}(I + \frac{n}{\sigma^2}\Lambda_R)^{-1}\boldsymbol{\mu}_R.$$

*Proof of Lemma 34.* Let $\tilde{\Lambda}_{\epsilon,R} = \text{diag}\{\epsilon, \lambda_1, ..., \lambda_R\}$. Since $\Lambda_R = \text{diag}\{0, \lambda_1, ..., \lambda_R\}$, we have that when $\epsilon$ is sufficiently small, $\|\frac{1}{\sigma^2}(I + \frac{n}{\sigma^2}\tilde{\Lambda}_{\epsilon,R})^{-1/2}\tilde{\Lambda}_{\epsilon,R}^{1/2}(\Phi_R^T\Phi_R - nI)\tilde{\Lambda}_{\epsilon,R}^{1/2}(I + \frac{n}{\sigma^2}\tilde{\Lambda}_{\epsilon,R})^{-1/2}\|_2 < 1$. Since all diagonal entries of $\tilde{\Lambda}_{\epsilon,R}$ are positive, we have

$$\begin{aligned}
&\frac{1}{2\sigma^2}\boldsymbol{\mu}_R^T\Phi_R^T(I + \frac{1}{\sigma^2}\Phi_R\tilde{\Lambda}_{\epsilon,R}\Phi_R^T)^{-1}\Phi_R\boldsymbol{\mu}_R \\
&= \frac{1}{2\sigma^2}\boldsymbol{\mu}_R^T\Phi_R^T\left[I - \Phi_R(\sigma^2 I + \tilde{\Lambda}_{\epsilon,R}\Phi_R^T\Phi_R)^{-1}\tilde{\Lambda}_{\epsilon,R}\Phi_R^T\right]\Phi_R\boldsymbol{\mu}_R \\
&= \frac{1}{2\sigma^2}\boldsymbol{\mu}_R^T\Phi_R^T\Phi_R\boldsymbol{\mu}_R - \frac{1}{2\sigma^2}\boldsymbol{\mu}_R^T\Phi_R^T\Phi_R(\sigma^2 I + \tilde{\Lambda}_{\epsilon,R}\Phi_R^T\Phi_R)^{-1}\tilde{\Lambda}_{\epsilon,R}\Phi_R^T\Phi_R\boldsymbol{\mu}_R \\
&= \frac{1}{2}\boldsymbol{\mu}_R^T\Phi_R^T\Phi_R(\sigma^2 I + \tilde{\Lambda}_{\epsilon,R}\Phi_R^T\Phi_R)^{-1}\boldsymbol{\mu}_R \\
&= \frac{1}{2}\boldsymbol{\mu}_R^T\tilde{\Lambda}_{\epsilon,R}^{-1}\tilde{\Lambda}_{\epsilon,R}\Phi_R^T\Phi_R(\sigma^2 I + \tilde{\Lambda}_{\epsilon,R}\Phi_R^T\Phi_R)^{-1}\boldsymbol{\mu}_R \\
&= \frac{1}{2}\boldsymbol{\mu}_R^T\tilde{\Lambda}_{\epsilon,R}^{-1}\boldsymbol{\mu}_R - \frac{1}{2}\boldsymbol{\mu}_R^T\tilde{\Lambda}_{\epsilon,R}^{-1}(I + \frac{1}{\sigma^2}\tilde{\Lambda}_{\epsilon,R}\Phi_R^T\Phi_R)^{-1}\boldsymbol{\mu}_R.
\end{aligned} \tag{85}$$

Using Lemma 27, we have

$$\begin{aligned}
&\frac{1}{2}\boldsymbol{\mu}_R^T\tilde{\Lambda}_{\epsilon,R}^{-1}\boldsymbol{\mu}_R - \frac{1}{2}\boldsymbol{\mu}_R^T\tilde{\Lambda}_{\epsilon,R}^{-1}(I + \frac{1}{\sigma^2}\tilde{\Lambda}_{\epsilon,R}\Phi_R^T\Phi_R)^{-1}\boldsymbol{\mu}_R \\
&= \frac{1}{2}\boldsymbol{\mu}_R^T\tilde{\Lambda}_{\epsilon,R}^{-1}\boldsymbol{\mu}_R - \frac{1}{2}\boldsymbol{\mu}_R^T\tilde{\Lambda}_{\epsilon,R}^{-1}(I + \frac{n}{\sigma^2}\tilde{\Lambda}_{\epsilon,R})^{-1}\boldsymbol{\mu}_R \\
&\quad + \frac{1}{2}\sum_{j=1}^{\infty}(-1)^{j+1}\boldsymbol{\mu}_R^T\tilde{\Lambda}_{\epsilon,R}^{-1}\left(\frac{1}{\sigma^2}(I + \frac{n}{\sigma^2}\tilde{\Lambda}_{\epsilon,R})^{-1}\tilde{\Lambda}_{\epsilon,R}(\Phi_R^T\Phi_R - nI)\right)^j(I + \frac{n}{\sigma^2}\tilde{\Lambda}_{\epsilon,R})^{-1}\boldsymbol{\mu}_R \\
&= \frac{n}{2\sigma^2}\boldsymbol{\mu}_R^T(I + \frac{n}{\sigma^2}\tilde{\Lambda}_{\epsilon,R})^{-1}\boldsymbol{\mu}_R \\
&\quad + \frac{1}{2}\sum_{j=1}^{\infty}(-1)^{j+1}\boldsymbol{\mu}_R^T\frac{1}{\sigma^2}(I + \frac{n}{\sigma^2}\tilde{\Lambda}_{\epsilon,R})^{-1}(\Phi_R^T\Phi_R - nI)\left(\frac{1}{\sigma^2}(I + \frac{n}{\sigma^2}\tilde{\Lambda}_{\epsilon,R})^{-1}\tilde{\Lambda}_{\epsilon,R}(\Phi_R^T\Phi_R - nI)\right)^{j-1} \\
&\qquad (I + \frac{n}{\sigma^2}\tilde{\Lambda}_{\epsilon,R})^{-1}\boldsymbol{\mu}_R
\end{aligned} \tag{86}$$

Letting $\epsilon \to 0$, we get

$$\begin{aligned}
T_{2,R}(D_n) &= \frac{1}{2\sigma^2}\boldsymbol{\mu}_R^T\Phi_R^T(I + \frac{1}{\sigma^2}\Phi_R\Lambda_R\Phi_R^T)^{-1}\Phi_R\boldsymbol{\mu}_R \\
&= \frac{n}{2\sigma^2}\boldsymbol{\mu}_R^T(I + \frac{n}{\sigma^2}\Lambda_R)^{-1}\boldsymbol{\mu}_R \\
&\quad + \frac{1}{2}\sum_{j=1}^{\infty}\left[(-1)^{j+1}\boldsymbol{\mu}_R^T\frac{1}{\sigma^2}(I + \frac{n}{\sigma^2}\Lambda_R)^{-1}(\Phi_R^T\Phi_R - nI)\left(\frac{1}{\sigma^2}(I + \frac{n}{\sigma^2}\Lambda_R)^{-1}\Lambda_R(\Phi_R^T\Phi_R - nI)\right)^{j-1} \right. \\
&\qquad \left. (I + \frac{n}{\sigma^2}\Lambda_R)^{-1}\boldsymbol{\mu}_R\right]
\end{aligned}$$

This concludes the proof. $\qquad\qquad\square$

**Lemma 35.** *Assume that $\sigma^2 = \Theta(1)$. Let $R = n^{\frac{1}{\alpha}+\kappa}$ where $0 < \kappa < \frac{\alpha-1-2\tau}{2\alpha^2}$. Under Assumptions 4, 5 and 6, with probability of at least $1-\delta$, we have*

$$T_{1,R}(D_n) = \left(\frac{1}{2}\log\det(I + \frac{n}{\sigma^2}\Lambda_R) - \frac{1}{2}\text{Tr}\left(I - (I + \frac{n}{\sigma^2}\Lambda_R)^{-1}\right)\right)(1 + o(1)) = \Theta(n^{\frac{1}{\alpha}}). \tag{87}$$

*Furthermore, if we assume $\mu_0 = 0$, we have*

$$T_{2,R}(D_n) = \left(\frac{n}{2\sigma^2}\boldsymbol{\mu}_R^T(I + \frac{n}{\sigma^2}\Lambda_R)^{-1}\boldsymbol{\mu}_R\right)(1+o(1)) = \begin{cases} \Theta(n^{\max\{0,1+\frac{1-2\beta}{\alpha}\}}), & \alpha \neq 2\beta - 1, \\ \Theta(\log n), & \alpha = 2\beta - 1. \end{cases} \quad (88)$$

*Proof of Lemma 35.* Let

$$A = (I + \frac{n}{\sigma^2}\Lambda_R)^{-\gamma/2}\Lambda_R^{\gamma/2}(\Phi_R^T\Phi_R - nI)\Lambda_R^{\gamma/2}(I + \frac{n}{\sigma^2}\Lambda_R)^{-\gamma/2}, \quad (89)$$

where $\frac{1+\alpha+2\tau}{2\alpha} < \gamma \leq 1$. By Corollary 22, with probability of at least $1-\delta$, we have

$$\|A\|_2 = \tilde{O}(n^{\frac{1-2\gamma\alpha+\alpha+2\tau}{2\alpha}}). \quad (90)$$

When $n$ is sufficiently large, $\|A\|_2$ is less than 1. Let $B = (I + \frac{n}{\sigma^2}\Lambda_R)^{-1/2}\Lambda_R^{1/2}(\Phi_R^T\Phi_R - nI)\Lambda_R^{1/2}(I + \frac{n}{\sigma^2}\Lambda_R)^{-1/2}$. Then $\|B\|_2 = \frac{\sigma^{2(1-\gamma)}}{n^{1-\gamma}}\|A\|_2 = \tilde{O}(n^{\frac{1-\alpha+2\tau}{2\alpha}})$. Using the Woodbury matrix identity, we compute $T_{1,R}(D_n)$ as follows:

$$\begin{aligned}
T_{1,R}(D_n) &= \tfrac{1}{2}\text{logdet}(I + \tfrac{1}{\sigma^2}\Lambda_R\Phi_R^T\Phi_R) - \tfrac{1}{2}\text{Tr}\Phi_R(\sigma^2 I + \Lambda_R\Phi_R^T\Phi_R)^{-1}\Lambda_R\Phi_R^T \\
&= \tfrac{1}{2}\text{logdet}(I + \tfrac{n}{\sigma^2}\Lambda_R) + \tfrac{1}{2}\text{logdet}[I + \tfrac{1}{\sigma^2}(I + \tfrac{n}{\sigma^2}\Lambda_R)^{-1/2}\Lambda_R^{1/2}(\Phi_R^T\Phi_R - nI)\Lambda_R^{1/2}(I + \tfrac{n}{\sigma^2}\Lambda_R)^{-1/2}] \\
&\quad - \tfrac{1}{2}\text{Tr}(\sigma^2 I + \Lambda\Phi_R^T\Phi_R)^{-1}\Lambda\Phi_R^T\Phi_R \\
&= \tfrac{1}{2}\text{logdet}(I + \tfrac{n}{\sigma^2}\Lambda_R) + \tfrac{1}{2}\text{Trlog}[I + \tfrac{1}{\sigma^2}B] - \tfrac{1}{2}\text{Tr}(I - \sigma^2(\sigma^2 I + \Lambda\Phi_R^T\Phi_R)^{-1})) \\
&= \tfrac{1}{2}\text{logdet}(I + \tfrac{n}{\sigma^2}\Lambda_R) + \tfrac{1}{2}\text{Tr}\sum_{j=1}^{\infty}\tfrac{(-1)^{j-1}}{j}(\tfrac{1}{\sigma^2}B)^j \\
&\quad - \tfrac{1}{2}\text{Tr}\left(I - (I + \tfrac{n}{\sigma^2}\Lambda_R)^{-1} + \sum_{j=1}^{\infty}(-1)^j\left(\tfrac{1}{\sigma^2}(I + \tfrac{n}{\sigma^2}\Lambda_R)^{-1}\Lambda_R(\Phi_R^T\Phi_R - nI)\right)^j(I + \tfrac{n}{\sigma^2}\Lambda_R)^{-1}\right) \\
&= \left(\tfrac{1}{2}\text{logdet}(I + \tfrac{n}{\sigma^2}\Lambda_R) - \tfrac{1}{2}\text{Tr}\left(I - (I + \tfrac{n}{\sigma^2}\Lambda_R)^{-1}\right)\right) + \tfrac{1}{2}\text{Tr}\sum_{j=1}^{\infty}\tfrac{(-1)^{j-1}}{j}(\tfrac{1}{\sigma^2}B)^j \\
&\quad - \tfrac{1}{2}\text{Tr}\left(\sum_{j=1}^{\infty}(-1)^j\tfrac{1}{\sigma^{2j}}(I + \tfrac{n}{\sigma^2}\Lambda_R)^{-1/2}B^j(I + \tfrac{n}{\sigma^2}\Lambda_R)^{-1/2}\right),
\end{aligned} \quad (91)$$

where in the last equality we apply Lemma 27.

Let $h(x) = \log(1+x) - (1 - \frac{1}{1+x})$. It is easy to verify that $h(x)$ is increasing on $[0, +\infty)$. As for the first term on the right hand side of (91), we have

$$\begin{aligned}
&\tfrac{1}{2}\text{logdet}(I + \tfrac{n}{\sigma^2}\Lambda_R) - \tfrac{1}{2}\text{Tr}\left(I - (I + \tfrac{n}{\sigma^2}\Lambda_R)^{-1}\right) \\
&= \tfrac{1}{2}\sum_{p=1}^{R}\left(\log(1 + \tfrac{n}{\sigma^2}\lambda_p) - (1 - \tfrac{1}{1 + \frac{n}{\sigma^2}\lambda_p})\right) \\
&= \tfrac{1}{2}\sum_{p=1}^{R}h(\tfrac{n}{\sigma^2}\lambda_p) \leq \tfrac{1}{2}\sum_{p=1}^{R}h(\tfrac{n}{\sigma^2}\overline{C_\lambda}p^{-\alpha}) \\
&\leq \tfrac{1}{2}h(\tfrac{n}{\sigma^2}\overline{C_\lambda}) + \tfrac{1}{2}\int_{[1,R]}h(\tfrac{n}{\sigma^2}\overline{C_\lambda}x^{-\alpha})\mathrm{d}x \\
&= \tfrac{1}{2}h(\tfrac{n}{\sigma^2}\overline{C_\lambda}) + \tfrac{1}{2}n^{1/\alpha}\int_{[1/n^{1/\alpha},R/n^{1/\alpha}]}h(\tfrac{\overline{C_\lambda}}{\sigma^2}x^{-\alpha})\mathrm{d}x \\
&= \Theta(n^{1/\alpha}),
\end{aligned}$$

where in the last equality we use the fact that $\int_{[0,+\infty]} h(x^{-\alpha})dx < \infty$. On the other hand, we have

$$\frac{1}{2}\text{logdet}(I+\frac{n}{\sigma^2}\Lambda_R)-\frac{1}{2}\text{Tr}\big(I-(I+\frac{n}{\sigma^2}\Lambda_R)^{-1}\big)$$

$$=\frac{1}{2}\sum_{p=1}^{R}h(\frac{n}{\sigma^2}\lambda_p)\geq\frac{1}{2}\sum_{p=1}^{R}h(\frac{n}{\sigma^2}\underline{C_\lambda}p^{-\alpha})$$

$$\geq\frac{1}{2}\int_{[1,R+1]}h(\frac{n}{\sigma^2}\underline{C_\lambda}x^{-\alpha})dx$$

$$=\frac{1}{2}n^{1/\alpha}\int_{[1/n^{1/\alpha},(R+1)/n^{1/\alpha}]}h(\frac{1}{\sigma^2}\underline{C_\lambda}x^{-\alpha})dx$$

$$=\Theta(n^{1/\alpha}).$$

Overall, we have $\frac{1}{2}\text{logdet}(I+\frac{n}{\sigma^2}\Lambda_R)-\frac{1}{2}\text{Tr}\big(I-(I+\frac{n}{\sigma^2}\Lambda_R)^{-1}\big)=\Theta(n^{1/\alpha})$.

As for the second term on the right hand side of (91), we have

$$\left|\text{Tr}\sum_{j=1}^{\infty}\frac{(-1)^{j-1}}{j}(\frac{1}{\sigma^2}B)^j\right|\leq R\sum_{j=1}^{\infty}\|\frac{1}{\sigma^2}B\|_2^j=R\sum_{j=1}^{\infty}\frac{1}{\sigma^{2j}}\tilde{O}(n^{\frac{j(1-\alpha+2\tau)}{2\alpha}})$$

$$=R\tilde{O}(n^{\frac{1-\alpha+2\tau}{2\alpha}})=\tilde{O}(n^{\frac{1}{\alpha}+\kappa+\frac{1-\alpha+2\tau}{2\alpha}}).$$

As for the third term on the right hand side of (91), we have

$$\left|\text{Tr}\left(\sum_{j=1}^{\infty}(-1)^j\frac{1}{\sigma^{2j}}(I+\frac{n}{\sigma^2}\Lambda_R)^{-1/2}B^j(I+\frac{n}{\sigma^2}\Lambda_R)^{-1/2}\right)\right|$$

$$\leq\sum_{j=1}^{\infty}\left|\text{Tr}\left(\frac{1}{\sigma^{2j}}(I+\frac{n}{\sigma^2}\Lambda_R)^{-1/2}B^j(I+\frac{n}{\sigma^2}\Lambda_R)^{-1/2}\right)\right|$$

$$\leq R\sum_{j=1}^{\infty}\left\|\frac{1}{\sigma^{2j}}(I+\frac{n}{\sigma^2}\Lambda_R)^{-1/2}B^j(I+\frac{n}{\sigma^2}\Lambda_R)^{-1/2}\right\|_2$$

$$\leq R\sum_{j=1}^{\infty}\left\|\frac{1}{\sigma^{2j}}(I+\frac{n}{\sigma^2}\Lambda_R)^{-1/2}B^j(I+\frac{n}{\sigma^2}\Lambda_R)^{-1/2}\right\|_2$$

$$\leq R\sum_{j=1}^{\infty}\left\|\frac{1}{\sigma^{2j}}B^j\right\|_2=\tilde{O}(n^{\frac{1}{\alpha}+\kappa+\frac{1-\alpha+2\tau}{2\alpha}}).$$

Then the asymptotics of $T_{1,R}(D_n)$ is given by

$$T_{1,R}(D_n)=\frac{1}{2}\text{logdet}(I+\frac{n}{\sigma^2}\Lambda_R)-\frac{1}{2}\text{Tr}\big(I-(I+\frac{n}{\sigma^2}\Lambda_R)^{-1}\big)+\tilde{O}(n^{\frac{1}{\alpha}+\kappa+\frac{1-\alpha+2\tau}{2\alpha}})+\tilde{O}(n^{\frac{1}{\alpha}+\kappa+\frac{1-\alpha+2\tau}{2\alpha}})$$

$$=\Theta(n^{1/\alpha})+\tilde{O}(n^{\frac{1}{\alpha}+\kappa+\frac{1-\alpha+2\tau}{2\alpha}})$$

$$=\Theta(n^{\frac{1}{\alpha}}),$$

where in the last inequality we use the assumption that $\kappa<\frac{\alpha-1-2\tau}{2\alpha}$. Since $\tilde{O}(n^{\frac{1}{\alpha}+\kappa+\frac{1-\alpha+2\tau}{2\alpha}})$ is lower order term compared to $\Theta(n^{\frac{1}{\alpha}})$, we further have

$$T_{1,R}(D_n)=\big(\frac{1}{2}\text{logdet}(I+\frac{n}{\sigma^2}\Lambda_R)-\frac{1}{2}\text{Tr}\big(I-(I+\frac{n}{\sigma^2}\Lambda_R)^{-1}\big)\big)(1+o(1)).$$

This concludes the proof of the first statement.

Let $\Lambda_{1:R}=\text{diag}\{\lambda_1,...,\lambda_R\}$, $\Phi_{1:R}=(\phi_1(\mathbf{x}),\phi_1(\mathbf{x}),...,\phi_R(\mathbf{x}))$ and $\boldsymbol{\mu}_{1:R}=(\mu_1,...,\mu_R)$. Since $\mu_0=0$, we have $T_{2,R}(D_n)=\frac{1}{2\sigma^2}\boldsymbol{\mu}_{1:R}^T\Phi_{1:R}^T(I+\frac{1}{\sigma^2}\Phi_{1:R}\Lambda_{1:R}\Phi_{1:R}^T)^{-1}\Phi_{1:R}\boldsymbol{\mu}_{1:R}$. According to Lemma 34, we

have

$$
\begin{aligned}
T_{2,R}(D_n) &= \frac{n}{2\sigma^2}\boldsymbol{\mu}_{1:R}^T(I+\frac{n}{\sigma^2}\Lambda_{1:R})^{-1}\boldsymbol{\mu}_{1:R} \\
&\quad + \frac{1}{2}\sum_{j=1}^{\infty}(-1)^{j+1}\boldsymbol{\mu}_{1:R}^T\frac{1}{\sigma^2}(I+\frac{n}{\sigma^2}\Lambda_{1:R})^{-1}(\Phi_{1:R}^T\Phi_{1:R}-nI)\left(\frac{1}{\sigma^2}(I+\frac{n}{\sigma^2}\Lambda_{1:R})^{-1}\Lambda_{1:R}(\Phi_{1:R}^T\Phi_{1:R}-nI)\right)^{j-1} \\
&= \frac{n}{2\sigma^2}\boldsymbol{\mu}_{1:R}^T(I+\frac{n}{\sigma^2}\Lambda_{1:R})^{-1}\boldsymbol{\mu}_{1:R} \\
&\quad + \frac{1}{2}\sum_{j=1}^{\infty}\left[(-1)^{j+1}\frac{1}{\sigma^{2j}}\boldsymbol{\mu}_{1:R}^T(I+\frac{n}{\sigma^2}\Lambda_{1:R})^{-1+\gamma/2}\Lambda_{1:R}^{-\gamma/2}A\left((I+\frac{n}{\sigma^2}\Lambda_{1:R})^{-1+\gamma}\Lambda_{1:R}^{1-\gamma}A\right)^{j-1}\right. \\
&\qquad\qquad \left.(I+\frac{n}{\sigma^2}\Lambda_{1:R})^{-1+\gamma/2}\Lambda_{1:R}^{-\gamma/2}\boldsymbol{\mu}_{1:R}\right]
\end{aligned}
$$

(92)

where in the second to last equality we used the definition of $A$ (89). As for the first term on the right hand side of (92), by Lemma 15, Assumption 4 and Assumption 5, we have

$$
\frac{n}{2\sigma^2}\boldsymbol{\mu}_{1:R}^T(I+\frac{n}{\sigma^2}\Lambda_{1:R})^{-1}\boldsymbol{\mu}_{1:R} \le \frac{n}{2\sigma^2}\sum_{p=1}^{R}\frac{C_\mu^2 p^{-2\beta}}{1+\frac{n}{\sigma^2}\underline{C_\lambda}p^{-\alpha}} = \begin{cases}\Theta(n^{\max\{0,1+\frac{1-2\beta}{\alpha}\}}), & \alpha\neq 2\beta-1, \\ \Theta(\log n), & \alpha=2\beta-1.\end{cases}
$$

On the other hand, by Assumption 5, assuming that $\sup_{i\ge 1}p_{i+1}-p_i=h$, we have

$$
\begin{aligned}
\frac{n}{2\sigma^2}\boldsymbol{\mu}_{1:R}^T(I+\frac{n}{\sigma^2}\Lambda_{1:R})^{-1}\boldsymbol{\mu}_{1:R} &\ge \frac{n}{2\sigma^2}\sum_{i=1}^{\lfloor\frac{R}{h}\rfloor}\frac{C_\mu^2 p_i^{-2\beta}}{1+\frac{n}{\sigma^2}\overline{C_\lambda}p_i^{-\alpha}} \\
&\ge \frac{n}{2\sigma^2}\sum_{i=1}^{\lfloor\frac{R}{h}\rfloor}\frac{C_\mu^2 i^{-2\beta}}{1+\frac{n}{\sigma^2}\overline{C_\lambda}(hi)^{-\alpha}} \\
&= \begin{cases}\Theta(n^{\max\{0,1+\frac{1-2\beta}{\alpha}\}}), & \alpha\neq 2\beta-1, \\ \Theta(\log n), & \alpha=2\beta-1.\end{cases}
\end{aligned}
$$

Overall, we have

$$
\frac{n}{2\sigma^2}\boldsymbol{\mu}_{1:R}^T(I+\frac{n}{\sigma^2}\Lambda_{1:R})^{-1}\boldsymbol{\mu}_{1:R} = \Theta(n^{\max\{0,1+\frac{1-2\beta}{\alpha}\}}\log^k n), \text{ where } k=\begin{cases}0, & \alpha\neq 2\beta-1, \\ 1, & \alpha=2\beta-1.\end{cases}
$$

By Lemma 16, we have

$$
\begin{aligned}
\|(I+\frac{n}{\sigma^2}\Lambda_{1:R})^{-1+\gamma/2}\Lambda_{1:R}^{-\gamma/2}\boldsymbol{\mu}_{1:R}\|_2^2 &\le \sum_{p=1}^{R}\frac{C_\mu^2 p^{-2\beta}(\underline{C_\lambda}p^{-\alpha})^{-\gamma}}{(1+\frac{n}{\sigma^2}\underline{C_\lambda}p^{-\alpha})^{2-\gamma}} \\
&= \tilde{O}(\max\{n^{-2+\gamma}, R^{1-2\beta+\alpha\gamma}\}) \\
&= \tilde{O}(n^{\max\{-2+\gamma,\frac{1-2\beta}{\alpha}+\gamma+\kappa(1-2\beta+\alpha\gamma)\}}).
\end{aligned}
$$

(93)

Using (90), the second term on the right hand side of (92) is computed as follows:

$$
\begin{aligned}
&\frac{1}{2}\sum_{j=1}^{\infty}\left[(-1)^{j+1}\frac{1}{\sigma^{2j}}\boldsymbol{\mu}_{1:R}^T(I+\frac{n}{\sigma^2}\Lambda_{1:R})^{-1+\gamma/2}\Lambda_{1:R}^{-\gamma/2}A\left((I+\frac{n}{\sigma^2}\Lambda_{1:R})^{-1+\gamma}\Lambda_{1:R}^{1-\gamma}A\right)^{j-1}\right. \\
&\qquad\qquad \left.(I+\frac{n}{\sigma^2}\Lambda_{1:R})^{-1+\gamma/2}\Lambda_{1:R}^{-\gamma/2}\boldsymbol{\mu}_{1:R}\right] \\
&\le \frac{1}{2}\sum_{j=1}^{\infty}\frac{1}{\sigma^{2j}}\|A\|^j\left(\frac{n}{\sigma^2}\right)^{(-1+\gamma)(j-1)}\|(I+\frac{n}{\sigma^2}\Lambda_{1:R})^{-1+\gamma/2}\Lambda_{1:R}^{-\gamma/2}\boldsymbol{\mu}_{1:R}\|_2^2 \\
&\le \frac{1}{2}\sum_{j=1}^{\infty}\frac{1}{\sigma^{2j}}\tilde{O}(n^{\frac{j(1-2\gamma\alpha+\alpha+2\tau)}{2\alpha}})\left(\frac{n}{\sigma^2}\right)^{(-1+\gamma)(j-1)}\tilde{O}(n^{\max\{-2+\gamma,\frac{1-2\beta}{\alpha}+\gamma+\kappa(1-2\beta+\alpha\gamma)\}}) \\
&= \tilde{O}(n^{\max\{-2+\gamma+\frac{1-2\gamma\alpha+\alpha+2\tau}{2\alpha},\frac{1-2\beta}{\alpha}+\gamma+\frac{1-2\gamma\alpha+\alpha+2\tau}{2\alpha}+\kappa(1-2\beta+\alpha\gamma)\}}) \\
&= \tilde{O}(n^{\max\{-2+\frac{1+\alpha+2\tau}{2\alpha},\frac{1-2\beta}{\alpha}+\frac{1+\alpha+2\tau}{2\alpha}+\kappa(1-2\beta+\alpha\gamma)\}}).
\end{aligned}
$$

(94)

Since $\frac{1+\alpha+2\tau}{2\alpha} < \frac{1+\alpha+2\tau}{\alpha+1+2\tau} = 1$, we have $-2 + \frac{1+\alpha+2\tau}{2\alpha} < 0$. Also we have

$$
\begin{aligned}
&\frac{1-2\beta}{\alpha} + \frac{1+\alpha+2\tau}{2\alpha} + \kappa(1-2\beta+\alpha\gamma) \\
&= \frac{1-2\beta}{\alpha} + 1 + \frac{1-\alpha+2\tau}{2\alpha} + \kappa(1-2\beta+\alpha\gamma) \\
&\leq \frac{1-2\beta}{\alpha} + 1 + \frac{1-\alpha+2\tau}{2\alpha} + \kappa\alpha\gamma \\
&< \frac{1-2\beta}{\alpha} + 1,
\end{aligned}
\tag{95}
$$

where the last inequality holds because $\kappa < \frac{\alpha-1-2\tau}{2\alpha^2}$ and $\gamma \leq 1$. Hence we have

$$
\begin{aligned}
T_{2,R}(D_n) &= \frac{n}{2\sigma^2}\boldsymbol{\mu}_{1:R}^T(I + \frac{n}{\sigma^2}\Lambda_{1:R})^{-1}\boldsymbol{\mu}_{1:R} + \tilde{O}(n^{\max\{-2+\frac{1+\alpha+2\tau}{2\alpha}, \frac{1-2\beta}{\alpha}+\frac{1+\alpha+2\tau}{2\alpha}+\kappa(1-2\beta+\alpha\gamma)\}}) \\
&= \Theta(n^{\max\{0,1+\frac{1-2\beta}{\alpha}\}}\log^k n) + \tilde{O}(n^{\max\{-2+\frac{1+\alpha+2\tau}{2\alpha}, \frac{1-2\beta}{\alpha}+\frac{1+\alpha+2\tau}{2\alpha}+\kappa(1-2\beta+\alpha\gamma)\}}) \\
&= \Theta(n^{\max\{0,1+\frac{1-2\beta}{\alpha}\}}\log^k n).
\end{aligned}
$$

where $k = \begin{cases} 0, & \alpha \neq 2\beta-1, \\ 1, & \alpha = 2\beta-1. \end{cases}$. Since $\tilde{O}(n^{\max\{-2+\frac{1+\alpha+2\tau}{2\alpha}, \frac{1-2\beta}{\alpha}+\frac{1+\alpha+2\tau}{2\alpha}+\kappa(1-2\beta+\alpha\gamma)\}})$ is lower order term compared to $\Theta(n^{\max\{0,1+\frac{1-2\beta}{\alpha}\}}\log^k n)$, we further have

$$
T_{2,R}(D_n) = \left(\frac{n}{2\sigma^2}\boldsymbol{\mu}_{1:R}^T(I + \frac{n}{\sigma^2}\Lambda_{1:R})^{-1}\boldsymbol{\mu}_{1:R}\right)(1+o(1))
$$

This concludes the proof of the second statement. □

**Lemma 36.** *Under Assumptions 4, 5 and 6, with probability of at least $1-5\delta$, we have*

$$
T_1(D_n) = \left(\frac{1}{2}\text{logdet}(I + \frac{n}{\sigma^2}\Lambda) - \frac{1}{2}\text{Tr}\left(I - (I + \frac{n}{\sigma^2}\Lambda)^{-1}\right)\right)(1+o(1)) = \Theta(n^{\frac{1}{\alpha}}), \tag{96}
$$

*Furthermore, let $\delta = n^{-q}$ where $0 \leq q < \min\{\frac{(2\beta-1)(\alpha-1-2\tau)}{4\alpha^2}, \frac{\alpha-1-2\tau}{2\alpha}\}$. If we assume $\mu_0 = 0$, we have*

$$
T_2(D_n) = \left(\frac{n}{2\sigma^2}\boldsymbol{\mu}^T(I + \frac{n}{\sigma^2}\Lambda)^{-1}\boldsymbol{\mu}\right)(1+o(1)) = \begin{cases} \Theta(n^{\max\{0,1+\frac{1-2\beta}{\alpha}\}}), & \alpha \neq 2\beta-1, \\ \Theta(\log n), & \alpha = 2\beta-1. \end{cases} \tag{97}
$$

*Proof of Lemma 36.* Let $R = n^{\frac{1}{\alpha}+\kappa}$ where $0 \leq \kappa < \frac{\alpha-1-2\tau}{2\alpha^2}$. By Lemmas 32 and 35, with probability of at least $1-5\delta$ we have

$$
|T_{1,R}(D_n) - T_1(D_n)| = \tilde{O}(n^{\frac{1}{\alpha}+\kappa(1-\alpha)}), \tag{98}
$$

and

$$
|T_{2,R}(D_n) - T_2(D_n)| = \tilde{O}\left((\frac{1}{\delta}+1)n^{\max\{(\frac{1}{\alpha}+\kappa)\frac{1-2\beta}{2}, 1+\frac{1-2\beta}{\alpha}+\frac{(1-2\beta)\kappa}{2}, -1-\kappa\alpha, 1+\frac{1-2\beta}{\alpha}-\kappa\alpha\}}\right) \tag{99}
$$

as well as

$$
T_{1,R}(D_n) = \left(\frac{1}{2}\text{logdet}(I + \frac{n}{\sigma^2}\Lambda_R) - \frac{1}{2}\text{Tr}\left(I - (I + \frac{n}{\sigma^2}\Lambda_R)^{-1}\right)\right)(1+o(1)) = \Theta(n^{\frac{1}{\alpha}}), \tag{100}
$$

and

$$
T_{2,R}(D_n) = \left(\frac{n}{2\sigma^2}\boldsymbol{\mu}^T(I + \frac{n}{\sigma^2}\Lambda)^{-1}\boldsymbol{\mu}\right)(1+o(1)) = \begin{cases} \Theta(n^{\max\{0,1+\frac{1-2\beta}{\alpha}\}}), & \alpha \neq 2\beta-1, \\ \Theta(\log n), & \alpha = 2\beta-1. \end{cases} \tag{101}
$$

We then have

$$
T_1(D_n) = T_{1,R}(D_n) + T_{1,R}(D_n) - T_1(D_n) = \Theta(n^{\frac{1}{\alpha}}) + \tilde{O}(n^{\frac{1}{\alpha}+\kappa(1-\alpha)}) = \Theta(n^{\frac{1}{\alpha}}).
$$

Since $\tilde{O}(n^{\frac{1}{\alpha}+\kappa(1-\alpha)})$ is lower order term compared to $\Theta(n^{\frac{1}{\alpha}})$, we further have

$$
T_1(D_n) = \left(\frac{1}{2}\text{logdet}(I + \frac{n}{\sigma^2}\Lambda_R) - \frac{1}{2}\text{Tr}\left(I - (I + \frac{n}{\sigma^2}\Lambda_R)^{-1}\right)\right)(1+o(1)) = \Theta(n^{\frac{1}{\alpha}})
$$

Besides, we have

$$\log\det(I+\frac{n}{\sigma^2}\Lambda)-\log\det(I+\frac{n}{\sigma^2}\Lambda_R)$$

$$=\sum_{p=R+1}^{\infty}\log(1+\frac{n}{\sigma^2}\lambda_p)\le\frac{n}{\sigma^2}\sum_{p=R+1}^{\infty}\lambda_p\le\frac{n}{\sigma^2}\sum_{p=R+1}^{\infty}C_\lambda p^{-\alpha}=\frac{n}{\sigma^2}O(R^{1-\alpha})$$

$$=\frac{n}{\sigma^2}O(n^{(1-\alpha)(\frac{1}{\alpha}+\kappa)})$$

$$=o(n^{\frac{1}{\alpha}}).$$

Then we have $\log\det(I+\frac{n}{\sigma^2}\Lambda_R)=\log\det(I+\frac{n}{\sigma^2}\Lambda)(1+o(1))$. Similarly we can prove $\mathrm{Tr}\big(I-(I+\frac{n}{\sigma^2}\Lambda)^{-1}\big)=\mathrm{Tr}\big(I-(I+\frac{n}{\sigma^2}\Lambda_R)^{-1}\big)(1+o(1))$. This concludes the proof of the first statement.

As for $T_2(D_n)$, we have

$$T_2(D_n)=T_{2,R}(D_n)+T_{2,R}(D_n)-T_2(D_n)$$

$$=\Theta(n^{\max\{0,1+\frac{1-2\beta}{\alpha}\}}\log^k n)+\tilde{O}\Big((\frac{1}{\delta}+1)n^{\max\{(\frac{1}{\alpha}+\kappa)\frac{1-2\beta}{2},1+\frac{1-2\beta}{\alpha}+\frac{(1-2\beta)\kappa}{2},-1-\kappa\alpha,1+\frac{1-2\beta}{\alpha}-\kappa\alpha\}}\Big)$$

$$=\Theta(n^{\max\{0,1+\frac{1-2\beta}{\alpha}\}}\log^k n)+\tilde{O}\Big(n^{q+\max\{(\frac{1}{\alpha}+\kappa)\frac{1-2\beta}{2},1+\frac{1-2\beta}{\alpha}+\frac{(1-2\beta)\kappa}{2},-1-\kappa\alpha,1+\frac{1-2\beta}{\alpha}-\kappa\alpha\}}\Big)$$

where we use $\delta=n^{-q}$, $k=\begin{cases}0, & \alpha\ne2\beta-1,\\1, & \alpha=2\beta-1.\end{cases}$

Since $0\le\kappa<\frac{\alpha-1-2\tau}{2\alpha^2}$ and $0\le q<\min\{\frac{(2\beta-1)(\alpha-1-2\tau)}{4\alpha^2},\frac{\alpha-1-2\tau}{2\alpha}\}$, we can choose $\kappa<\frac{\alpha-1-2\tau}{2\alpha^2}$ and $\kappa$ is arbitrarily close to $\frac{\alpha-1-2\tau}{2\alpha^2}$ such that $0\le q<\min\{\frac{(2\beta-1)\kappa}{2},\kappa\alpha\}$. Then we have $(\frac{1}{\alpha}+\kappa)\frac{1-2\beta}{2}+q<0$, $-1-\kappa\alpha+q<0$, $\frac{(1-2\beta)\kappa}{2}+q<0$ and $-\kappa\alpha+q<0$. So we have

$$T_{2,R}(D_n)=\Theta(n^{\max\{0,1+\frac{1-2\beta}{\alpha}\}}\log^k n).$$

Since $\tilde{O}\Big((\frac{1}{\delta}+1)n^{\max\{(\frac{1}{\alpha}+\kappa)\frac{1-2\beta}{2},1+\frac{1-2\beta}{\alpha}+\frac{(1-2\beta)\kappa}{2},-1-\kappa\alpha,1+\frac{1-2\beta}{\alpha}-\kappa\alpha\}}\Big)$ is lower order term compared to $\Theta(n^{\max\{0,1+\frac{1-2\beta}{\alpha}\}}\log^k n)$, we further have

$$T_2(D_n)=T_{2,R}(D_n)(1+o(1))=\Big(\frac{n}{2\sigma^2}\boldsymbol{\mu}_R^T(I+\frac{n}{\sigma^2}\Lambda_R)^{-1}\boldsymbol{\mu}_R\Big)(1+o(1)).$$

Furthermore, we have

$$\boldsymbol{\mu}^T(I+\frac{n}{\sigma^2}\Lambda)^{-1}\boldsymbol{\mu}-\boldsymbol{\mu}_R^T(I+\frac{n}{\sigma^2}\Lambda_R)^{-1}\boldsymbol{\mu}_R$$

$$=\sum_{p=R+1}^{\infty}\frac{\mu_p^2}{(1+\frac{n}{\sigma^2}\lambda_p)}\le\sum_{p=R+1}^{\infty}\mu_p^2\le\frac{n}{\sigma^2}\sum_{p=R+1}^{\infty}C_\mu^2 p^{-2\beta}=O(R^{1-2\beta})$$

$$=O(n^{(1-2\beta)(\frac{1}{\alpha}+\kappa)})$$

$$=o(n^{\frac{1-2\beta}{\alpha}}).$$

Then we have $\boldsymbol{\mu}^T(I+\frac{n}{\sigma^2}\Lambda)^{-1}\boldsymbol{\mu}=\boldsymbol{\mu}_R^T(I+\frac{n}{\sigma^2}\Lambda_R)^{-1}\boldsymbol{\mu}_R(1+o(1))$. This concludes the proof of the second statement. $\square$

*Proof of Theorem 7.* Using Lemma 36 and noting that $\frac{1}{\alpha}>0$, with probability of at least $1-5\tilde{\delta}$, we have

$$\mathbb{E}_\epsilon F^0(D_n)=T_1(D_n)+T_2(D_n)$$

$$=\Big[\frac{1}{2}\log\det(I+\frac{n}{\sigma^2}\Lambda_R)-\frac{1}{2}\mathrm{Tr}\Big(I-(I+\frac{n}{\sigma^2}\Lambda_R)^{-1}\Big)$$

$$+\frac{n}{2\sigma^2}\boldsymbol{\mu}_R^T(I+\frac{n}{\sigma^2}\Lambda_R)^{-1}\boldsymbol{\mu}_R\Big](1+o(1))$$

$$=\Theta(n^{\max\{\frac{1}{\alpha},\frac{1-2\beta}{\alpha}+1\}})$$

Letting $\delta=5\tilde{\delta}$, we get the result. $\square$

In the case of $\mu_0 > 0$, we have the following lemma:

**Lemma 37.** *Assume that $\sigma^2 = \Theta(1)$. Let $R = n^{\frac{1}{\alpha}+\kappa}$ where $0 < \kappa < \frac{\alpha-1-2\tau}{\alpha^2}$. Assume that $\mu_0 > 0$. Under Assumptions 4, 5 and 6, for sufficiently large $n$ with probability of at least $1-4\delta$ we have*

$$|T_{2,R}(D_n) - T_2(D_n)| = \tilde{O}\left((\frac{1}{\delta}+1)n^{\max\{1+(\frac{1}{\alpha}+\kappa)\frac{1-2\beta}{2},1-\kappa\alpha\}}\right). \tag{102}$$

*Proof of Lemma 37.* As for $|T_2(D_n) - T_{2,R}(D_n)|$, we have

$$|T_2(D_n) - T_{2,R}(D_n)| = \left| f(\mathbf{x})^T (I + \frac{\Phi\Lambda\Phi^T}{\sigma^2})^{-1} f(\mathbf{x}) - f_R(\mathbf{x})^T (I + \frac{\Phi\Lambda\Phi^T}{\sigma^2})^{-1} f_R(\mathbf{x}) \right|$$
$$+ \left| f_R(\mathbf{x})^T (I + \frac{\Phi\Lambda\Phi^T}{\sigma^2})^{-1} f_R(\mathbf{x}) - f_R(\mathbf{x})^T (I + \frac{\Phi_R\Lambda_R\Phi_R^T}{\sigma^2})^{-1} f_R(\mathbf{x}) \right|. \tag{103}$$

For the first term on the right-hand side of (103), we have

$$\left| f(\mathbf{x})^T (I + \frac{\Phi\Lambda\Phi^T}{\sigma^2})^{-1} f(\mathbf{x}) - f_R(\mathbf{x})^T (I + \frac{\Phi\Lambda\Phi^T}{\sigma^2})^{-1} f_R(\mathbf{x}) \right|$$
$$\leq 2 \left| f_{>R}(\mathbf{x})^T (I + \frac{\Phi\Lambda\Phi^T}{\sigma^2})^{-1} f_R(\mathbf{x}) \right| + \left| f_{>R}(\mathbf{x})^T (I + \frac{\Phi\Lambda\Phi^T}{\sigma^2})^{-1} f_{>R}(\mathbf{x}) \right|$$
$$\leq 2\|f_{>R}(\mathbf{x})\|_2 \|(I + \frac{\Phi\Lambda\Phi^T}{\sigma^2})^{-1} f_R(\mathbf{x})\|_2 + \|f_{>R}(\mathbf{x})\|_2 \|(I + \frac{\Phi\Lambda\Phi^T}{\sigma^2})^{-1}\|_2 \|f_{>R}(\mathbf{x})\|_2$$
$$\leq 2\|f_{>R}(\mathbf{x})\|_2 \|(I + \frac{\Phi\Lambda\Phi^T}{\sigma^2})^{-1} f_R(\mathbf{x})\|_2 + \|f_{>R}(\mathbf{x})\|_2^2.$$

Applying Corollary 19 and Lemma 31, with probability of at least $1-4\delta$, we have

$$\left| f(\mathbf{x})^T (I + \frac{\Phi\Lambda\Phi^T}{\sigma^2})^{-1} f(\mathbf{x}) - f_R(\mathbf{x})^T (I + \frac{\Phi\Lambda\Phi^T}{\sigma^2})^{-1} f_R(\mathbf{x}) \right|$$
$$\leq 2\tilde{O}\left(\sqrt{(\frac{1}{\delta}+1)nR^{1-2\beta}}\right) \tilde{O}(\sqrt{(\frac{1}{\delta}+1)n}) + \tilde{O}((\frac{1}{\delta}+1)nR^{1-2\beta})$$
$$= 2\tilde{O}\left((\frac{1}{\delta}+1)n^{1+(\frac{1}{\alpha}+\kappa)\frac{1-2\beta}{2}}\right) + \tilde{O}((\frac{1}{\delta}+1)n^{1+(\frac{1}{\alpha}+\kappa)(1-2\beta)})$$
$$= 2\tilde{O}\left((\frac{1}{\delta}+1)n^{1+(\frac{1}{\alpha}+\kappa)\frac{1-2\beta}{2}}\right).$$

As for the second term on the right-hand side of (80), according to Lemma 28, Corollary 26 and Lemma 30, we have

$$\left| f_R(\mathbf{x})^T (I + \frac{\Phi\Lambda\Phi^T}{\sigma^2})^{-1} f_R(\mathbf{x}) - f_R(\mathbf{x})^T (I + \frac{\Phi_R\Lambda_R\Phi_R^T}{\sigma^2})^{-1} f_R(\mathbf{x}) \right|$$
$$= \left| \sum_{j=1}^{\infty} (-1)^j f_R(\mathbf{x})^T \left( (I + \frac{\Phi_R\Lambda_R\Phi_R^T}{\sigma^2})^{-1} \frac{\Phi_{>R}\Lambda_{>R}\Phi_{>R}^T}{\sigma^2} \right)^j (I + \frac{\Phi_R\Lambda_R\Phi_R^T}{\sigma^2})^{-1} f_R(\mathbf{x}) \right|$$
$$\leq \sum_{j=1}^{\infty} \|(I + \frac{\Phi_R\Lambda_R\Phi_R^T}{\sigma^2})^{-1}\|_2^{j-1} \cdot \|\frac{\Phi_{>R}\Lambda_{>R}\Phi_{>R}^T}{\sigma^2}\|_2^j \cdot \|(I + \frac{\Phi_R\Lambda_R\Phi_R^T}{\sigma^2})^{-1} f_R(\mathbf{x})\|_2^2 \tag{104}$$
$$= \sum_{j=1}^{\infty} \tilde{O}(n^{-j\kappa\alpha}) \tilde{O}((\frac{1}{\delta}+1)n)$$
$$= \tilde{O}((\frac{1}{\delta}+1)n^{1-\kappa\alpha}).$$

By (80), we have

$$|T_2(D_n) - T_{2,R}(D_n)| = \tilde{O}\left((\frac{1}{\delta}+1)n^{1+(\frac{1}{\alpha}+\kappa)\frac{1-2\beta}{2}}\right) + \tilde{O}((\frac{1}{\delta}+1)n^{1-\kappa\alpha})$$

$$= \tilde{O}\left((\frac{1}{\delta}+1)n^{\max\{1+(\frac{1}{\alpha}+\kappa)\frac{1-2\beta}{2},1-\kappa\alpha\}}\right).$$

$\square$

**Lemma 38.** *Assume that $\sigma^2 = \Theta(1)$. Let $R = n^{\frac{1}{\alpha}+\kappa}$ where $0 < \kappa < \min\{\frac{\alpha-1-2\tau}{2\alpha^2}, \frac{2\beta-1}{\alpha^2}\}$. Assume that $\mu_0 > 0$. Under Assumptions 4, 5 and 6, with probability of at least $1-\delta$, we have*

$$T_{2,R}(D_n) = \frac{n}{2\sigma^2}\mu_0^2 + \tilde{O}(n^{\max\{\frac{1+7\alpha+2\tau}{8\alpha}, 1+\frac{1-2\beta}{\alpha}\}}). \tag{105}$$

*Proof of Lemma 38.* Let

$$A = (I + \frac{n}{\sigma^2}\Lambda_R)^{-\gamma/2}\Lambda_R^{\gamma/2}(\Phi_R^T\Phi_R - nI)\Lambda_R^{\gamma/2}(I + \frac{n}{\sigma^2}\Lambda_R)^{-\gamma/2}, \tag{106}$$

where $\frac{1+\alpha+2\tau}{2\alpha} < \gamma \le 1$. By Corollary 22, with probability of at least $1-\delta$, we have

$$\|A\|_2 = \tilde{O}(n^{\frac{1-2\gamma\alpha+\alpha+2\tau}{2\alpha}}). \tag{107}$$

When $n$ is sufficiently large, $\|A\|_2$ is less than 1. Let $\boldsymbol{\mu}_{R,1} = (\mu_0, 0, ..., 0)$ and $\boldsymbol{\mu}_{R,2} = (0, \mu_1, ..., \mu_R)$. Then $\boldsymbol{\mu}_R = \boldsymbol{\mu}_{R,1} + \boldsymbol{\mu}_{R,2}$. Let $\tilde{\Lambda}_{1,R} = \text{diag}\{1, \lambda_1, ..., \lambda_R\}$ and $I_{0,R} = (0, 1, ..., 1)$. Then $\Lambda_R = \tilde{\Lambda}_{1,R}I_{0,R}$. Let $B = (I + \frac{n}{\sigma^2}\Lambda_R)^{-1/2}\tilde{\Lambda}_{1,R}^{1/2}(\Phi_R^T\Phi_R - nI)\tilde{\Lambda}_{1,R}^{1/2}(I + \frac{n}{\sigma^2}\Lambda_R)^{-1/2}$. By Corollary 23, we have $\|B\|_2 = O(\sqrt{\log\frac{R}{\delta}}n^{\frac{1}{2}})$. By Lemma 34, we have

$$T_{2,R}(D_n) = \frac{n}{2\sigma^2}\boldsymbol{\mu}_R^T(I + \frac{n}{\sigma^2}\Lambda_R)^{-1}\boldsymbol{\mu}_R$$

$$+ \frac{1}{2}\sum_{j=1}^{\infty}\left[(-1)^{j+1}\boldsymbol{\mu}_R^T\frac{1}{\sigma^2}(I + \frac{n}{\sigma^2}\Lambda_R)^{-1}(\Phi_R^T\Phi_R - nI)\left(\frac{1}{\sigma^2}(I + \frac{n}{\sigma^2}\Lambda_R)^{-1}\Lambda_R(\Phi_R^T\Phi_R - nI)\right)^{j-1}\right.$$

$$\left.(I + \frac{n}{\sigma^2}\Lambda_R)^{-1}\boldsymbol{\mu}_R\right]$$

$$\tag{108}$$

As for the first term on the right hand side of (108), by Lemma 15, we have

$$\frac{n}{2\sigma^2}\mu^T(I + \frac{n}{\sigma^2}\Lambda)^{-1}\mu \le \frac{n}{2\sigma^2}\left(\mu_0^2 + \sum_{p=1}^{R}\frac{C_\mu^2 p^{-2\beta}}{1 + \frac{n}{\sigma^2}\underline{C_\lambda}p^{-\alpha}}\right) = \frac{n}{2\sigma^2}\mu_0^2 + \tilde{O}(n^{\max\{0, 1+\frac{1-2\beta}{\alpha}\}}).$$

We define $Q_{1,j}$, $Q_{2,j}$ and $Q_{3,j}$ by

$$Q_{1,j} = \boldsymbol{\mu}_{R,1}^T\frac{1}{\sigma^2}(I + \frac{n}{\sigma^2}\Lambda_R)^{-1}(\Phi_R^T\Phi_R - nI)\left(\frac{1}{\sigma^2}(I + \frac{n}{\sigma^2}\Lambda_R)^{-1}\Lambda_R(\Phi_R^T\Phi_R - nI)\right)^{j-1}$$

$$(I + \frac{n}{\sigma^2}\Lambda_R)^{-1}\boldsymbol{\mu}_{R,1}$$

$$Q_{2,j} = \boldsymbol{\mu}_{R,1}^T\frac{1}{\sigma^2}(I + \frac{n}{\sigma^2}\Lambda_R)^{-1}(\Phi_R^T\Phi_R - nI)\left(\frac{1}{\sigma^2}(I + \frac{n}{\sigma^2}\Lambda_R)^{-1}\Lambda_R(\Phi_R^T\Phi_R - nI)\right)^{j-1} \tag{109}$$

$$(I + \frac{n}{\sigma^2}\Lambda_R)^{-1}\boldsymbol{\mu}_{R,2}$$

$$Q_{3,j} = \boldsymbol{\mu}_{R,2}^T\frac{1}{\sigma^2}(I + \frac{n}{\sigma^2}\Lambda_R)^{-1}(\Phi_R^T\Phi_R - nI)\left(\frac{1}{\sigma^2}(I + \frac{n}{\sigma^2}\Lambda_R)^{-1}\Lambda_R(\Phi_R^T\Phi_R - nI)\right)^{j-1}$$

$$(I + \frac{n}{\sigma^2}\Lambda_R)^{-1}\boldsymbol{\mu}_{R,2}$$

The quantity $Q_{3,j}$ actually shows up in the case of $\mu_0 = 0$ in the proof of Lemma 35. By (92), (94) and (95), we have that

$$|\sum_{j=1}^{\infty}(-1)^{j+1}Q_{3,j}| = |\sum_{j=1}^{\infty}(-1)^{j+1}\tilde{O}(n^{\frac{(j-1)(1-\alpha+2\tau)}{2\alpha}})o(n^{\max\{0,1+\frac{1-2\beta}{\alpha}\}})| = o(n^{\max\{0,1+\frac{1-2\beta}{\alpha}\}}).$$

$$(110)$$

For $Q_{1,j}$, we have

$$\begin{aligned}
Q_{1,1} &= \frac{1}{\sigma^{2j}}\boldsymbol{\mu}_{R,1}^T(I+\frac{n}{\sigma^2}\Lambda_R)^{-1+\frac{\gamma}{2}}B(I+\frac{n}{\sigma^2}\Lambda_R)^{-1+\frac{\gamma}{2}}\boldsymbol{\mu}_{R,1} \\
&\leq \frac{1}{\sigma^{2j}}\|\boldsymbol{\mu}_{R,1}\|_2^2\|(I+\frac{n}{\sigma^2}\Lambda_R)^{-1+\frac{\gamma}{2}}\|_2^2\|B\|_2 \\
&= O(\sqrt{\log\frac{R}{\delta}}n^{\frac{1}{2}}),
\end{aligned}$$

where in the last equality we use $\|B\|_2 = O(\sqrt{\log\frac{R}{\delta}}n^{\frac{1}{2}})$. For $j \geq 2$, we have

$$\begin{aligned}
Q_{1,j} &= \frac{1}{\sigma^{2j}}\boldsymbol{\mu}_{R,1}^T(I+\frac{n}{\sigma^2}\Lambda_R)^{-1+\frac{\gamma}{2}}B\left((I+\frac{n}{\sigma^2}\Lambda_R)^{-1+\gamma}\Lambda_R^{1-\gamma}A\right)^{j-2}(I+\frac{n}{\sigma^2}\Lambda_R)^{-1+\gamma}\Lambda_R^{1-\gamma} \\
&\qquad B(I+\frac{n}{\sigma^2}\Lambda_R)^{-1+\frac{\gamma}{2}}\boldsymbol{\mu}_{R,1} \\
&\leq \frac{1}{\sigma^{2j}}\|\boldsymbol{\mu}_{R,1}\|_2^2\|(I+\frac{n}{\sigma^2}\Lambda_R)^{-1+\frac{\gamma}{2}}\|_2^2\|B\|_2^2\|A\|_2^{j-2}\|(I+\frac{n}{\sigma^2}\Lambda_R)^{-1+\gamma}\Lambda_R^{1-\gamma}\|_2^{j-1} \\
&= O(\log\frac{R}{\delta}n\cdot n^{\frac{(j-2)(1-2\gamma\alpha+\alpha+2\tau)}{2\alpha}}\cdot n^{-(1-\gamma)(j-1)}) \\
&= O(\log\frac{R}{\delta}n^{\gamma}\cdot n^{\frac{(j-2)(1-\alpha+2\tau)}{2\alpha}}).
\end{aligned}$$

Then we have

$$|\sum_{j=1}^{\infty}(-1)^{j+1}Q_{1,j}| \leq O(\sqrt{\log\frac{R}{\delta}}n^{\frac{1}{2}}) + \sum_{j=2}^{\infty}O(\log\frac{R}{\delta}n^{\gamma}\cdot n^{\frac{(j-2)(1-\alpha+2\tau)}{2\alpha}}) = O(\log\frac{R}{\delta}n^{\gamma}) \qquad (111)$$

For $Q_{2,j}$, we have

$$\begin{aligned}
Q_{2,j} &= \frac{1}{\sigma^{2j}}\boldsymbol{\mu}_{R,1}^T(I+\frac{n}{\sigma^2}\Lambda_R)^{-1+\frac{\gamma}{2}}B\left((I+\frac{n}{\sigma^2}\Lambda_R)^{-1+\gamma}\Lambda_R^{1-\gamma}A\right)^{j-1}(I+\frac{n}{\sigma^2}\Lambda)^{-1+\frac{\gamma}{2}}\tilde{\Lambda}_{1,R}^{-\frac{\gamma}{2}}\boldsymbol{\mu}_{R,2} \\
&\leq \frac{1}{\sigma^{2j}}\|\boldsymbol{\mu}_{R,1}\|_2\|B\|_2\|A\|_2^{j-1}\|(I+\frac{n}{\sigma^2}\Lambda_R)^{-1+\gamma}\Lambda_R^{1-\gamma}\|_2^{j-1}\|(I+\frac{n}{\sigma^2}\Lambda)^{-1+\frac{\gamma}{2}}\tilde{\Lambda}_{1,R}^{-\frac{\gamma}{2}}\boldsymbol{\mu}_{R,2}\|_2 \\
&= O(\sqrt{\log\frac{R}{\delta}}n^{\frac{1}{2}}\cdot n^{\frac{(j-1)(1-\alpha+2\tau)}{2\alpha}})\|(I+\frac{n}{\sigma^2}\Lambda)^{-1+\frac{\gamma}{2}}\tilde{\Lambda}_{1,R}^{-\frac{\gamma}{2}}\boldsymbol{\mu}_{R,2}\|_2.
\end{aligned}$$

Since $\|(I+\frac{n}{\sigma^2}\Lambda)^{-1+\frac{\gamma}{2}}\tilde{\Lambda}_{1,R}^{-\frac{\gamma}{2}}\boldsymbol{\mu}_{R,2}\|_2$ is actually the case of $\mu_0 = 0$, we can use (93) in the proof of Lemma 35 and get

$$\begin{aligned}
\|(I+\frac{n}{\sigma^2}\Lambda)^{-1+\frac{\gamma}{2}}\tilde{\Lambda}_{1,R}^{-\frac{\gamma}{2}}\boldsymbol{\mu}_{R,2}\|_2^2 &= \|(I+\frac{n}{\sigma^2}\Lambda_{1:R})^{-1+\gamma/2}\Lambda_{1:R}^{-\gamma/2}\boldsymbol{\mu}_{1:R}\|_2^2 \\
&= \tilde{O}(n^{\max\{-2+\gamma,\frac{1-2\beta}{\alpha}+\gamma+\kappa(1-2\beta+\alpha\gamma)\}}) \\
&= \tilde{O}(n^{\max\{-2+\gamma,\frac{1-2\beta}{\alpha}+\gamma+\kappa(1-2\beta+\alpha\gamma)\}}) \\
&= o(n^{\gamma}),
\end{aligned}$$

$$(112)$$

where in the last equality we use $\kappa < \frac{2\beta-1}{\alpha^2}$. Then we have

$$|\sum_{j=1}^{\infty}(-1)^{j+1}Q_{2,j}| \leq \sum_{j=1}^{\infty}o(\sqrt{\log\frac{R}{\delta}}n^{\frac{1+\gamma}{2}}\cdot n^{\frac{(j-1)(1-\alpha+2\tau)}{2\alpha}}) = o(\sqrt{\log\frac{R}{\delta}}n^{\frac{1+\gamma}{2}}) \qquad (113)$$

Choosing $\gamma = \frac{1}{2}(1 + \frac{1+\alpha+2\tau}{2\alpha}) = \frac{1+3\alpha+2\tau}{4\alpha} < 1$, we have

$$
T_{2,R}(D_n) = \frac{n}{2\sigma^2}\boldsymbol{\mu}_R^T(I + \frac{n}{\sigma^2}\Lambda_R)^{-1}\boldsymbol{\mu}_R + \sum_{j=1}^{\infty}(-1)^{j+1}(Q_{1,j} + Q_{2,j} + Q_{3,j})
$$

$$
= \frac{n}{2\sigma^2}\mu_0^2 + \tilde{O}(n^{\max\{0,1+\frac{1-2\beta}{\alpha}\}}) + o(n^{\max\{0,1+\frac{1-2\beta}{\alpha}\}}) + O(\log\frac{R}{\delta}n^{\gamma}) + o(\sqrt{\log\frac{R}{\delta}}n^{\frac{1+\gamma}{2}})
$$

$$
= \frac{n}{2\sigma^2}\mu_0^2 + \tilde{O}(n^{\max\{\frac{1+\gamma}{2},1+\frac{1-2\beta}{\alpha}\}})
$$

$$
= \frac{n}{2\sigma^2}\mu_0^2 + \tilde{O}(n^{\max\{\frac{1+7\alpha+2\tau}{8\alpha},1+\frac{1-2\beta}{\alpha}\}}).
$$

$\square$

*Proof of Theorem 8.* Let $R = n^{\frac{1}{\alpha}+\kappa}$ where $0 < \kappa < \min\{\frac{\alpha-1-2\tau}{2\alpha^2}, \frac{2\beta-1}{\alpha^2}\}$. Since $0 \leq q < \min\{\frac{2\beta-1}{2}, \alpha\} \cdot \min\{\frac{\alpha-1-2\tau}{2\alpha^2}, \frac{2\beta-1}{\alpha^2}\}$, we can choose $\kappa < \min\{\frac{\alpha-1-2\tau}{2\alpha^2}, \frac{2\beta-1}{\alpha^2}\}$ and $\kappa$ is arbitrarily close to $\kappa < \min\{\frac{\alpha-1-2\tau}{2\alpha^2}, \frac{2\beta-1}{\alpha^2}\}$ such that $0 \leq q < \min\{\frac{(2\beta-1)\kappa}{2}, \kappa\alpha\}$. Then we have $(\frac{1}{\alpha}+\kappa)\frac{1-2\beta}{2} + q < 0$, and $-\kappa\alpha + q < 0$. As for $T_2(D_n)$, we have

$$
T_2(D_n) \leq T_{2,R}(D_n) + |T_{2,R}(D_n) - T_2(D_n)|
$$

$$
= \frac{n}{2\sigma^2}\mu_0^2 + \tilde{O}(n^{\max\{\frac{1+7\alpha+2\tau}{8\alpha},1+\frac{1-2\beta}{\alpha}\}}) + \tilde{O}\left((\frac{1}{\delta}+1)n^{\max\{1+(\frac{1}{\alpha}+\kappa)\frac{1-2\beta}{2},1-\kappa\alpha\}}\right)
$$

$$
= \frac{n}{2\sigma^2}\mu_0^2 + \tilde{O}(n^{\max\{\frac{1+7\alpha+2\tau}{8\alpha},1+\frac{1-2\beta}{\alpha}\}}) + \tilde{O}\left(n^{q+\max\{1+(\frac{1}{\alpha}+\kappa)\frac{1-2\beta}{2},1-\kappa\alpha\}}\right)
$$

$$
= \frac{n}{2\sigma^2}\mu_0^2 + o(n).
$$

By Lemma 36, we have $T_1(D_n) = O(n^{\frac{1}{\alpha}})$. Hence $\mathbb{E}_\epsilon F^0(D_n) = T_1(D_n) + T_2(D_n) = \frac{n}{2\sigma^2}\mu_0^2 + o(n)$. $\square$

## D.2 Proofs related to the asymptotics of the generalization error

**Lemma 39.** *Assume $\sigma^2 = \Theta(n^t)$ where $1 - \frac{\alpha}{1+2\tau} < t < 1$. Let $R = n^{(\frac{2\alpha-1}{\alpha(\alpha-1)}+1)(1-t)}$. Under Assumptions 4, 5 and 6, with probability of at least $1-\delta$ over sample inputs $(x_i)_{i=1}^n$, we have*

$$
G_1(D_n) = \frac{1+o(1)}{2\sigma^2}\left(\mathrm{Tr}(I + \frac{n}{\sigma^2}\Lambda_R)^{-1}\Lambda_R - \|\Lambda_R^{1/2}(I + \frac{n}{\sigma^2}\Lambda_R)^{-1}\|_F^2\right) = \frac{1}{\sigma^2}\Theta\left(n^{\frac{(1-\alpha)(1-t)}{\alpha}}\right). \quad (114)
$$

*Proof of Lemma 39.* Let $G_{1,R}(D_n) = \mathbb{E}_{(x_{n+1},y_{n+1})}(T_{1,R}(D_{n+1}) - T_{1,R}(D_n))$, where $R = n^C$ for some constant C. By Lemma 32, we have that

$$
|G_1(D_n) - G_{1,R}(D_n)| = \left|\mathbb{E}_{(x_{n+1},y_{n+1})}[T_1(D_{n+1}) - T_{1,R}(D_{n+1})] - [T_1(D_n) - T_{1,R}(D_n)]\right|
$$

$$
= \left|\mathbb{E}_{(x_{n+1},y_{n+1})}O((n+1)R^{1-\alpha})\right| + \left|O(nR^{1-\alpha})\right| \quad (115)
$$

$$
= O(\frac{1}{\sigma^2}nR^{1-\alpha}).
$$

Define $\eta_R = (\phi_0(x_{n+1}), \phi_1(x_{n+1}), ..., \phi_R(x_{n+1}))^T$ and $\widetilde{\Phi}_R = (\Phi_R^T, \eta_R)^T$. As for $G_{1,R}(D_n)$, we have

$$
G_{1,R}(D_n) = \mathbb{E}_{(x_{n+1},y_{n+1})}(T_{1,R}(D_{n+1}) - T_{1,R}(D_n))
$$

$$
= \mathbb{E}_{(x_{n+1},y_{n+1})}\left(\frac{1}{2}\mathrm{logdet}(I + \frac{\widetilde{\Phi}_R\Lambda_R\widetilde{\Phi}_R^T}{\sigma^2}) - \frac{1}{2}\mathrm{Tr}(I - (I + \frac{\widetilde{\Phi}_R\Lambda_R\widetilde{\Phi}_R^T}{\sigma^2})^{-1})\right)
$$

$$
- \left(\frac{1}{2}\mathrm{logdet}(I + \frac{\Phi_R\Lambda_R\Phi_R^T}{\sigma^2}) - \frac{1}{2}\mathrm{Tr}(I - (I + \frac{\Phi_R\Lambda_R\Phi_R^T}{\sigma^2})^{-1})\right) \quad (116)
$$

$$
= \frac{1}{2}\left(\mathbb{E}_{(x_{n+1},y_{n+1})}\mathrm{logdet}(I + \frac{\widetilde{\Phi}_R\Lambda_R\widetilde{\Phi}_R^{T}}{\sigma^2}) - \mathrm{logdet}(I + \frac{\Phi_R\Lambda_R\Phi_R^T}{\sigma^2})\right)
$$

$$
- \frac{1}{2}\left(\mathbb{E}_{(x_{n+1},y_{n+1})}\mathrm{Tr}(I - (I + \frac{\widetilde{\Phi}_R\Lambda_R\widetilde{\Phi}_R^T}{\sigma^2})^{-1}) - \mathrm{Tr}(I - (I + \frac{\Phi_R\Lambda_R\Phi_R^T}{\sigma^2})^{-1})\right).
$$

As for the first term in the right hand side (116), we have

$$\frac{1}{2}\left(\mathbb{E}_{(x_{n+1},y_{n+1})}\text{logdet}(I+\frac{\widetilde{\Phi}_R\Lambda_R\widetilde{\Phi}_R^T}{\sigma^2})-\text{logdet}(I+\frac{\Phi_R\Lambda_R\Phi_R^T}{\sigma^2})\right)$$

$$=\frac{1}{2}\left(\mathbb{E}_{(x_{n+1},y_{n+1})}\text{logdet}(I+\frac{\Lambda_R\widetilde{\Phi}_R^T\widetilde{\Phi}_R}{\sigma^2})-\text{logdet}(I+\frac{\Lambda_R\Phi_R^T\Phi_R}{\sigma^2})\right)$$

$$=\frac{1}{2}\left(\mathbb{E}_{(x_{n+1},y_{n+1})}\text{logdet}(I+\frac{\Lambda_R\Phi_R^T\Phi_R+\eta_R\eta_R^T}{\sigma^2})-\text{logdet}(I+\frac{\Lambda_R\Phi_R^T\Phi_R}{\sigma^2})\right)$$

$$=\frac{1}{2}\left(\mathbb{E}_{(x_{n+1},y_{n+1})}\text{logdet}\left((I+\frac{\Lambda_R\Phi_R^T\Phi_R}{\sigma^2})^{-1}(I+\frac{\Lambda_R\Phi_R^T\Phi_R}{\sigma^2}+\frac{\Lambda_R\eta_R\eta_R^T}{\sigma^2})\right)\right)$$

$$=\frac{1}{2}\left(\mathbb{E}_{(x_{n+1},y_{n+1})}\text{logdet}\left(I+(I+\frac{\Lambda_R\Phi_R^T\Phi_R}{\sigma^2})^{-1}\frac{\Lambda_R\eta_R\eta_R^T}{\sigma^2}\right)\right)$$

$$=\frac{1}{2}\left(\mathbb{E}_{(x_{n+1},y_{n+1})}\log\left(1+\frac{1}{\sigma^2}\eta_R^T(I+\frac{\Lambda_R\Phi_R^T\Phi_R}{\sigma^2})^{-1}\Lambda_R\eta_R\right)\right)$$

Let

$$A=(I+\frac{n}{\sigma^2}\Lambda_R)^{-1/2}\Lambda_R^{1/2}(\Phi_R^T\Phi_R-nI)\Lambda_R^{1/2}(I+\frac{n}{\sigma^2}\Lambda_R)^{-1/2}. \tag{117}$$

According to Corollary 22, with probability of at least $1-\delta$, we have $\|\frac{1}{\sigma^2}A\|_2 = O(\sqrt{\log\frac{R}{\delta}}n^{\frac{1-\alpha+2\tau}{2\alpha}-\frac{(1+2\tau)t}{2\alpha}}) = o(1)$. When $n$ is sufficiently large, $\|\frac{1}{\sigma^2}A\|_2$ is less than 1. By Lemma 27, we have

$$\eta_R^T(I+\frac{\Lambda_R\Phi_R^T\Phi_R}{\sigma^2})^{-1}\Lambda_R\eta_R$$

$$=\eta_R^T(I+\frac{n}{\sigma^2}\Lambda_R)^{-1}\Lambda_R\eta_R+\sum_{j=1}^{\infty}(-1)^j\eta_R^T\left(\frac{1}{\sigma^2}(I+\frac{n}{\sigma^2}\Lambda_R)^{-1}\Lambda_R(\Phi_R^T\Phi_R-nI)\right)^j(I+\frac{n}{\sigma^2}\Lambda_R)^{-1}\Lambda_R\eta_R$$

$$=\eta_R^T(I+\frac{n}{\sigma^2}\Lambda_R)^{-1}\Lambda_R\eta_R+\sum_{j=1}^{\infty}(-1)^j\frac{1}{\sigma^{2j}}\eta_R^T(I+\frac{n}{\sigma^{2j}}\Lambda_R)^{-1/2}\Lambda_R^{1/2}A^j(I+\frac{n}{\sigma^2}\Lambda_R)^{-1/2}\Lambda_R^{1/2}\eta_R$$

$$\leq\eta_R^T(I+\frac{n}{\sigma^2}\Lambda_R)^{-1}\Lambda_R\eta_R+\sum_{j=1}^{\infty}\|\frac{1}{\sigma^2}A\|_2^j\|(I+\frac{n}{\sigma^2}\Lambda_R)^{-1/2}\Lambda_R^{1/2}\eta_R\|_2^2$$

$$\leq\sum_{p=1}^{R}\phi_p^2(x_{n+1})\frac{\overline{C_\lambda}p^{-\alpha}}{1+n\underline{C_\lambda}p^{-\alpha}/\sigma^2}+\sum_{j=1}^{\infty}\|\frac{1}{\sigma^2}A\|_2^j\sum_{p=1}^{R}\phi_p^2(x_{n+1})\frac{\overline{C_\lambda}p^{-\alpha}}{1+n\underline{C_\lambda}p^{-\alpha}/\sigma^2}$$

$$\leq\sum_{p=1}^{R}\frac{\overline{C_\lambda}p^{-\alpha}p^{2\tau}}{1+n\underline{C_\lambda}p^{-\alpha}/\sigma^2}+\sum_{j=1}^{\infty}\|\frac{1}{\sigma^2}A\|_2^j\sum_{p=1}^{R}\frac{\overline{C_\lambda}p^{-\alpha}p^{2\tau}}{1+n\underline{C_\lambda}p^{-\alpha}/\sigma^2}$$

$$\leq O(n^{\frac{(1-\alpha+2\tau)(1-t)}{\alpha}})+\sum_{j=1}^{\infty}\|\frac{1}{\sigma^2}A\|_2^j O(n^{\frac{(1-\alpha+2\tau)(1-t)}{\alpha}})$$

$$=O(n^{\frac{(1-\alpha+2\tau)(1-t)}{\alpha}})=o(1), \tag{118}$$

where we use Lemma 15 in the last inequality. Next we have

$$\frac{1}{2}\left(\mathbb{E}_{(x_{n+1},y_{n+1})}\log\det(I+\frac{\widetilde{\Phi}_R\Lambda_R\widetilde{\Phi}_R^T}{\sigma^2})-\log\det(I+\frac{\Phi_R\Lambda_R\Phi_R^T}{\sigma^2})\right)$$

$$=\frac{1}{2}\left(\mathbb{E}_{(x_{n+1},y_{n+1})}\log\left(1+\frac{1}{\sigma^2}\eta_R^T(I+\frac{\Lambda_R\Phi_R^T\Phi_R}{\sigma^2})^{-1}\Lambda_R\eta_R\right)\right)$$

$$=\frac{1}{2}\left(\mathbb{E}_{(x_{n+1},y_{n+1})}\left(\frac{1}{\sigma^2}\eta_R^T(I+\frac{\Lambda_R\Phi_R^T\Phi_R}{\sigma^2})^{-1}\Lambda_R\eta_R\right)(1+o(1))\right)$$

$$=\frac{1}{2\sigma^2}\left(\text{Tr}(I+\frac{\Lambda_R\Phi_R^T\Phi_R}{\sigma^2})^{-1}\Lambda_R\right)(1+o(1)),$$

where in the last equality we use the fact that $\mathbb{E}_{(x_{n+1},y_{n+1})}\eta_R\eta_R^T=I$. By Lemma 27, we have

$$\text{Tr}(I+\frac{\Lambda_R\Phi_R^T\Phi_R}{\sigma^2})^{-1}\Lambda_R$$

$$=\text{Tr}(I+\frac{n}{\sigma^2}\Lambda_R)^{-1}\Lambda_R+\sum_{j=1}^{\infty}(-1)^j\text{Tr}\left(\frac{1}{\sigma^2}(I+\frac{n}{\sigma^2}\Lambda_R)^{-1}\Lambda_R(\Phi_R^T\Phi_R-nI)\right)^j(I+\frac{n}{\sigma^2}\Lambda_R)^{-1}\Lambda_R$$

$$=\text{Tr}(I+\frac{n}{\sigma^2}\Lambda_R)^{-1}\Lambda_R+\sum_{j=1}^{\infty}(-1)^j\text{Tr}\frac{1}{\sigma^{2j}}(I+\frac{n}{\sigma^2}\Lambda_R)^{-1/2}\Lambda_R^{1/2}A^j(I+\frac{n}{\sigma^2}\Lambda_R)^{-1/2}\Lambda_R^{1/2}.$$

By Lemma 15, we have

$$\text{Tr}(I+\frac{n}{\sigma^2}\Lambda_R)^{-1}\Lambda_R\leq\sum_{p=1}^{R}\frac{\overline{C_\lambda}p^{-\alpha}}{1+n\underline{C_\lambda}p^{-\alpha}/\sigma^2}=\Theta(n^{\frac{(1-\alpha)(1-t)}{\alpha}})$$

$$\text{Tr}(I+\frac{n}{\sigma^2}\Lambda_R)^{-1}\Lambda_R\geq\sum_{p=1}^{R}\frac{\underline{C_\lambda}p^{-\alpha}}{1+n\overline{C_\lambda}p^{-\alpha}/\sigma^2}=\Theta(n^{\frac{(1-\alpha)(1-t)}{\alpha}}).$$

Overall,

$$\text{Tr}(I+\frac{n}{\sigma^2}\Lambda_R)^{-1}\Lambda_R=\Theta(n^{\frac{(1-\alpha)(1-t)}{\alpha}}). \tag{119}$$

Since $\|\frac{1}{\sigma^2}A\|_2^j=o(1)$, we have that the absolute values of diagonal entries of $\frac{1}{\sigma^{2j}}A^j$ are at most $o(1)$. Let $(A^j)_{p,p}$ denote the $(p,p)$-th entry of the matrix $A^j$. Then we have

$$\left|\text{Tr}\frac{1}{\sigma^{2j}}(I+\frac{n}{\sigma^2}\Lambda_R)^{-1/2}\Lambda_R^{1/2}A^j(I+\frac{n}{\sigma^2}\Lambda_R)^{-1/2}\Lambda_R^{1/2}\right|$$

$$=\left|\sum_{p=1}^{R}\frac{\lambda_p\frac{1}{\sigma^{2j}}(A^j)_{p,p}}{1+n\lambda_p/\sigma^2}\right|\leq\sum_{p=1}^{R}\frac{\lambda_p\|\frac{1}{\sigma^{2j}}A\|_2^j}{1+n\lambda_p/\sigma^2}=\Theta(n^{\frac{(1-\alpha)(1-t)}{\alpha}})\tilde{O}(n^{\frac{j(1-\alpha+2\tau-(1+2\tau)t)}{2\alpha}}(\log R)^{j/2}), \tag{120}$$

where in the last step we used (119). According to (119) and (120), we have

$$\frac{1}{2}\left(\mathbb{E}_{(x_{n+1},y_{n+1})}\log\det(I+\frac{\widetilde{\Phi}_R\Lambda_R\widetilde{\Phi}_R^T}{\sigma^2})-\log\det(I+\frac{\Phi_R\Lambda_R\Phi_R^T}{\sigma^2})\right)$$

$$=\frac{1}{2\sigma^2}\left(\text{Tr}(I+\frac{\Lambda_R\Phi_R^T\Phi_R}{\sigma^2})^{-1}\Lambda_R\right)(1+o(1))$$

$$=\frac{1}{\sigma^2}\Theta(n^{\frac{(1-\alpha)(1-t)}{\alpha}})+\frac{1}{\sigma^2}\sum_{j=1}^{\infty}\Theta(n^{\frac{(1-\alpha)(1-t)}{\alpha}})\tilde{O}(n^{\frac{j(1-\alpha+2\tau-(1+2\tau)t)}{2\alpha}}(\log R)^{j/2}) \tag{121}$$

$$=\frac{1}{\sigma^2}\Theta(n^{\frac{(1-\alpha)(1-t)}{\alpha}})+\frac{1}{\sigma^2}\Theta(n^{\frac{(1-\alpha)(1-t)}{\alpha}})o(1)=\frac{1}{\sigma^2}\Theta(n^{\frac{(1-\alpha)(1-t)}{\alpha}})$$

$$=\frac{1}{2\sigma^2}\left(\text{Tr}(I+\frac{n}{\sigma^2}\Lambda_R)^{-1}\Lambda_R\right)(1+o(1)).$$

Using the Woodbury matrix identity, the second term in the right hand side (116) is given by

$$\frac{1}{2}\left(\mathbb{E}_{(x_{n+1},y_{n+1})}\text{Tr}(I-(I+\frac{\widetilde{\Phi}_R\Lambda_R\widetilde{\Phi}_R^T}{\sigma^2})^{-1}-\text{Tr}(I-(I+\frac{\Phi_R\Lambda_R\Phi_R^T}{\sigma^2})^{-1}\right)$$

$$=\frac{1}{2}\left(\mathbb{E}_{(x_{n+1},y_{n+1})}\text{Tr}(\frac{1}{\sigma^2}\widetilde{\Phi}_R(I+\frac{1}{\sigma^2}\Lambda_R\widetilde{\Phi}_R^T\widetilde{\Phi}_R)^{-1}\Lambda_R\widetilde{\Phi}_R^T-\text{Tr}(\frac{1}{\sigma^2}\Phi_R(I+\frac{1}{\sigma^2}\Lambda_R\Phi_R^T\Phi_R)^{-1}\Lambda_R\Phi_R^T\right)$$

$$=\frac{1}{2}\left(\mathbb{E}_{(x_{n+1},y_{n+1})}\text{Tr}(\frac{1}{\sigma^2}(I+\frac{1}{\sigma^2}\Lambda_R\widetilde{\Phi}_R^T\widetilde{\Phi}_R)^{-1}\Lambda_R\widetilde{\Phi}_R^T\widetilde{\Phi}_R-\text{Tr}(\frac{1}{\sigma^2}(I+\frac{1}{\sigma^2}\Lambda_R\Phi_R^T\Phi_R)^{-1}\Lambda_R\Phi_R^T\Phi_R\right)$$

$$=-\frac{1}{2}\left(\mathbb{E}_{(x_{n+1},y_{n+1})}\text{Tr}(I+\frac{1}{\sigma^2}\Lambda_R\widetilde{\Phi}_R^T\widetilde{\Phi}_R)^{-1}-\text{Tr}(I+\frac{1}{\sigma^2}\Lambda_R\Phi_R^T\Phi_R)^{-1}\right)$$

$$=-\frac{1}{2}\left(\mathbb{E}_{(x_{n+1},y_{n+1})}\text{Tr}(I+\frac{1}{\sigma^2}\Lambda_R\Phi_R^T\Phi_R+\frac{1}{\sigma^2}\Lambda_R\eta_R\eta_R^T)^{-1}-\text{Tr}(I+\frac{1}{\sigma^2}\Lambda_R\Phi_R^T\Phi_R)^{-1}\right)$$

$$=\frac{1}{2\sigma^2}\left(\mathbb{E}_{(x_{n+1},y_{n+1})}\text{Tr}\frac{(I+\frac{1}{\sigma^2}\Lambda_R\Phi_R^T\Phi_R)^{-1}\Lambda_R\eta_R\eta_R^T(I+\frac{1}{\sigma^2}\Lambda_R\Phi_R^T\Phi_R)^{-1}}{1+\frac{1}{\sigma^2}\eta_R^T(I+\frac{1}{\sigma^2}\Lambda_R\Phi_R^T\Phi_R)^{-1}\Lambda_R\eta_R}\right),$$

where the last equality uses the Sherman–Morrison formula. According to (118), we get

$$\frac{1}{2\sigma^2}\left(\mathbb{E}_{(x_{n+1},y_{n+1})}\text{Tr}\frac{(I+\frac{1}{\sigma^2}\Lambda_R\Phi_R^T\Phi_R)^{-1}\Lambda_R\eta_R\eta_R^T(I+\frac{1}{\sigma^2}\Lambda_R\Phi_R^T\Phi_R)^{-1}}{1+\frac{1}{\sigma^2}\eta_R^T(I+\frac{1}{\sigma^2}\Lambda_R\Phi_R^T\Phi_R)^{-1}\Lambda_R\eta_R}\right)$$

$$=\frac{1}{2\sigma^2}\left(\mathbb{E}_{(x_{n+1},y_{n+1})}\text{Tr}(I+\frac{1}{\sigma^2}\Lambda_R\Phi_R^T\Phi_R)^{-1}\Lambda_R\eta_R\eta_R^T(I+\frac{1}{\sigma^2}\Lambda_R\Phi_R^T\Phi_R)^{-1}(1+o(1))\right)$$

$$=\frac{1+o(1)}{2\sigma^2}\text{Tr}(I+\frac{1}{\sigma^2}\Lambda_R\Phi_R^T\Phi_R)^{-1}\Lambda_R(I+\frac{1}{\sigma^2}\Lambda_R\Phi_R^T\Phi_R)^{-1}$$

$$=\frac{1+o(1)}{2\sigma^2}\text{Tr}\Lambda_R^{1/2}(I+\frac{1}{\sigma^2}\Lambda_R^{1/2}\Phi_R^T\Phi_R\Lambda_R^{1/2})^{-1}\Lambda_R^{1/2}(I+\frac{1}{\sigma^2}\Lambda_R\Phi_R^T\Phi_R)^{-1}$$

$$=\frac{1+o(1)}{2\sigma^2}\text{Tr}(I+\frac{1}{\sigma^2}\Lambda_R^{1/2}\Phi_R^T\Phi_R\Lambda_R^{1/2})^{-1}\Lambda_R^{1/2}(I+\frac{1}{\sigma^2}\Lambda_R\Phi_R^T\Phi_R)^{-1}\Lambda_R^{1/2}$$

$$=\frac{1+o(1)}{2\sigma^2}\text{Tr}(I+\frac{1}{\sigma^2}\Lambda_R^{1/2}\Phi_R^T\Phi_R\Lambda_R^{1/2})^{-1}\Lambda_R(I+\frac{1}{\sigma^2}\Lambda_R^{1/2}\Phi_R^T\Phi_R\Lambda_R^{1/2})^{-1}$$

$$=\frac{1+o(1)}{2\sigma^2}\|\Lambda_R^{1/2}(I+\frac{1}{\sigma^2}\Lambda_R^{1/2}\Phi_R^T\Phi_R\Lambda_R^{1/2})^{-1}\|_F^2$$

$$=\frac{1+o(1)}{2\sigma^2}\|\Lambda_R^{1/2}(I+\frac{n}{\sigma^2}\Lambda_R)^{-1/2}(I+\frac{1}{\sigma^2}A)^{-1}(I+\frac{n}{\sigma^2}\Lambda_R)^{-1/2}\|_F^2,$$

where in the penultimate equality we use $\text{Tr}(BB^T)=\|B\|_F^2$, $\|B\|_F$ is the Frobenius norm of $A$, and in the last equality we use the definition of $A$ (117). Then we have

$$\frac{1+o(1)}{2\sigma^2}\|\Lambda_R^{1/2}(I+\frac{n}{\sigma^2}\Lambda_R)^{-1/2}(I+\frac{1}{\sigma^2}A)^{-1}(I+\frac{n}{\sigma^2}\Lambda_R)^{-1/2}\|_F^2$$

$$=\frac{1+o(1)}{2\sigma^2}\|\Lambda_R^{1/2}(I+\frac{n}{\sigma^2}\Lambda_R)^{-1/2}(I+\sum_{j=1}^{\infty}(-1)^j\frac{1}{\sigma^{2j}}A^j)(I+\frac{n}{\sigma^2}\Lambda_R)^{-1/2}\|_F^2$$

$$=\frac{1+o(1)}{2\sigma^2}\|\Lambda_R^{1/2}(I+\frac{n}{\sigma^2}\Lambda_R)^{-1}+\sum_{j=1}^{\infty}(-1)^j\frac{1}{\sigma^{2j}}\Lambda_R^{1/2}(I+\frac{n}{\sigma^2}\Lambda_R)^{-1/2}A^j(I+\frac{n}{\sigma^2}\Lambda_R)^{-1/2}\|_F^2.$$

$$\tag{122}$$

By Lemma 15, we have

$$\|\Lambda_R^{1/2}(I+\frac{n}{\sigma^2}\Lambda_R)^{-1}\|_F\leq\sqrt{\sum_{p=1}^{R}\frac{\overline{C_\lambda}p^{-\alpha}}{(1+n\underline{C_\lambda}p^{-\alpha}/\sigma^2)^2}}=\Theta(n^{\frac{(1-\alpha)(1-t)}{2\alpha}})$$

$$\|\Lambda_R^{1/2}(I+\frac{n}{\sigma^2}\Lambda_R)^{-1}\|_F\geq\sqrt{\sum_{p=1}^{R}\frac{\underline{C_\lambda}p^{-\alpha}}{(1+n\overline{C_\lambda}p^{-\alpha}/\sigma^2)^2}}=\Theta(n^{\frac{(1-\alpha)(1-t)}{2\alpha}}).$$

Overall, we have

$$\|\Lambda_R^{1/2}(I+\frac{n}{\sigma^2}\Lambda_R)^{-1}\|_F=\Theta(n^{\frac{(1-\alpha)(1-t)}{2\alpha}}). \tag{123}$$

Since $\|\frac{1}{\sigma^2}A\|_2=O(\sqrt{\log\frac{R}{\delta}}n^{\frac{1-\alpha+2\tau}{2\alpha}-\frac{(1+2\tau)t}{2\alpha}})=o(1)$, we have

$$\begin{aligned}
&\|\frac{1}{\sigma^{2j}}\Lambda_R^{1/2}(I+\frac{n}{\sigma^2}\Lambda_R)^{-1/2}A^j(I+\frac{n}{\sigma^2}\Lambda_R)^{-1/2}\|_F\\
&\leq\|\Lambda_R^{1/2}(I+\frac{n}{\sigma^2}\Lambda_R)^{-1/2}\|_F\|\frac{1}{\sigma^2}A\|_2^j\|(I+\frac{n}{\sigma^2}\Lambda_R)^{-1/2}\|_2\\
&=O(n^{\frac{(1-\alpha)(1-t)}{2\alpha}})\tilde{O}(n^{\frac{j(1-\alpha+2\tau-(1+2\tau)t)}{2\alpha}}(\log R)^{j/2}),
\end{aligned} \tag{124}$$

where in the first inequality we use the fact that $\|AB\|_F\leq\|A\|_F\|B\|_2$ when $B$ is symmetric. By Lemma 15, we have

$$\begin{aligned}
&\frac{1}{\sigma^{2j}}\Big|\mathrm{Tr}\Lambda_R^{1/2}(I+\frac{n}{\sigma^2}\Lambda_R)^{-1}\Lambda_R^{1/2}(I+\frac{n}{\sigma^2}\Lambda_R)^{-1/2}A^j(I+\frac{n}{\sigma^2}\Lambda_R)^{-1/2}\Big|\\
&=\Big|\sum_{p=1}^R\frac{\lambda_p((\frac{1}{\sigma^2}A)^j)_{p,p}}{(1+n\lambda_p/\sigma^2)^2}\Big|\leq\sum_{p=1}^R\frac{\lambda_p\|\frac{1}{\sigma^2}A\|_2^j}{(1+n\lambda_p/\sigma^2)^2}=\Theta(n^{\frac{(1-\alpha)(1-t)}{\alpha}})\tilde{O}(n^{\frac{j(1-\alpha+2\tau-(1+2\tau)t)}{2\alpha}}(\log R)^{j/2}),
\end{aligned} \tag{125}$$

According to (123), (124) and (125), we have

$$\begin{aligned}
&\frac{1}{2}\left(\mathbb{E}_{(x_{n+1},y_{n+1})}\mathrm{Tr}(I-(I+\frac{\widetilde{\Phi}_R\Lambda_R\widetilde{\Phi}_R^T}{\sigma^2})^{-1}-\mathrm{Tr}(I-(I+\frac{\Phi_R\Lambda_R\Phi_R^T}{\sigma^2})^{-1})\right)\\
&=\frac{1+o(1)}{2\sigma^2}\mathrm{Tr}(I+\frac{1}{\sigma^2}\Lambda_R\Phi_R^T\Phi_R)^{-1}\Lambda_R(I+\frac{1}{\sigma^2}\Lambda_R\Phi_R^T\Phi_R)^{-1}\\
&=\frac{1+o(1)}{2\sigma^2}\|\Lambda_R^{1/2}(I+\frac{n}{\sigma^2}\Lambda_R)^{-1}+\sum_{j=1}^{\infty}(-1)^j\frac{1}{\sigma^{2j}}\Lambda_R^{1/2}(I+\frac{n}{\sigma^2}\Lambda_R)^{-1/2}A^j(I+\frac{n}{\sigma^2}\Lambda_R)^{-1/2}\|_F^2\\
&=\frac{1+o(1)}{2\sigma^2}\left(\|\Lambda_R^{1/2}(I+\frac{n}{\sigma^2}\Lambda_R)^{-1}\|_F^2+\sum_{j=1}^{\infty}\left\|\frac{1}{\sigma^{2j}}\Lambda_R^{1/2}(I+\frac{n}{\sigma^2}\Lambda_R)^{-1/2}A^j(I+\frac{n}{\sigma^2}\Lambda_R)^{-1/2}\right\|_F^2\right.\\
&\quad\left.+2\mathrm{Tr}\Lambda_R^{1/2}(I+\frac{n}{\sigma^2}\Lambda_R)^{-1}\sum_{j=1}^{\infty}(-1)^j\frac{1}{\sigma^{2j}}\Lambda_R^{1/2}(I+\frac{n}{\sigma^2}\Lambda_R)^{-1/2}A^j(I+\frac{n}{\sigma^2}\Lambda_R)^{-1/2}\right)\\
&=\frac{1+o(1)}{2\sigma^2}\left(\Theta(n^{\frac{(1-\alpha)(1-t)}{\alpha}})+\sum_{j=1}^{\infty}\frac{1}{\sigma^{2j}}O(n^{\frac{(1-\alpha)(1-t)}{\alpha}})\tilde{O}(n^{\frac{j(1-\alpha+2\tau-(1+2\tau)t)}{2\alpha}}(\log R)^{j/2})\right.\\
&\quad\left.+2\sum_{j=1}^{\infty}\frac{1}{\sigma^{2j}}\Theta(n^{\frac{(1-\alpha)(1-t)}{\alpha}})\tilde{O}(n^{\frac{j(1-\alpha+2\tau-(1+2\tau)t)}{2\alpha}}(\log R)^{j/2})\right)\\
&=\frac{1}{\sigma^2}\Theta(n^{\frac{(1-\alpha)(1-t)}{\alpha}})=\frac{1+o(1)}{2\sigma^2}\|\Lambda_R^{1/2}(I+\frac{n}{\sigma^2}\Lambda_R)^{-1}\|_F^2.
\end{aligned} \tag{126}$$

Combining (121) and (126) we get that $G_{1,R}(D_n)=\frac{1+o(1)}{2\sigma^2}(\mathrm{Tr}(I+\frac{n}{\sigma^2}\Lambda_R)^{-1}\Lambda_R+\|\Lambda_R^{1/2}(I+\frac{n}{\sigma^2}\Lambda_R)^{-1}\|_F^2)=\frac{1}{\sigma^2}\Theta(n^{\frac{(1-\alpha)(1-t)}{\alpha}})$. From (115) we have that $G_1(D_n)\leq G_{1,R}(D_n)+|G_1(D_n)-G_{1,R}(D_n)|=\frac{1}{\sigma^2}\Theta(n^{\frac{(1-\alpha)(1-t)}{\alpha}})+O(n\frac{1}{\sigma^2}R^{1-\alpha})$. Choosing $R=n^{(\frac{2\alpha-1}{\alpha(\alpha-1)}+1)(1-t)}$ we conclude the proof. $\square$

**Lemma 40.** *Assume $\sigma^2=\Theta(n^t)$ where $1-\frac{\alpha}{1+2\tau}<t<1$. Let $S=n^D$. Assume that $\|\xi\|_2=1$. When $n$ is sufficiently large, with probability of at least $1-2\delta$ we have*

$$\|(I+\frac{1}{\sigma^2}\Phi_S\Lambda_S\Phi_S^T)^{-1}\Phi_S\Lambda_S\xi\|_2=O(\sqrt{(\frac{1}{\delta}+1)n}\cdot n^{-(1-t)}). \tag{127}$$

*Proof of Lemma 40.* Using the Woodbury matrix identity, we have that

$$
\begin{aligned}
((I+\frac{1}{\sigma^2}\Phi_S\Lambda_S\Phi_S^T)^{-1}\Phi_S\Lambda_S\xi &= \left[I-\Phi_S(\sigma^2 I+\Lambda_S\Phi_S^T\Phi_S)^{-1}\Lambda_S\Phi_S^T\right]\Phi_S\Lambda_S\xi \\
&= \Phi_S\Lambda_S\xi-\Phi_S(\sigma^2 I+\Lambda_S\Phi_S^T\Phi_S)^{-1}\Lambda_S\Phi_S^T\Phi_S\Lambda_S\xi \\
&= \Phi_S(I+\frac{1}{\sigma^2}\Lambda_S\Phi_S^T\Phi_S)^{-1}\Lambda_S\xi.
\end{aligned}
\tag{128}
$$

Let $A=(I+\frac{n}{\sigma^2}\Lambda_S)^{-\gamma/2}\Lambda_S^{\gamma/2}(\Phi_S^T\Phi_S-nI)\Lambda_S^{\gamma/2}(I+\frac{n}{\sigma^2}\Lambda_S)^{-\gamma/2}$, where $\gamma > \frac{1+\alpha+2\tau-(1+2\tau+2\alpha)t}{2\alpha(1-t)}$.
By Corollary 22, with probability of at least $1-\delta$, we have $\|\frac{1}{\sigma^2}A\|_2=\tilde{O}(n^{\frac{1+\alpha+2\tau-(1+2\tau+2\alpha)t}{2\alpha}-\gamma(1-t)})$.
When $n$ is sufficiently large, $\|\frac{1}{\sigma^2}A\|_2$ is less than 1. By Lemma 27, we have

$$
\begin{aligned}
&(I+\frac{1}{\sigma^2}\Lambda_S\Phi_S^T\Phi_S)^{-1} \\
&=(I+\frac{n}{\sigma^2}\Lambda_S)^{-1}+\sum_{j=1}^{\infty}(-1)^j\left(\frac{1}{\sigma^2}(I+\frac{n}{\sigma^2}\Lambda_S)^{-1}\Lambda_S(\Phi_S^T\Phi_S-nI)\right)^j(I+\frac{n}{\sigma^2}\Lambda_S)^{-1}.
\end{aligned}
$$

Then we have

$$
\begin{aligned}
&\|(I+\frac{1}{\sigma^2}\Lambda_S\Phi_S^T\Phi_S)^{-1}\Lambda_S\xi\|_2 \\
&=\left\|\left((I+\frac{n}{\sigma^2}\Lambda_S)^{-1}+\sum_{j=1}^{\infty}(-1)^j\left(\frac{1}{\sigma^2}(I+\frac{n}{\sigma^2}\Lambda_S)^{-1}\Lambda_S(\Phi_S^T\Phi_S-nI)\right)^j(I+\frac{n}{\sigma^2}\Lambda_S)^{-1}\right)\Lambda_S\xi\right\|_2 \\
&\leq\left(\|(I+\frac{n}{\sigma^2}\Lambda_S)^{-1}\Lambda_S\xi\|_2+\sum_{j=1}^{\infty}\left\|\left(\frac{1}{\sigma^2}(I+\frac{n}{\sigma^2}\Lambda_S)^{-1}\Lambda_S(\Phi_S^T\Phi_S-nI)\right)^j(I+\frac{n}{\sigma^2}\Lambda_S)^{-1}\Lambda_S\xi\right\|_2\right).
\end{aligned}
\tag{129}
$$

For the first term in the right hand side of the last equation, we have

$$
\|(I+\frac{n}{\sigma^2}\Lambda_S)^{-1}\Lambda_S\xi\|_2\leq\|(I+\frac{n}{\sigma^2}\Lambda_S)^{-1}\Lambda_S\|_2\|\xi\|_2\leq\frac{\sigma^2}{n}=O(n^{-(1-t)}).
\tag{130}
$$

Using the fact that $\|\frac{1}{\sigma^2}A\|_2=\tilde{O}(n^{\frac{1+\alpha+2\tau-(1+2\tau+2\alpha)t}{2\alpha}-\gamma(1-t)})$ and $\|(I+\frac{n}{\sigma^2}\Lambda_S)^{-1}\Lambda_S\|_2\leq n^{-1}$, we have

$$
\begin{aligned}
&\left\|\left(\frac{1}{\sigma^2}(I+\frac{n}{\sigma^2}\Lambda_S)^{-1}\Lambda_S(\Phi_S^T\Phi_S-nI)\right)^j(I+\frac{n}{\sigma^2}\Lambda_S)^{-1}\Lambda_S\xi\right\|_2 \\
&=\frac{1}{\sigma^{2j}}\left\|(I+\frac{n}{\sigma^2}\Lambda_S)^{-1+\frac{\gamma}{2}}\Lambda_S^{1-\frac{\gamma}{2}}\left(A(I+\frac{n}{\sigma^2}\Lambda_S)^{-1+\gamma}\Lambda_S^{1-\gamma}\right)^{j-1}A(I+\frac{n}{\sigma^2}\Lambda_S)^{-1+\frac{\gamma}{2}}\Lambda_S^{-\frac{\gamma}{2}}\Lambda_S\xi\right\|_2 \\
&\leq n^{(1-t)(-1+\frac{\gamma}{2}+(-1+\gamma)(j-1))}\tilde{O}(n^{\frac{j(1+\alpha+2\tau-(1+2\tau+2\alpha)t)}{2\alpha}-j\gamma(1-t)})\|(I+\frac{n}{\sigma^2}\Lambda_S)^{-1+\frac{\gamma}{2}}\Lambda_S^{1-\frac{\gamma}{2}}\xi\|_2 \\
&=\tilde{O}(n^{-\frac{\gamma}{2}(1-t)+\frac{(1-\alpha+2\tau-(1+2\tau)t)j}{2\alpha}})\|(I+\frac{n}{\sigma^2}\Lambda_S)^{-1+\frac{\gamma}{2}}\Lambda_S^{1-\frac{\gamma}{2}}\|_2\|\xi\|_2 \\
&=\tilde{O}(n^{-\frac{\gamma}{2}(1-t)+\frac{(1-\alpha+2\tau-(1+2\tau)t)j}{2\alpha}})O(n^{(-1+\gamma/2)(1-t)}) \\
&=\tilde{O}(n^{-(1-t)+\frac{(1-\alpha+2\tau-(1+2\tau)t)j}{2\alpha}}).
\end{aligned}
\tag{131}
$$

Using (129), (130) and (131), we have

$$
\begin{aligned}
&\|(I+\frac{1}{\sigma^2}\Lambda_S\Phi_S^T\Phi_S)^{-1}\Lambda_S\xi\|_2 \\
&=\left(\tilde{O}(n^{-(1-t)})+\sum_{j=1}^{\infty}\tilde{O}(n^{-1+\frac{(1-\alpha+2\tau-(1+2\tau)t)j}{2\alpha}})\right) \\
&=\left(\tilde{O}(n^{-(1-t)})+\tilde{O}(n^{-1+\frac{1-\alpha+2\tau-(1+2\tau)t}{2\alpha}})\right) \\
&=\tilde{O}(n^{-(1-t)}).
\end{aligned}
\tag{132}
$$

By Corollary 20, with probability of at least $1-\delta$, we have

$$\|\Phi_S(I+\tfrac{1}{\sigma^2}\Lambda_S\Phi_S^T\Phi_S)^{-1}\Lambda_S\xi\|_2 = \tilde{O}(\sqrt{(\tfrac{1}{\delta}+1)n}\|(I+\tfrac{1}{\sigma^2}\Lambda_S\Phi_S^T\Phi_S)^{-1}\Lambda_S\xi\|_2)$$

$$= \tilde{O}(\sqrt{(\tfrac{1}{\delta}+1)n}\cdot n^{-(1-t)}).$$

From (128) we get $\|(I+\tfrac{1}{\sigma^2}\Phi_S\Lambda_S\Phi_S^T)^{-1}f_S(\mathbf{x})\|_2 = \tilde{O}(\sqrt{(\tfrac{1}{\delta}+1)n}\cdot n^{-(1-t)})$. This concludes the proof. $\qquad\square$

**Lemma 41.** *Assume $\sigma^2 = \Theta(n^t)$ where $1 - \frac{\alpha}{1+2\tau} < t < 1$. Let $\delta = n^{-q}$ where $0 \le q < \frac{[\alpha-(1+2\tau)(1-t)](2\beta-1)}{4\alpha^2}$. Under Assumptions 4, 5 and 6, assume that $\mu_0 = 0$. Let $R = n^{(\frac{1}{\alpha}+\kappa)(1-t)}$ where $0 < \kappa < \frac{\alpha-1-2\tau+(1+2\tau)t}{2\alpha^2(1-t)}$. Then with probability of at least $1-6\delta$ over sample inputs $(x_i)_{i=1}^n$, we have $G_2(D_n) = \frac{(1+o(1))}{2\sigma^2}\|(I+\frac{n}{\sigma^2}\Lambda_R)^{-1}\boldsymbol{\mu}_R\|_2^2 = \frac{1}{\sigma^2}\Theta(n^{\max\{-2(1-t),\frac{(1-2\beta)(1-t)}{\alpha}\}}\log^{k/2}n)$, where*
$$k = \begin{cases} 0, & 2\alpha \ne 2\beta-1, \\ 1, & 2\alpha = 2\beta-1. \end{cases}$$

*Proof of Lemma 41.* Let $S = n^D$. Let $G_{2,S}(D_n) = \mathbb{E}_{(x_{n+1},y_{n+1})}(T_{2,S}(D_{n+1}) - T_{2,S}(D_n))$. By Lemma 33, when $S > n^{\max\{1,\frac{-t}{(\alpha-1-2\tau)}\}}$ with probability of at least $1-3\delta$ we have that

$$|G_2(D_n) - G_{2,S}(D_n)| = |\mathbb{E}_{(x_{n+1},y_{n+1})}[T_2(D_{n+1}) - T_{2,S}(D_{n+1})] - [T_2(D_n) - T_{2,S}(D_n)]|$$

$$= \left|\mathbb{E}_{(x_{n+1},y_{n+1})}\tilde{O}\left((\tfrac{1}{\delta}+1)\tfrac{1}{\sigma^2}(n+1)S^{\max\{1/2-\beta,1-\alpha+2\tau\}}\right) - \tilde{O}\left((\tfrac{1}{\delta}+1)\tfrac{1}{\sigma^2}nS^{\max\{1/2-\beta,1-\alpha+2\tau\}}\right)\right|$$

$$= \tilde{O}\left((\tfrac{1}{\delta}+1)\tfrac{1}{\sigma^2}nS^{\max\{1/2-\beta,1-\alpha+2\tau\}}\right) \tag{133}$$

$$\tag{134}$$

Let $\Lambda_{1:S} = \mathrm{diag}\{\lambda_1,\ldots,\lambda_S\}$, $\Phi_{1:S} = (\phi_1(\mathbf{x}),\phi_1(\mathbf{x}),\ldots,\phi_S(\mathbf{x}))$ and $\boldsymbol{\mu}_{1:S} = (\mu_1,\ldots,\mu_S)$. Since $\mu_0 = 0$, we have $T_{2,S}(D_n) = \frac{1}{2\sigma^2}\boldsymbol{\mu}_{1:S}^T\Phi_{1:S}^T(I+\frac{1}{\sigma^2}\Phi_{1:S}\Lambda_{1:S}\Phi_{1:S}^T)^{-1}\Phi_{1:S}\boldsymbol{\mu}_{1:S}$. Define $\eta_{1:S} = (\phi_1(x_{n+1}),\ldots,\phi_S(x_{n+1}))^T$ and $\widetilde{\Phi}_{1:S} = (\Phi_{1:S}^T,\eta_{1:S})^T$. In the proof of Lemma 34, we showed that

$$T_{2,S}(D_n) = \frac{1}{2\sigma^2}\boldsymbol{\mu}_{1:S}^T\Phi_{1:S}^T(I+\frac{1}{\sigma^2}\Phi_{1:S}\Lambda_{1:S}\Phi_{1:S}^T)^{-1}\Phi_{1:S}\boldsymbol{\mu}_{1:S}$$

$$= \frac{1}{2}\boldsymbol{\mu}_{1:S}^T\Lambda_{1:S}^{-1}\boldsymbol{\mu}_{1:S} - \frac{1}{2}\boldsymbol{\mu}_{1:S}^T\Lambda_{1:S}^{-1}(I+\frac{1}{\sigma^2}\Lambda_{1:S}\Phi_{1:S}^T\Phi_{1:S})^{-1}\boldsymbol{\mu}_{1:S}.$$

We have
$$G_{2,S}(D_n) = \mathbb{E}_{(x_{n+1},y_{n+1})}(T_{2,S}(D_{n+1}) - T_{2,S}(D_n))$$

$$= \mathbb{E}_{(x_{n+1},y_{n+1})}\left(\frac{1}{2}\boldsymbol{\mu}_{1:S}^T\Lambda_{1:S}^{-1}\boldsymbol{\mu}_{1:S} - \frac{1}{2}\boldsymbol{\mu}_{1:S}^T\Lambda_{1:S}^{-1}(I+\frac{1}{\sigma^2}\Lambda_{1:S}\widetilde{\Phi}_S^T\widetilde{\Phi}_S)^{-1}\boldsymbol{\mu}_{1:S}\right)$$

$$\quad - \left(\frac{1}{2}\boldsymbol{\mu}_{1:S}^T\Lambda_{1:S}^{-1}\boldsymbol{\mu}_{1:S} - \frac{1}{2}\boldsymbol{\mu}_{1:S}^T\Lambda_{1:S}^{-1}(I+\frac{1}{\sigma^2}\Lambda_{1:S}\Phi_{1:S}^T\Phi_{1:S})^{-1}\boldsymbol{\mu}_{1:S}\right)$$

$$= \mathbb{E}_{(x_{n+1},y_{n+1})}\left(\frac{1}{2}\boldsymbol{\mu}_{1:S}^T\Lambda_{1:S}^{-1}(I+\frac{1}{\sigma^2}\Lambda_{1:S}\Phi_{1:S}^T\Phi_{1:S})^{-1}\boldsymbol{\mu}_{1:S} - \frac{1}{2}\boldsymbol{\mu}_{1:S}^T\Lambda_{1:S}^{-1}(I+\frac{1}{\sigma^2}\Lambda_{1:S}\widetilde{\Phi}_S^T\widetilde{\Phi}_S)^{-1}\boldsymbol{\mu}_{1:S}\right)$$

$$= \mathbb{E}_{(x_{n+1},y_{n+1})}\left(\frac{1}{2\sigma^2}\boldsymbol{\mu}_{1:S}^T\Lambda_{1:S}^{-1}\frac{(I+\frac{1}{\sigma^2}\Lambda_{1:S}\Phi_{1:S}^T\Phi_{1:S})^{-1}\Lambda_{1:S}\eta_{1:S}\eta_{1:S}^T(I+\frac{1}{\sigma^2}\Lambda_{1:S}\Phi_{1:S}^T\Phi_{1:S})^{-1}}{1+\frac{1}{\sigma^2}\eta_{1:S}^T(I+\frac{1}{\sigma^2}\Lambda_{1:S}\Phi_{1:S}^T\Phi_{1:S})^{-1}\Lambda_{1:S}\eta_{1:S}}\boldsymbol{\mu}_{1:S}\right)$$

$$= \mathbb{E}_{(x_{n+1},y_{n+1})}\left(\frac{1}{2\sigma^2}\frac{\boldsymbol{\mu}_{1:S}^T(I+\frac{1}{\sigma^2}\Phi_{1:S}^T\Phi_{1:S}\Lambda_{1:S})^{-1}\eta_{1:S}\eta_{1:S}^T(I+\frac{1}{\sigma^2}\Lambda_{1:S}\Phi_{1:S}^T\Phi_{1:S})^{-1}\boldsymbol{\mu}_{1:S}}{1+\frac{1}{\sigma^2}\eta_{1:S}^T(I+\frac{1}{\sigma^2}\Lambda_{1:S}\Phi_{1:S}^T\Phi_{1:S})^{-1}\Lambda_{1:S}\eta_{1:S}}\right)$$

$$= \mathbb{E}_{(x_{n+1},y_{n+1})}\left(\frac{1+o(1)}{2\sigma^2}\boldsymbol{\mu}_{1:S}^T(I+\frac{1}{\sigma^2}\Phi_{1:S}^T\Phi_{1:S}\Lambda_{1:S})^{-1}\eta_{1:S}\eta_{1:S}^T(I+\frac{1}{\sigma^2}\Lambda_{1:S}\Phi_{1:S}^T\Phi_{1:S})^{-1}\boldsymbol{\mu}_{1:S}\right)$$

$$= \frac{1+o(1)}{2\sigma^2}\boldsymbol{\mu}_{1:S}^T(I+\frac{1}{\sigma^2}\Phi_{1:S}^T\Phi_{1:S}\Lambda_{1:S})^{-1}(I+\frac{1}{\sigma^2}\Lambda_{1:S}\Phi_{1:S}^T\Phi_{1:S})^{-1}\boldsymbol{\mu}_{1:S}$$

$$= \frac{1+o(1)}{2\sigma^2}\|(I+\frac{1}{\sigma^2}\Lambda_{1:S}\Phi_{1:S}^T\Phi_{1:S})^{-1}\boldsymbol{\mu}_{1:S}\|_2^2, \tag{135}$$

where in the fourth to last equality we used the Sherman–Morrison formula, in the third inequality we used (118), and in the last equality we used the fact that $\mathbb{E}_{(x_{n+1},y_{n+1})}\eta_{1:S}\eta_{1:S}^T = I$.

Let $\hat{\boldsymbol{\mu}}_{1:R} = (\mu_1,...,\mu_R,0,...,0) \in \mathbb{R}^S$. Then we have

$$\|(I+\frac{1}{\sigma^2}\Lambda_{1:S}\Phi_{1:S}^T\Phi_{1:S})^{-1}\boldsymbol{\mu}_{1:S}\|_2 \leq \|(I+\frac{1}{\sigma^2}\Lambda_{1:S}\Phi_{1:S}^T\Phi_{1:S})^{-1}\hat{\boldsymbol{\mu}}_{1:R}\|_2 + \|(I+\frac{1}{\sigma^2}\Lambda_{1:S}\Phi_{1:S}^T\Phi_{1:S})^{-1}(\boldsymbol{\mu}_{1:S}-\hat{\boldsymbol{\mu}}_{1:R})\|_2,$$
$$\|(I+\frac{1}{\sigma^2}\Lambda_{1:S}\Phi_{1:S}^T\Phi_{1:S})^{-1}\boldsymbol{\mu}_{1:S}\|_2 \geq \|(I+\frac{1}{\sigma^2}\Lambda_{1:S}\Phi_{1:S}^T\Phi_{1:S})^{-1}\hat{\boldsymbol{\mu}}_{1:R}\|_2 - \|(I+\frac{1}{\sigma^2}\Lambda_{1:S}\Phi_{1:S}^T\Phi_{1:S})^{-1}(\boldsymbol{\mu}_{1:S}-\hat{\boldsymbol{\mu}}_{1:R})\|_2.$$
$$(136)$$

Let $R = n^{(\frac{1}{\alpha}+\kappa)(1-t)}$ where $0 < \kappa < \frac{\alpha-1-2\tau+(1+2\tau)t}{2\alpha^2(1-t)}$. In Lemma 29, (62), we showed that with probability of at least $1-\delta$,

$$\|(I+\frac{1}{\sigma^2}\Lambda_{1:R}\Phi_{1:R}^T\Phi_{1:R})^{-1}\boldsymbol{\mu}_{1:R}\|_2 = \Theta(n^{(1-t)\max\{-1,\frac{1-2\beta}{2\alpha}\}}\log^{k/2}n)$$
$$= (1+o(1))\|(I+\frac{n}{\sigma^2}\Lambda_{1:R})^{-1}\boldsymbol{\mu}_{1:R}\|_2, \tag{137}$$

where $k = \begin{cases} 0, & 2\alpha \neq 2\beta-1, \\ 1, & 2\alpha = 2\beta-1. \end{cases}$. The same proof holds if we replace $\Phi_{1:R}$ with $\Phi_{1:S}$, $\Lambda_{1:R}$ with $\Lambda_{1:S}$, and $\boldsymbol{\mu}_{1:R}$ with $\hat{\boldsymbol{\mu}}_{1:R}$. We have

$$\|(I+\frac{1}{\sigma^2}\Lambda_{1:S}\Phi_{1:S}^T\Phi_{1:S})^{-1}\hat{\boldsymbol{\mu}}_{1:R}\|_2 = \Theta(n^{(1-t)\max\{-1,\frac{1-2\beta}{2\alpha}\}}\log^{k/2}n)$$
$$= (1+o(1))\|(I+\frac{n}{\sigma^2}\Lambda_{1:S})^{-1}\hat{\boldsymbol{\mu}}_{1:R}\|_2. \tag{138}$$

Next we bound $\|(I + \frac{1}{\sigma^2}\Lambda_{1:S}\Phi_{1:S}^T\Phi_{1:S})^{-1}(\boldsymbol{\mu}_{1:S} - \hat{\boldsymbol{\mu}}_{1:R})\|_2$. By Assumption 5, we have that $\|\boldsymbol{\mu}_{1:S}-\hat{\boldsymbol{\mu}}_{1:R}\|_2 = O(R^{\frac{1-2\beta}{2}})$. For any $\xi \in \mathbb{R}^S$ and $\|\xi\|_2 = 1$, using the Woodbury matrix identity, with probability of at least $1-2\delta$ we have

$$|\xi^T(I+\frac{1}{\sigma^2}\Lambda_{1:S}\Phi_{1:S}^T\Phi_{1:S})^{-1}(\boldsymbol{\mu}_{1:S}-\hat{\boldsymbol{\mu}}_{1:R})|$$
$$= |\xi^T\left(I - \frac{1}{\sigma^2}\Lambda_{1:S}\Phi_{1:S}^T(I+\frac{1}{\sigma^2}\Phi_{1:S}\Lambda_{1:S}\Phi_{1:S}^T)^{-1}\Phi_{1:S}\right)(\boldsymbol{\mu}_{1:S}-\hat{\boldsymbol{\mu}}_{1:R})|$$
$$= |\xi^T(\boldsymbol{\mu}_{1:S}-\hat{\boldsymbol{\mu}}_{1:R}) - \frac{1}{\sigma^2}\xi^T\Lambda_{1:S}\Phi_{1:S}^T(I+\frac{1}{\sigma^2}\Phi_{1:S}\Lambda_{1:S}\Phi_{1:S}^T)^{-1}\Phi_{1:S}(\boldsymbol{\mu}_{1:S}-\hat{\boldsymbol{\mu}}_{1:R})|$$
$$\leq \|\xi\|_2\|\boldsymbol{\mu}_{1:S}-\hat{\boldsymbol{\mu}}_{1:R}\|_2 + \frac{1}{\sigma^2}|\xi^T\Lambda_{1:S}\Phi_{1:S}^T(I+\frac{1}{\sigma^2}\Phi_{1:S}\Lambda_{1:S}\Phi_{1:S}^T)^{-1}\Phi_{1:S}(\boldsymbol{\mu}_{1:S}-\hat{\boldsymbol{\mu}}_{1:R})|$$
$$\leq O(R^{\frac{1-2\beta}{2}}) + \frac{1}{\sigma^2}\|(I+\frac{1}{\sigma^2}\Phi_{1:S}\Lambda_{1:S}\Phi_{1:S}^T)^{-1}\Phi_{1:S}\Lambda_{1:S}\xi\|_2\|\Phi_{1:S}(\boldsymbol{\mu}_{1:S}-\hat{\boldsymbol{\mu}}_{1:R})\|_2$$
$$= O(R^{\frac{1-2\beta}{2}}) + \frac{1}{\sigma^2}O(\sqrt{(\frac{1}{\delta}+1)n} \cdot n^{-(1-t)})O(\sqrt{(\frac{1}{\delta}+1)n}R^{\frac{1-2\beta}{2}})$$
$$= O((\frac{1}{\delta}+1)R^{\frac{1-2\beta}{2}}),$$

where in the second to last step we used Corollary 20 to show $\|\Phi_{1:S}(\boldsymbol{\mu}_{1:S} - \hat{\boldsymbol{\mu}}_{1:R})\|_2 = O(\sqrt{(\frac{1}{\delta}+1)n}R^{\frac{1-2\beta}{2}})$ with probability of at least $1 - \delta$, and Lemma 40 to show that $\|(I + \frac{1}{\sigma^2}\Phi_{1:S}\Lambda_{1:S}\Phi_{1:S}^T)^{-1}\Phi_{1:S}\Lambda_{1:S}\xi\|_2 = O(\sqrt{(\frac{1}{\delta}+1)n} \cdot n^{-1})$ with probability of at least $1-\delta$. Since $R = n^{(\frac{1}{\alpha}+\kappa)(1-t)}$, we have

$$|\xi^T(I+\frac{1}{\sigma^2}\Lambda_{1:S}\Phi_{1:S}^T\Phi_{1:S})^{-1}(\boldsymbol{\mu}_{1:S}-\hat{\boldsymbol{\mu}}_{1:R})| = O((\frac{1}{\delta}+1)n^{\frac{(1-2\beta)(1-t)}{2\alpha}+\frac{(1-2\beta)(1-t)\kappa}{2}}).$$

Since $\xi$ is arbitrary, we have $\|(I + \frac{1}{\sigma^2}\Lambda_{1:S}\Phi_{1:S}^T\Phi_{1:S})^{-1}(\boldsymbol{\mu}_{1:S} - \hat{\boldsymbol{\mu}}_{1:R})\|_2 = O((\frac{1}{\delta} + 1)n^{\frac{(1-2\beta)(1-t)}{2\alpha}+\frac{(1-2\beta)(1-t)\kappa}{2}})$. Since $0 \leq q < \frac{[\alpha-(1+2\tau)(1-t)](2\beta-1)}{4\alpha^2}$ and $0 < \kappa < \frac{\alpha-1-2\tau+(1+2\tau)t}{2\alpha^2(1-t)}$, we can choose $\kappa < \frac{\alpha-1-2\tau+(1+2\tau)t}{2\alpha^2(1-t)}$ and $\kappa$ is arbitrarily close to $\kappa < \frac{\alpha-1-2\tau+(1+2\tau)t}{2\alpha^2(1-t)}$ such that

$0 \leq q < \frac{(2\beta-1)(1-t)\kappa}{2}$. Then we have $\frac{(1-2\beta)(1-t)\kappa}{2} + q < 0$. From (136) and (138), we have

$$\|(I+\frac{1}{\sigma^2}\Lambda_{1:S}\Phi_{1:S}^T\Phi_{1:S})^{-1}\boldsymbol{\mu}_{1:S}\|_2 = \Theta(n^{\max\{-(1-t),\frac{(1-2\beta)(1-t)}{2\alpha}\}}\log^{k/2}n) + O((\frac{1}{\delta}+1)n^{\frac{(1-2\beta)(1-t)}{2\alpha}+\frac{(1-2\beta)(1-t)\kappa}{2}})$$

$$=\Theta(n^{\max\{-(1-t),\frac{(1-2\beta)(1-t)}{2\alpha}\}}\log^{k/2}n) + O((n^{q+\frac{(1-2\beta)(1-t)}{2\alpha}+\frac{(1-2\beta)(1-t)\kappa}{2}})$$

$$=\Theta(n^{\max\{-(1-t),\frac{(1-2\beta)(1-t)}{2\alpha}\}}\log^{k/2}n)$$

$$=(1+o(1))\|(I+\frac{n}{\sigma^2}\Lambda_{1:S})^{-1}\hat{\boldsymbol{\mu}}_{1:R}\|_2$$

$$=(1+o(1))\|(I+\frac{n}{\sigma^2}\Lambda_R)^{-1}\boldsymbol{\mu}_R\|_2.$$

(139)

Hence $G_{2,S}(D_n) = \frac{1+o(1)}{2\sigma^2}\|(I+\frac{1}{\sigma^2}\Lambda_{1:S}\Phi_{1:S}^T\Phi_{1:S})^{-1}\boldsymbol{\mu}_{1:S}\|_2^2 = \frac{1}{\sigma^2}\Theta(n^{(1-t)\max\{-2,\frac{1-2\beta}{\alpha}\}}\log^{k/2}n)$. Then by (133), we have

$$G_2(D_n) = \frac{1}{\sigma^2}\Theta(n^{\max\{-2(1-t),\frac{(1-2\beta)(1-t)}{\alpha}\}}\log^{k/2}n) + \tilde{O}\left((\frac{1}{\delta}+1)\frac{n}{\sigma^2}S^{\max\{1/2-\beta,1-\alpha+2\tau\}}\right).$$

Choosing $S = n^{\max\left\{1,\frac{-t}{(\alpha-1-2\tau)},\left(\frac{1+q+\min\{2,\frac{2\beta-1}{\alpha}\}}{\min\{\beta-1/2,\alpha-1-2\tau\}}+1\right)(1-t)\right\}}$, we get the result. $\square$

*Proof of Theorem 9.* From Lemmas 39 and 41 and $\frac{1}{\alpha}-1 > -2$, we have that with probability of at least $1 - 7\tilde{\delta}$,

$$\mathbb{E}_{\boldsymbol{\epsilon}}G(D_n) = \frac{1+o(1)}{2\sigma^2}(\mathrm{Tr}(I+\frac{n}{\sigma^2}\Lambda_R)^{-1}\Lambda_R - \|\Lambda_R^{1/2}(I+\frac{n}{\sigma^2}\Lambda_R)^{-1}\|_F^2 + \|(I+\frac{n}{\sigma^2}\Lambda_R)^{-1}\boldsymbol{\mu}_R\|_2^2)$$

$$=\frac{1}{\sigma^2}\Theta(n^{\frac{(1-\alpha)(1-t)}{\alpha}}) + \frac{1}{\sigma^2}\Theta(n^{\max\{-2(1-t),\frac{(1-2\beta)(1-t)}{\alpha}\}}\log^{k/2}n)$$

$$=\frac{1}{\sigma^2}\Theta(n^{\max\{\frac{(1-\alpha)(1-t)}{\alpha},\frac{(1-2\beta)(1-t)}{\alpha}\}})$$

(140)

where $k = \begin{cases} 0, & 2\alpha \neq 2\beta-1 \\ 1, & 2\alpha = 2\beta-1 \end{cases}$, and $R = n^{(\frac{1}{\alpha}+\kappa)(1-t)}, \kappa > 0$.

Furthermore, we have

$$\mathrm{Tr}(I+\frac{n}{\sigma^2}\Lambda)^{-1}\Lambda - \mathrm{Tr}(I+\frac{n}{\sigma^2}\Lambda_R)^{-1}\Lambda_R$$

$$=\sum_{p=R+1}^{\infty}\frac{\lambda_p}{1+\frac{n}{\sigma^2}\lambda_p} \leq \sum_{p=R+1}^{\infty}\frac{C_\lambda p^{-\alpha}}{1+\frac{n}{\sigma^2}C_\lambda p^{-\alpha}} \leq \sum_{p=R+1}^{\infty}C_\lambda p^{-\alpha} = \frac{n}{\sigma^2}O(R^{1-\alpha})$$

$$=O(n^{(1-\alpha)(1-t)(\frac{1}{\alpha}+\kappa)})$$

$$=o(n^{\frac{(1-\alpha)(1-t)}{\alpha}}).$$

Then we have

$$\mathrm{Tr}(I+\frac{n}{\sigma^2}\Lambda_R)^{-1}\Lambda_R = \mathrm{Tr}(I+\frac{n}{\sigma^2}\Lambda)^{-1}\Lambda(1+o(1)). \tag{141}$$

Similarly we can prove

$$\|\Lambda_R^{1/2}(I+\frac{n}{\sigma^2}\Lambda_R)^{-1}\|_F^2 = \|\Lambda^{1/2}(I+\frac{n}{\sigma^2}\Lambda)^{-1}\|_F^2(1+o(1)) \tag{142}$$

$$\|(I+\frac{n}{\sigma^2}\Lambda_R)^{-1}\boldsymbol{\mu}_R\|_2^2 = \|(I+\frac{n}{\sigma^2}\Lambda)^{-1}\boldsymbol{\mu}\|_2^2(1+o(1)) \tag{143}$$

Letting $\delta = 7\tilde{\delta}$, the proof is complete. $\square$

In the case of $\mu_0 > 0$, we have the following lemma:

**Lemma 42.** *Let $\delta = n^{-q}$ where $0 \leq q < \frac{[\alpha-(1+2\tau)(1-t)](2\beta-1)}{4\alpha^2}$. Under Assumptions 4, 5 and 6, assume that $\mu_0 > 0$. Then with probability of at least $1 - 6\delta$ over sample inputs $(x_i)_{i=1}^n$, we have $G_2(D_n) = \frac{1}{2\sigma^2}\mu_0^2 + o(1)$.*

*Proof of Lemma 42.* Let $S = n^D$. Let $G_{2,S}(D_n) = \mathbb{E}_{(x_{n+1},y_{n+1})}(T_{2,S}(D_{n+1}) - T_{2,S}(D_n))$. By Lemma 33, when $S > n^{\max\{1, \frac{-t}{(\alpha-1-2\tau)}\}}$, with probability of at least $1 - 3\delta$ we have that

$$|G_2(D_n) - G_{2,S}(D_n)| = |\mathbb{E}_{(x_{n+1},y_{n+1})}[T_2(D_{n+1}) - T_{2,S}(D_{n+1})] - [T_2(D_n) - T_{2,S}(D_n)]|$$

$$= \left|\mathbb{E}_{(x_{n+1},y_{n+1})}\tilde{O}\left((\tfrac{1}{\delta}+1)\tfrac{1}{\sigma^2}(n+1)S^{\max\{1/2-\beta,1-\alpha+2\tau\}}\right) - \tilde{O}\left((\tfrac{1}{\delta}+1)\tfrac{1}{\sigma^2}nS^{\max\{1/2-\beta,1-\alpha+2\tau\}}\right)\right|$$

$$= \tilde{O}\left((\tfrac{1}{\delta}+1)\tfrac{1}{\sigma^2}nS^{\max\{1/2-\beta,1-\alpha+2\tau\}}\right)$$

Let $\Lambda_S = \text{diag}\{\lambda_1, \dots, \lambda_S\}$, $\Phi_S = (\phi_1(\mathbf{x}), \phi_1(\mathbf{x}), \dots, \phi_S(\mathbf{x}))$ and $\boldsymbol{\mu}_S = (\mu_1, \dots, \mu_S)$. Define $\eta_S = (\phi_0(x_{n+1}), \phi_1(x_{n+1}), \dots, \phi_S(x_{n+1}))^T$ and $\widetilde{\Phi}_S = (\Phi_S^T, \eta_S)^T$. By the same technique as in the proof of Lemma 34, we replace $\Lambda_R$ by $\tilde{\Lambda}_{\epsilon,R} = \text{diag}\{\epsilon, \lambda_1, \dots, \lambda_R\}$, let $\epsilon \to 0$ and show the counterpart of the result (135) in the proof of Lemma 41:

$$G_{2,S}(D_n) = \mathbb{E}_{(x_{n+1},y_{n+1})}(T_{2,S}(D_{n+1}) - T_{2,S}(D_n))$$

$$= \mathbb{E}_{(x_{n+1},y_{n+1})}\left(\frac{1}{2\sigma^2}\frac{\boldsymbol{\mu}_S^T(I+\frac{1}{\sigma^2}\Phi_S^T\Phi_S\Lambda_S)^{-1}\eta_S\eta_S^T(I+\frac{1}{\sigma^2}\Lambda_S\Phi_S^T\Phi_S)^{-1}\boldsymbol{\mu}_S}{1+\frac{1}{\sigma^2}\eta_S^T(I+\frac{1}{\sigma^2}\Lambda_S\Phi_S^T\Phi_S)^{-1}\Lambda_S\eta_S}\right)$$

$$= \mathbb{E}_{(x_{n+1},y_{n+1})}\left(\frac{1+o(1)}{2\sigma^2}\boldsymbol{\mu}_S^T(I+\frac{1}{\sigma^2}\Phi_S^T\Phi_S\Lambda_S)^{-1}\eta_S\eta_S^T(I+\frac{1}{\sigma^2}\Lambda_S\Phi_S^T\Phi_S)^{-1}\boldsymbol{\mu}_S\right)$$

$$= \frac{1+o(1)}{2\sigma^2}\boldsymbol{\mu}_S^T(I+\frac{1}{\sigma^2}\Phi_S^T\Phi_S\Lambda_S)^{-1}(I+\frac{1}{\sigma^2}\Lambda_S\Phi_S^T\Phi_S)^{-1}\boldsymbol{\mu}_S$$

$$= \frac{1+o(1)}{2\sigma^2}\|(I+\frac{1}{\sigma^2}\Lambda_S\Phi_S^T\Phi_S)^{-1}\boldsymbol{\mu}_S\|_2^2, \tag{144}$$

where in the fourth to last equality we used the Sherman–Morrison formula, in the third inequality we used (118), and in the last equality we used the fact that $\mathbb{E}_{(x_{n+1},y_{n+1})}\eta_{1:S}\eta_{1:S}^T = I$.

Let $\hat{\boldsymbol{\mu}}_R = (\mu_0, \mu_1, \dots, \mu_R, 0, \dots, 0) \in \mathbb{R}^S$. Then we have

$$\|(I+\frac{1}{\sigma^2}\Lambda_S\Phi_S^T\Phi_S)^{-1}\boldsymbol{\mu}_S\|_2 \le \|(I+\frac{1}{\sigma^2}\Lambda_S\Phi_S^T\Phi_S)^{-1}\hat{\boldsymbol{\mu}}_R\|_2 + \|(I+\frac{1}{\sigma^2}\Lambda_S\Phi_S^T\Phi_S)^{-1}(\boldsymbol{\mu}_S-\hat{\boldsymbol{\mu}}_R)\|_2,$$

$$\|(I+\frac{1}{\sigma^2}\Lambda_S\Phi_S^T\Phi_S)^{-1}\boldsymbol{\mu}_S\|_2 \ge \|(I+\frac{1}{\sigma^2}\Lambda_S\Phi_S^T\Phi_S)^{-1}\hat{\boldsymbol{\mu}}_R\|_2 - \|(I+\frac{1}{\sigma^2}\Lambda_S\Phi_S^T\Phi_S)^{-1}(\boldsymbol{\mu}_S-\hat{\boldsymbol{\mu}}_R)\|_2. \tag{145}$$

Choose $R = n^{(\frac{1}{\alpha}+\kappa)(1-t)}$ where $0 < \kappa < \frac{\alpha-1-2\tau+(1+2\tau)t}{\alpha^2(1-t)}$. In Lemma 29, (62), we showed that with probability of at least $1 - \delta$,

$$\|(I+\frac{1}{\sigma^2}\Lambda_{1:R}\Phi_{1:R}^T\Phi_{1:R})^{-1}\boldsymbol{\mu}_{1:R}\|_2 = \Theta(n^{(1-t)\max\{-1,\frac{1-2\beta}{2\alpha}\}}\log^{k/2}n)$$
$$= (1+o(1))\|(I+\frac{n}{\sigma^2}\Lambda_{1:R})^{-1}\boldsymbol{\mu}_{1:R}\|_2, \tag{146}$$

where $k = \begin{cases} 0, & 2\alpha \ne 2\beta-1, \\ 1, & 2\alpha = 2\beta-1. \end{cases}$ The same proof holds if we replace $\Phi_{1:R}$ with $\Phi_{1:S}$, $\Lambda_{1:R}$ with $\Lambda_{1:S}$, and $\boldsymbol{\mu}_{1:R}$ with $\hat{\boldsymbol{\mu}}_{1:R}$. We have

$$\|(I+\frac{1}{\sigma^2}\Lambda_{1:S}\Phi_{1:S}^T\Phi_{1:S})^{-1}\hat{\boldsymbol{\mu}}_{1:R}\|_2 = \Theta(n^{(1-t)\max\{-1,\frac{1-2\beta}{2\alpha}\}}\log^{k/2}n)$$
$$= (1+o(1))\|(I+\frac{n}{\sigma^2}\Lambda_{1:S})^{-1}\hat{\boldsymbol{\mu}}_{1:R}\|_2. \tag{147}$$

So we have

$$\|(I+\frac{1}{\sigma^2}\Lambda_S\Phi_S^T\Phi_S)^{-1}\hat{\boldsymbol{\mu}}_R\|_2 = \mu_0 + \Theta(n^{(1-t)\max\{-1,\frac{1-2\beta}{2\alpha}\}}\log^{k/2}n)$$
$$= \mu_0 + o(1). \tag{148}$$

Next we bound $\|(I + \frac{1}{\sigma^2}\Lambda_S\Phi_S^T\Phi_S)^{-1}(\boldsymbol{\mu}_S - \hat{\boldsymbol{\mu}}_R)\|_2$. By Assumption 5, we have that $\|\boldsymbol{\mu}_S - \hat{\boldsymbol{\mu}}_R\|_2 = O(R^{\frac{1-2\beta}{2}})$. For any $\xi \in \mathbb{R}^S$ and $\|\xi\|_2 = 1$, using the Woodbury matrix

identity, with probability of at least $1-2\delta$ we have

$$
|\xi^T(I+\frac{1}{\sigma^2}\Lambda_S\Phi_S^T\Phi_S)^{-1}(\boldsymbol{\mu}_S-\hat{\boldsymbol{\mu}}_R)|
$$
$$
=|\xi^T\left(I-\frac{1}{\sigma^2}\Lambda_S\Phi_S^T(I+\frac{1}{\sigma^2}\Phi_S\Lambda_S\Phi_S^T)^{-1}\Phi_S\right)(\boldsymbol{\mu}_S-\hat{\boldsymbol{\mu}}_R)|
$$
$$
=|\xi^T(\boldsymbol{\mu}_S-\hat{\boldsymbol{\mu}}_R)-\frac{1}{\sigma^2}\xi^T\Lambda_S\Phi_S^T(I+\frac{1}{\sigma^2}\Phi_S\Lambda_S\Phi_S^T)^{-1}\Phi_S(\boldsymbol{\mu}_S-\hat{\boldsymbol{\mu}}_R)|
$$
$$
\leq\|\xi\|_2\|\boldsymbol{\mu}_S-\hat{\boldsymbol{\mu}}_R\|_2+\frac{1}{\sigma^2}|\xi^T\Lambda_S\Phi_S^T(I+\frac{1}{\sigma^2}\Phi_S\Lambda_S\Phi_S^T)^{-1}\Phi_S(\boldsymbol{\mu}_S-\hat{\boldsymbol{\mu}}_R)|
$$
$$
\leq O(R^{\frac{1-2\beta}{2}})+\frac{1}{\sigma^2}\|(I+\frac{1}{\sigma^2}\Phi_S\Lambda_S\Phi_S^T)^{-1}\Phi_S\Lambda_S\xi\|_2\|\Phi_S(\boldsymbol{\mu}_S-\hat{\boldsymbol{\mu}}_R)\|_2
$$
$$
=O(R^{\frac{1-2\beta}{2}})+\frac{1}{\sigma^2}O(\sqrt{(\frac{1}{\delta}+1)n}\cdot n^{-(1-t)})O(\sqrt{(\frac{1}{\delta}+1)n}R^{\frac{1-2\beta}{2}})
$$
$$
=O((\frac{1}{\delta}+1)R^{\frac{1-2\beta}{2}}),
$$

where in the second to last step we used Corollary 20 to show $\|\Phi_S(\boldsymbol{\mu}_S-\hat{\boldsymbol{\mu}}_R)\|_2=O(\sqrt{(\frac{1}{\delta}+1)n}R^{\frac{1-2\beta}{2}})$ with probability of at least $1-\delta$, and Lemma 40 to show that $\|(I+\frac{1}{\sigma^2}\Phi_S\Lambda_S\Phi_S^T)^{-1}\Phi_S\Lambda_S\xi\|_2=O(\sqrt{(\frac{1}{\delta}+1)n}\cdot n^{-(1-t)})$ with probability of at least $1-\delta$. Since $R=n^{(\frac{1}{\alpha}+\kappa)(1-t)}$, we have

$$
|\xi^T(I+\frac{1}{\sigma^2}\Lambda_S\Phi_S^T\Phi_S)^{-1}(\boldsymbol{\mu}_S-\hat{\boldsymbol{\mu}}_R)|=O((\frac{1}{\delta}+1)n^{\frac{(1-2\beta)(1-t)}{2\alpha}+\frac{(1-2\beta)(1-t)\kappa}{2}}).
$$

Since $\xi$ is arbitrary, we have $\|(I+\frac{1}{\sigma^2}\Lambda_S\Phi_S^T\Phi_S)^{-1}(\boldsymbol{\mu}_S-\hat{\boldsymbol{\mu}}_R)\|_2=O((\frac{1}{\delta}+1)n^{\frac{(1-2\beta)(1-t)}{2\alpha}+\frac{(1-2\beta)(1-t)\kappa}{2}})$. Since $0\leq q<\frac{[\alpha-(1+2\tau)(1-t)](2\beta-1)}{4\alpha^2}$ and $0<\kappa<\frac{\alpha-1-2\tau+(1+2\tau)t}{2\alpha^2(1-t)}$, we can choose $\kappa<\frac{\alpha-1-2\tau+(1+2\tau)t}{2\alpha^2(1-t)}$ and $\kappa$ is arbitrarily close to $\kappa<\frac{\alpha-1-2\tau+(1+2\tau)t}{2\alpha^2(1-t)}$ such that $0\leq q<\frac{(2\beta-1)(1-t)\kappa}{2}$. Then we have $\frac{(1-2\beta)(1-t)\kappa}{2}+q<0$. From (145) and (148), we have

$$
\|(I+\frac{1}{\sigma^2}\Lambda_S\Phi_S^T\Phi_S)^{-1}\boldsymbol{\mu}_S\|_2=\mu_0+\Theta(n^{(1-t)\max\{-1,\frac{1-2\beta}{2\alpha}\}}\log^{k/2}n)+O((\frac{1}{\delta}+1)n^{\frac{(1-2\beta)(1-t)}{2\alpha}+\frac{(1-2\beta)(1-t)\kappa}{2}})
$$
$$
=\mu_0+\Theta(n^{(1-t)\max\{-1,\frac{1-2\beta}{2\alpha}\}}\log^{k/2}n)
$$
$$
=\mu_0+o(1).
$$
$$
\tag{149}
$$

Hence $G_{2,S}(D_n)=\frac{1+o(1)}{2\sigma^2}\|(I+\frac{1}{\sigma^2}\Lambda_S\Phi_S^T\Phi_S)^{-1}\boldsymbol{\mu}_S\|_2^2=\frac{1}{2\sigma^2}\mu_0^2+o(1)$. Then by (144), $G_2(D_n)=\frac{1}{2\sigma^2}\mu_0^2+o(1)+\tilde{O}((\frac{1}{\delta}+1)nS^{\max\{1/2-\beta,1-\alpha\}})$.

Choosing $S=n^{\max\left\{1,\frac{-t}{(\alpha-1-2\tau)},\left(\frac{1+q+\min\{2,\frac{2\beta-1}{\alpha}\}}{\min\{\beta-1/2,\alpha-1-2\tau\}}+1\right)(1-t)\right\}}$, we get the result. $\qquad\square$

*Proof of Theorem 11.* According to Lemma 42, $G_2(D_n)=\frac{1}{2\sigma^2}\mu_0^2+o(1)$. By Lemma 39, we have $G_1(D_n)=\Theta(n^{\frac{(1-\alpha)(1-t)}{\alpha}})$. Then $\mathbb{E}_\epsilon G(D_n)=G_1(D_n)+G_2(D_n)=\frac{1}{2\sigma^2}\mu_0^2+o(1)$. $\qquad\square$

### D.3 PROOFS RELATED TO THE EXCESS MEAN SQUARED GENERALIZATION ERROR

*Proof of Theorem 12.* For $\mu_0 = 0$, we can show that

$$
\begin{aligned}
\mathbb{E}_\epsilon M(D_n) &= \mathbb{E}_\epsilon \mathbb{E}_{x_{n+1}} [\bar{m}(x_{n+1}) - f(x_{n+1})]^2 \\
&= \mathbb{E}_\epsilon \mathbb{E}_{x_{n+1}} [K_{x_{n+1}\mathbf{x}}(K_n + \sigma_{\text{model}}^2 I_n)^{-1}\mathbf{y} - f(x_{n+1})]^2 \\
&= \mathbb{E}_\epsilon \mathbb{E}_{x_{n+1}} [\eta^T \Lambda \Phi^T [\Phi\Lambda\Phi^T + \sigma_{\text{model}}^2 I_n)^{-1}(\Phi\mu + \epsilon) - \eta^T \mu]^2 \\
&= \mathbb{E}_\epsilon \mathbb{E}_{x_{n+1}} [\eta^T \Lambda \Phi^T (\Phi\Lambda\Phi^T + \sigma_{\text{model}}^2 I_n)^{-1}\epsilon]^2 \\
&\quad + \mathbb{E}_{x_{n+1}} \left[\eta^T \left(\Lambda\Phi^T (\Phi\Lambda\Phi^T + \sigma_{\text{model}}^2 I_n)^{-1}\Phi - I\right)\mu\right]^2 \\
&= \sigma_{\text{true}}^2 \text{Tr}\Lambda\Phi^T (\Phi\Lambda\Phi^T + \sigma_{\text{model}}^2 I_n)^{-2}\Phi\Lambda \\
&\quad + \mu^T \left(I + \tfrac{1}{\sigma_{\text{model}}^2}\Phi^T\Phi\Lambda\right)^{-1}\left(I + \tfrac{1}{\sigma_{\text{model}}^2}\Lambda\Phi^T\Phi\right)^{-1}\mu \\
&= \tfrac{\sigma_{\text{true}}^2}{\sigma_{\text{model}}^2}\text{Tr}(I + \tfrac{\Lambda\Phi^T\Phi}{\sigma_{\text{model}}^2})^{-1}\Lambda - \text{Tr}(I + \tfrac{\Lambda\Phi^T\Phi}{\sigma_{\text{model}}^2})^{-2}\Lambda + \|(I + \tfrac{1}{\sigma_{\text{model}}^2}\Lambda\Phi^T\Phi)^{-1}\mu\|_2^2.
\end{aligned}
$$

According to (139) from the proof of Lemma 41, the truncation procedure (133) and (143), with probability of at least $1 - \delta$ we have

$$
\|(I + \tfrac{1}{\sigma_{\text{model}}^2}\Lambda\Phi^T\Phi)^{-1}\mu\|_2^2 = \Theta(n^{\max\{-2(1-t), \frac{(1-2\beta)(1-t)}{\alpha}\}}\log^{k/2}n) = (1+o(1))\|(I + \tfrac{n}{\sigma_{\text{model}}^2}\Lambda)^{-1}\boldsymbol{\mu}\|_2^2,
$$

where $k = \begin{cases} 0, & 2\alpha \neq 2\beta - 1, \\ 1, & 2\alpha = 2\beta - 1. \end{cases}$

According to (121) and (126) from the proof of Lemma 39, the truncation procedure (115), (141) and (142), with probability of at least $1 - \delta$ we have

$$
\begin{aligned}
&\text{Tr}(I + \tfrac{\Lambda\Phi^T\Phi}{\sigma_{\text{model}}^2})^{-1}\Lambda - \text{Tr}(I + \tfrac{\Lambda\Phi^T\Phi}{\sigma_{\text{model}}^2})^{-2}\Lambda \\
&= \left(\text{Tr}(I + \tfrac{n}{\sigma_{\text{model}}^2}\Lambda)^{-1}\Lambda\right)(1 + o(1)) - \|\Lambda^{1/2}(I + \tfrac{n}{\sigma_{\text{model}}^2}\Lambda)^{-1}\|_F^2(1 + o(1)) \\
&= \Theta(n^{\frac{(1-\alpha)(1-t)}{\alpha}}).
\end{aligned}
$$

Combining the above two equations we get

$$
\begin{aligned}
\mathbb{E}_\epsilon M(D_n) &= (1+o(1))\left(\tfrac{\sigma_{\text{true}}^2}{\sigma_{\text{model}}^2}\left(\text{Tr}(I + \tfrac{n}{\sigma_{\text{model}}^2}\Lambda)^{-1}\Lambda - \|\Lambda^{1/2}(I + \tfrac{n}{\sigma_{\text{model}}^2}\Lambda)^{-1}\|_F^2\right) + \|(I + \tfrac{n}{\sigma_{\text{model}}^2}\Lambda)^{-1}\boldsymbol{\mu}\|_2^2\right) \\
&= \tfrac{\sigma_{\text{true}}^2}{\sigma_{\text{model}}^2}\Theta(n^{\frac{(1-\alpha)(1-t)}{\alpha}}) + \Theta(n^{\max\{-2(1-t), \frac{(1-2\beta)(1-t)}{\alpha}\}}\log^{k/2}n) \\
&= \sigma_{\text{true}}^2\Theta(n^{\frac{1-\alpha-t}{\alpha}}) + \Theta(n^{\max\{-2(1-t), \frac{(1-2\beta)(1-t)}{\alpha}\}}\log^{k/2}n) \\
&= \Theta\left(\max\{\sigma_{\text{true}}^2 n^{\frac{1-\alpha-t}{\alpha}}, n^{\frac{(1-2\beta)(1-t)}{\alpha}}\}\right)
\end{aligned}
$$

When $\mu_0 > 0$, according to (149) in the proof of Lemma 42 and the truncation procedure (133), with probability of at least $1 - \delta$ we have

$$
\begin{aligned}
\mathbb{E}_\epsilon M(D_n) &= \Theta(n^{\frac{(1-\alpha)(1-t)}{\alpha}}) + \mu_0^2 + o(1) \\
&= \mu_0^2 + o(1).
\end{aligned}
$$

$\square$

