# OpenReview forum: "Learning Curves for Gaussian Process Regression with Power-Law Priors and Targets"
_ICLR.cc/2022/Conference — ICLR 2022 Poster_

### Official Review · Reviewer_LR4t · 2021-10-18

**Correctness:** 3
**Technical Novelty And Significance:** 2
**Empirical Novelty And Significance:** 2
**Recommendation:** 6
**Confidence:** 4

**Main Review:**

Overall the writing is clear. I found it helpful that the authors gave proof sketches and summary of the main steps detailed in the rather long appendix. The results improved upon those in the literature by being more widely applicable and with less stringent assumptions. Moreover, several rates derived by past authors are special cases of the general rates derived in this paper.

Here are some questions /comments:
1. The error variance $\sigma^2$ is assumed to be known which is rarely the case in practice. Also, the ReLU network mentioned in the paper is noiseless, i.e., there is no $\epsilon_i$. So how do the results presented in this paper reconcile with real-world practice?

2. Are the rates the best possible under power-law decay of the target and prior kernel? Is it possible to get better rates by using different kernel functions or eigenfunctions?

3. It can be seen that $\alpha$ encodes the prior smoothness, and it is related to the order of the kernel family used. In the experiments, the authors used the first-order arc-cosine kernel to yield $\alpha=4$. What is the rationale of choosing first-order and not higher orders?

4. The interplay between $\beta$ and $\alpha$ is summarized in the second paragraph of the conclusion. However, how do we apply this conclusion in practice since we often do not know $\beta$?

5. How did you compute the blue lines in Figure 1? In particular, where did $2.0795$ and $0.7875$ come from in $2.0795n^{-0.7875}$ for the excess mean square error of $f_1$?

############################################################################################################

TYPOS:
1. The line after (2), $K_{x\mathbf{X}}$ should be $(k(x,x_1),\dotsc,k(x,x_n))$ a row vector. Also to be clear, you need to assume that the kernel is symmetric in order for $K_{x\mathbf{X}}=K_{\mathbf{X}x}^T$ to be true.
2. In (3), I think $q(x,y)$ should be $q(y|x)$
3. The display after (17),  $\|(\mathbf{I}+\mathbf{\Phi\Lambda\Phi}^T/\sigma^2)^{-1}\|_2$
4. In (20), $F_0(D_n)$ should be $F^0(D_n)=T_1(D_n)+T_2(D_n)$ and not $\mathbb{E}_{\mathbf{\epsilon}}F^0(D_n)=T_1(D_n)+T_2(D_n)$ as defined in the line after (13).
5. In the Proof sketch of Theorem 9 the first step, I think you need to take expectations with respect to the errors in $G_{1,R}(D_n)$ and $G_{2,R}(D_n)$. See also the proofs.
6. In the Proof sketch of Theorem 9 the second step, what is $\delta$?

**Summary Of The Paper:**

Working under the nonparametric regression model, the authors established asymptotic rates for the Bayesian generalization error, normalized stochastic complexity and excess mean square error. The authors adopted the Bayesian approach by endowing the unknown regression function with a Gaussian process prior, and the covariance function is parameterized by a kernel function. By diagonalizing this kernel, the authors used the resulting eigenfunctions to represent the target function. They assumed that the kernel eigenvalues and the target basis coefficients decay according to the power-law, i.e., polynomial decay. Lastly, a simulation experiment was performed to verify the claims.

**Summary Of The Review:**

I feel that some parts of the paper need more explanation and exposition. For example, discussion about the rates' optimality and modeling choices such as knowing $\sigma$ would significantly improve this paper.

---

> ### Author Response · Authors · 2021-11-19
> **Response to Reviewer LR4t**
>
> Thanks for your thougtful review and valuable suggestions.
>
> For your questions and comments:
>
> * In regard to your comments "The error variance $\sigma^2$ is assumed to be known which is rarely the case in practice" and "discussion on modeling choices":
>
>     This is a valid point. In general the $\sigma$ of the data and the $\sigma$ of the model can be different. **In our updated results we now distinguish between the variance of the noise ($\sigma_{\mathrm{true}}$) and that of the model ($\sigma_{\mathrm{model}}$).**  This allows us to show, for example in kernel ridge regression (KRR), how the model variance (equivalent to the regularization strength in KRR) influences the mean squared generalization error of KRR (please see Remark 14). In a similar spirit, we also incorporate the noise and model variances in the Bayesian generalization error bound in Theorem 9.
>
> * In response to your comment about $\epsilon_i$, if we understand your question correctly, our response is as follows: In our results $\epsilon_i$ is the noise in the label which has variance $\sigma_{\mathrm{true}}$. Our results are general and cover both the noiseless and the noisy cases. For example, in the noiseless setting for KRR ($\sigma_{\mathrm{true}}=0$) with constant regularization ($t=0$), our Theorem 12 implies that the mean squared error behaves as $\Theta(n^{\frac{1-2\beta}{\alpha}})$. Please see Remark 14.
>
>
> * In response to "Are the rates the best possible under power-law decay of the target and prior kernel? Is it possible to get better rates by using different kernel functions or eigenfunctions?"
>
>      **Yes! Indeed our updated manuscript now shows optimality** (please see common response C1).
>
> * "What is the rationale of choosing first-order and not higher orders?"
>
>     We can choose higher order arc-cosine kernels in the experiments without changing other parameters. The empirical and theoretical rates will still match. **In Appendix A, we have included more experiments confirming our theory for arc-cosine kernels of different orders, with and without biases.**
>
>
> * "How do we apply this conclusion in practice since we often do not know $\beta$?".
>
>     This is a good point. In order to apply the insights from the interplay, we first need to estimate the value of $\beta$ in practice, which we are currently investigating both theoretically and experimentally.
>
> * "How did you compute the blue lines in Figure 1?"
>
>     In Figure 1, the orange curves show the linear regression fit for the experimental values (in blue) of the log excess mean squared error as a function of log $n$. This description now also appears in the caption of the Figure 1.
>
> * In regard to your comment "explanation and exposition / discussion about optimality of the rates".
>
>     We have improved the exposition by providing more intuition (wherever applicable) on how to interpret our bounds (see text just below Eqs. 13-16, in Section 3 P.5, and the paragraph following Eq.20 in P.7) and added new experiments (Appendix A) demonstrating the theory following your valuable suggestion.
>
>     In the revision we now address the optimality question as well. We have improved all our results by showing that our previous upper bounds are in fact optimal. Hence in the revision our results are now expressed in $\Theta$-notation rather than $O$-notation.
>
> For the typos:
>
> (1). "The line after (2), $K_{x\mathbf{X}}$ should be a row vector."
>
> We fixed the typo. Thanks for pointing out.
>
> (2). "In (3), I think $q(x,y)$ should be $q(y|x)$".
>
> No, it should be $q(x,y)$.
>
> (3). "The display after (17), $|(\mathbf{I}+\mathbf{\Phi\Lambda\Phi}^T/\sigma^2)^{-1}|_2$":
>
> We fixed the typo. Thanks for pointing out.
>
> (4 and 5). For your comments on $F_0(D_n)$ and $G(D_n)$, the components $T_1$ and $T_2$ arise from the decomposition of $\mathbb{E} F^0(D_n)$, i.e., $\mathbb{E}F^0(D_n)=T_1(D_n)+T_2(D_n)$ (please see Eqs. 8, 13 and 14). Likewise  $\mathbb{E} G(D_n) = G_1(D_n)+G_2(D_n)$ (please see Eqs. 15 and 16 and the text following that).
>
> The proof of Theorem 9 follows steps similar to that of Theorem 7 and we had to shorten the proof sketch of the former (in the main text) due to space constraints. Moreover, the statements of all our results now address optimality with the statements in the main text and the proofs in the appendix updated accordingly (please see common response C1).
>
> (6). "In the Proof sketch of Theorem 9 the second step, what is $\delta$?"
>
> $\delta$ appeared in the statement of the Theorem 9 in the initial submission and refers to the failure probability. We now give an explicit range for this failure probability as a function of $n$, $\alpha$ and $\beta$ in all our theorems.
> Please see common response C3.
>
>
> We made thorough efforts to address all your comments. We believe that the revised manuscript improved significantly over the initial submission. We hope that you will agree and consider raising your score to an accept.

---

> > ### Comment · Reviewer_LR4t · 2021-11-19
> > **Response to author**
> >
> > Thank you very much for your detailed explanation.

---

> > > ### Author Response · Authors · 2021-11-20
> > > **Re: Response to author**
> > >
> > > Thank you for the prompt acknowledgement. As mentioned, we made thorough efforts to address all your previous comments. We hope our responses and updated manuscript will facilitate your assessment and motivate you to update your recommendation. Kindly do let us know if there is anything else you would like to see improved or clarified in the few remaining days of the rebuttal period. We remain attentive to your feedback!

---

### Official Review · Reviewer_e3WT · 2021-10-19

**Correctness:** 3
**Technical Novelty And Significance:** 3
**Empirical Novelty And Significance:** 2
**Recommendation:** 6
**Confidence:** 4

**Main Review:**

Studying the asymptotic decay of the generalisation error for power-law spectrum and target function decay is a classic subject in the Kernel ridge regression (KRR) literature, where these are known as capacity and source conditions. Indeed, the rate $n^{\max\left(\frac{1}{\alpha}-1, \frac{1-2\beta}{\alpha}\right)}$ reported in this work for the decay of the generalisation MSE is known in the KRR literature. The exponent $n^{\frac{1}{\alpha}-1}$ appears in Caponnetto and De Vito in 2007 [1] for optimally regularised KRR and a particular choice of target decay (eq. 19 of [1]), while the branch $n^{\frac{1-2\beta}{\alpha}}$ was reported more recently in [2,3]. The existence of a region with exponent $n^{\frac{1}{\alpha}-1}$ for other choices of regularisation and the cross-over between these two regimes is a particular case of the discussion in [4] (cross-over between blue and orange regions in Fig. 1, with purple point corresponding to [1]).

In my opinion the biggest contribution in this work is to put on rigorous footing some of the results appearing in [3,4] which were derived using semi-heuristic methods, and to extend them beyond the Gaussian design setting (note that [4] provides numerical evidence suggesting this decay holds more broadly) within the GP framework.

I do believe this is a valuable contribution, but the way the authors present it with respect to previous literature - specially in the abstract and introduction - is misleading. For instance, the abstract suggests the rates and cross-over reported for the MSE are new, and fail to cite [2-4]. The introduction only superficially makes contact with what is known from the KRR literature, and would benefit from a discussion like the one sketched above.

Specific comments / questions / suggestions:

1.  Top of page 5: *As mentioned in the introduction, this assumption is adopted in many recent works (Bahri et al., 2021;
Canatar et al., 2021; Nitanda and Suzuki, 2021). Velikanov and Yarotsky (2021)*.
This is a classic assumption on the KRR literature, and to my knowledge were introduced by Caponnetto and De Vito in the mid 2000s, see [1]. They are commonly known in this context as source and capacity conditions.
2.  Proof sketch, top of page 6: is there any intuition behind the choice of truncation $R=n^{\frac{2}{1+\alpha}-\kappa}$? Can this be related to hypercontractivity property in the concentration results from [5]?
3. The notation for functions, vector and matrices is confusing. Explicitly writing the conventions used, or writing more often the dimensions of vectors and matrices could help clarify this. For example, $\Phi_{R}\in\mathbb{R}^{n\times R}$, $K_{n}\in\mathbb{R}^{n\times n}$, etc.
4. Figure 1 is very hard to read, as one needs to zoom +300% to read the axis and caption. The matching of the power-laws would be easier to visualise in a log-log plot in terms of the slopes.
5. Remark 14: As mentioned before, even though [4] derives the rates for Gaussian designs, they provide numerical evidence that this result works more broadly.
6. Is it possible to recover the other rates from [4] from the GP framework?

[1] Caponnetto, A., De Vito, E., *Optimal Rates for the Regularized Least-Squares Algorithm*. Found Comput Math 7, 331-368 (2007). https://doi.org/10.1007/s10208-006-0196-8

[2] Stefano Spigler, Mario Geiger, and Matthieu Wyart. *Asymptotic learning curves of kernel methods: empirical data versus teacher-student paradigm*. Journal of Statistical Mechanics: Theory and Experiment, 2020(12):124001, 2020.

[3] Blake Bordelon, Abdulkadir Canatar, and Cengiz Pehlevan. *Spectrum dependent learning curves in kernel regression and wide neural networks*. In International Conference on Machine Learning, pages 1024-1034. PMLR, 2020

[4] Hugo Cui, Bruno Loureiro, Florent Krzakala, Lenka Zdeborová, *Generalization Error Rates in Kernel Regression: The Crossover from the Noiseless to Noisy Regime*, arXiv: 2105.15004 [stat.ML]

[5] Song Mei, Theodor Misiakiewicz, Andrea Montanari, *Generalization error of random features and kernel methods: hypercontractivity and kernel matrix concentration*, arXiv:2101.10588 [stat.ML]

**Summary Of The Paper:**

This paper considers the problem of fitting a target function corrupted by additive iid Gaussian noise using with a zero mean Gaussian Process (GP).

 It provides an asymptotic characterisation of the typical Bayesian and mean-squared generalisation errors under the assumption of power-law decay of the covariance eigenvalues and the target function coefficients when expressed in the covariance eigenbasis.

 The main result is to derive the exponents characterising the rate of decay of the aforementioned errors with the number of samples $n$, as a function of the spectrum and target function decays. The key technical step is to prove the concentration of $\Phi_{R}^{\top}\Phi_{R}\in\mathbb{R}^{R\times R}$ around $n I$ where $\Phi_{R}\in\mathbb{R}^{n\times R}$ the truncated matrix of eigenvectors evaluated at the data points, and $R=n^{\frac{2}{1+\alpha}-\kappa}$ with $0\leq \kappa < \frac{\alpha-1}{\alpha(\alpha+1)}$.

Finally, the authors provide one numerical experiment with the arc-cosine kernel and data uniformly distributed in the unit circle.

**Summary Of The Review:**

The results in this work are interesting, but in my opinion the discussion is largely misplaced with respect to existing literature. In particular the abstract and introduction. I am willing to raise my score in case the authors address these issues in a revision.

------------------------------------------------------------------------
**Update after the discussion period**

During the discussion period the authors have addressed all my questions and have welcomed my suggestions. In particular, they have extended their GP treatment to include the "strongly regularised" regime, finding results which are consistent with the literature. Although the rates derived in this work are known, this work derives them under a different framework, and broadens the scope under which they can be rigorously established.

For these reasons, I am updating my score towards an accept.

---

> ### Author Response · Authors · 2021-11-19
> **Response to Reviewer e3WT**
>
> We are glad that you consider our contributions valuable. Thank you for the thoughtful review and valuable comments.
>
> * In regard to your comment "the way the authors present it with respect to previous literature - specially in the abstract and introduction - is misleading,": Please note that we did cite references [2-4] and in fact we had highlighted these as the most important references in our initial submission. Following your suggestion, we have now **added a new paragraph titled "Asymptotics of the generalization error of kernel ridge regression (KRR)"** in the Introduction (P.2) that provides a better context for our results in relation to the existing literature on KRR. This includes the work of Caponnetto and De Vito (2007) (reference [1]) as well as other relevant works. We also include reference to the **"source" and "capacity" conditions as suggested wherever applicable and have also modified the abstract.**
>
> * With respect to novelty, we now **emphasize more clearly the relations and differences between our contribution and previous works** (see Contributions, P.2 and 3). Here is a brief summary:
>
> 1) Our Theorem 12 generalizes the result of Cui at al (2021) in three aspects:
>
>     - We show optimality of the bound compared to Cui et al (2021) who obtained an upper bound.
>
>     - We do not require a Gaussian design assumption, which they do, and hence our result is more generally applicable.
>
>     - We quantify precisely the improvement implied by our high-probability bounds compared to the bounds in expectation in Cui et al (2021). Please see Remark 14.
>
> 2) For the unrealizable case (the setting $\mu_0>0$ in Theorem 12) when the target function is outside the span of the features, we show that the generalization error converges to a constant that depends on $\mu_0$.
>
> 3) Our Theorem 9 recovers as a special case an earlier result due to Sollich and Halees (2002) when the model is correctly specified, showing that the generalization error scales as $\Theta(n^{\frac{1}{\alpha}-1})$ (see Remark 10). To the best of our knowledge, the exponent $n^{\frac{1}{\alpha}-1}$ appeared first in Sollich and Halees (2002) in the context of GPR and later in Caponnetto and De Vito (2007) in the context of optimally regularized KRR.
>
> 4) Our Theorem 12 when applied to KRR in the noiseless setting ($\sigma_{\mathrm{true}}=0$) with constant regularization ($t=0$), implies that the mean squared error behaves as $\Theta(n^{\frac{1-2\beta}{\alpha}})$. This recovers as a special case a result in Bordelon et al. (2020, Section 2.2). Also see Remark 14.
>
>
>
>
> For your specific comments / questions / suggestions:
>
> * In relation to the intuition for the choice of the truncation parameter, we have now added an intuition: The choice of the truncation parameter $R$ is governed by the fact that $\lambda_R =\Theta( R^{-\alpha})=n^{-1+\kappa\alpha}=o(n^{-1})$. This means that remainder is small compared with the truncated part.
>
> * In regard to the dimension of vectors and matrices, we have now stated it explicitly wherever applicable.
>
> * For Figure 1, we now use a larger font and a log-log plot as suggested.
>
> * In regard to "even though [4] derives the rates for Gaussian designs, they provide numerical evidence". Thanks for pointing this out. We have acknowledged this in a new paragraph titled "Asymptotics of the generalization error of kernel ridge regression (KRR)" in the Introduction (P.2) where we write: "Cui et al (2021) [4] present a unifying picture of the excess error decay rates under the  capacity and source conditions in terms of the interplay between noise and regularization illustrating their results with real datasets."
>
> * "Is it possible to recover the other rates from [4] from the GP framework?" We now exactly **recover the results pertaining to the "strong regularization" setting in Cui et al (2021) [4], additionally also showing optimality**. Please see Remark 14.
>
>
> Thanks again for your useful feedback. We have put significant efforts into addressing all of your concerns and believe that the manuscript is much improved as a consequence of this. If you agree, then please consider raising your score towards an accept.

---

> > ### Author Response · Authors · 2021-11-21
> > **Any further discussion before the end of rebuttal?**
> >
> > Dear reviewer,
> >
> > We made thorough efforts to address all comments from your initial review. We hope our responses and updated manuscript facilitate your assessment and motivate you to update your recommendation. Kindly do let us know if there is anything else you would like to see improved or clarified before the end of the rebuttal period. We remain attentive to your feedback!

---

> > ### Comment · Reviewer_e3WT · 2021-11-23
> > **Response to authors**
> >
> > Thank you for your reply and for taking all my comments into account. I am currently in the process of reading the changes, and will update my review accordingly during the remaining discussion period. In particular, I find very interesting that other regimes for the regularisation can also be derived from the GP framework.
> >
> > I know that the period for authors to update the manuscript is over, but a very minor comment I have spotted in the updated abstract: after you removed the explicit expression for the decay of the excess error, the sentence:
> >
> > > Under similar assumptions, we leverage the equivalence between GPR and kernel ridge regression (KRR) to show the generalization error of KRR.
> >
> > might be missing an object (to show that the error what?). Note that in my review I just suggested the authors acknowledge the rates mentioned in the abstract were known from previous literature from [1,3,4] + Sollich and Halees (2002) (by the way thank you for pointing this out, I was not aware one of the rates appeared in this work before [1]). To be clear about it, I don't think this diminishes this work. As you point out, some of these works derive these rates under stringent assumptions, and therefore the fact you can show it in a broader context is itself interesting in my opinion.

---

> > > ### Author Response · Authors · 2021-11-27
> > > **re: Response to authors**
> > >
> > > Thanks for acknowledging our responses. We are glad to read that you find the work interesting and plan to update your review. In the abstract what we mean is  `to describe the generalization of KRR'; which we will edit for clarity.

---

### Official Review · Reviewer_nLLV · 2021-10-29

**Correctness:** 3
**Technical Novelty And Significance:** 3
**Empirical Novelty And Significance:** 3
**Recommendation:** 8
**Confidence:** 3

**Main Review:**

## Strengths
- The early sections of the paper are clear and motivate the paper well.
- The results appear to be interesting, the problem of characterizing learning curves for Gaussian processes under mis-specification is important.

## Weaknesses
- Parts of the main results feel like a bit of list of results. I think the exposition in these sections could be improved by providing more context for each result as well as how the results differ. I would like to see more examples of how they apply to concrete examples.
- It isn't entirely clear to me how to interpret the results. In particular, the rate of decay depends on the smoothness of the kernel (unsurprising), but also on how quickly the coefficients of the observed function decay with a decomposition with respect to the Mercer basis of the kernel. I suspect that they can only decay quickly if the function is smooth, but the converse is not at all clear to me. I realize you cite other papers for this result, but for the purposes of being self-contained.
- The connection to neural networks is interesting, but the result is really a result on Gaussian processes. The connection to neural nets seems a bit orthogonal to the results in the paper, and I am not convinced it should be emphasized so heavily.

## Questions/comments
- Make explicit if $f$ is assumed to be in $L^2(\Omega,\rho)$ (I assume this is needed for the decomposition in eqn 9).
- Does $\phi_0$ in general depend on $f$? If so, perhaps consider noting this for clarity. (My understanding is that it is the projection of $f$ onto the orthogonal complement of the $\phi_p$).
- The assumption that eigenfunctions are uniformly bounded seems quite restrictive. I realize it is present in at least several other works in the GP literature (e.g. Braun, 2006, Vakili et al 2020). However, I it seems somewhat restrictive (for example for a squared exponential kernel and Gaussian inputs, this condition doesn't hold, Zhu et al 1997 eqn 44, of course this also doesn't have polynomial decay, just an example where the bounded eigenfunction condition doesn't hold). The authors provide an example on the sphere where this condition holds. Are there examples on compact subsets of $\mathbb{R}^d$ where this condition holds? Alternatively, can the results be generalized to relax this condition?
- If I understand correctly, Theorem 8 in some sense corresponds to the "unrealizable" setting, in which the target function is outside the span of the features. In this case, should the question be about how accurately the posterior approximate $f$, or how accurately the posterior approximates the projection of $f$ onto the span of the features? (Essentially, I am wondering if this result is in some sense vaccuous in a not terribly interesting way). Any insight the authors can provide would be helpful.
- I am concerned remark 10 is incorrect/misleading and I would appreciate clarification. If I understand correctly, the suggested conclusion is that a sample from the GP prior (almost surely?) satisfies the decay conditon with $\beta=\alpha/2$. I don't see why this would be the case. In particular, for this to be true, it seems we would need there to exist a constant $C$ such that $\sup_{p} \omega_p \leq C$ which is almost surely not the case. Perhaps this only contributes some log factors (see Boucheron et al, 2012, Section 2.5), which will end up hidden in the $\tilde{O}$ notation anyway, but I think some subtlety may have been lost here. If not, I would appreciate clarification.
- Can lower bounds on any of your results be established? I realize this may be beyond the scope of this paper, but it would be interesting to know if the upper bounds presented are optimal.

## Minor comments and questions:
- "For inputs distributed uniformly on the unit interval, Ritter et al. (1995) showed that $r$-times mean square differentiable processes feature an asymptotic power-law decay of the form $λ_p∝p^{−(2r+2)}$" --Does this not follow from the earlier work due to Widom? Is the difference that Ritter doesn't need to assume stationarity?
- Distributions and densities seem to be conflated slightly in the problem setup. In particular, at times $\mathcal{N}$ is used to refer to a distribution while at other times it is the density of the distribution.
- When citing a textbook, please provide a section or page number.
- For Mercer's theorem to apply you need some assumptions on the kernel/input distribution. A relatively standard set of assumptions is compact support for the input distribution and a continuous kernel, though I believe there are more general formulations.

### References for review
- Bucheron, Lugogsi, Massart. Concentration inequalities. 2012.
- Braun, 2006. Accurate Error Bounds for the Eigenvalues of the Kernel Matrix. https://jmlr.org/papers/volume7/braun06a/braun06a.pdf
- Vakili, Khezeli, Picheny. 2020. On Information Gain and Regret Bounds in Gaussian Process Bandits.
-  Zhu,  Williams, Rohwer, Morciniec. 1997. Gaussian regression  and optimal finite  dimensional  linear  models.

**Summary Of The Paper:**

The paper consider asymptotic properties of Gaussian process models where the eigenvalues from Mercer's theorem exhibit polynomial decay. Unlike the earlier analysis of Sollich, the authors do not assume that the model is correctly specified. The authors provide a short experiment to illustrate the theory.

**Summary Of The Review:**

Overall, the result seems interesting. I have only minor concerns about the correctness, but I think these can be resolved (or it is possible I have misunderstood). I think the exposition could be improved to provide better context for the results, and tie them more concretely to real problems. Overall, I think the paper is above the threshold for acceptance.

---

> ### Author Response · Authors · 2021-11-19
> **Response to Reviewer nLLV part 1**
>
> Thank you for the thoughtful review and suggestions.
>
> * In response to your comment "providing more context for each result as well as how the results differ. I would like to see more examples of how they apply to concrete examples", we have added more details on how our results differ from other related works. Please see our common response (C5) as well as the response to reviewer e3WT. **We now include more concrete examples** for other arc-cosine kernels in the Appendix A where we show how the smoothness of the activation function influence the decay rate of the eigenvalues, and thence influence the decay rate of generalization error.
>
>
> * In response to "I suspect that they can only decay quickly if the function is smooth, but the converse is not at all clear to me", indeed the converse is true as we explain below:
> Consider a function $f$ defined on the sphere $\mathbb{S}^{d-1}$ which has an expansion $f=\sum_{k=0}^\infty \sum_{j=1}^{N(p,k)}a_{k,j}Y_{k,j}(x)$ w.r.t. the spherical harmonics basis $Y_{k,j}$, $k\geq 0,1\leq j\leq N(d,k)$ where $N(d,k)$ is the number of spherical harmonics of degree $k$. If $f$ is $s$-times differentiable with derivatives bounded by $\eta$, i.e. $\|(-\Delta)^{s/2}f\|\leq\eta$ where $\Delta$ is the Laplace-Beltrami operator on the sphere, then we have $\sum_{k=0}^\infty\sum_{j=1}^{N(d,k)} a_{k,j}^2(k(k+d-2))^{s}\leq\eta^2$. So when the $\{a_{k,j}\}$, ${k\geq0,1\leq j\leq N(d,k)}$ decay fast as $k\to\infty$, the above equation holds for large $s$. Then $\|(-\Delta)^{s/2}f\|\leq\eta$ for large $s$, which means that the function is smooth. So the converse is true.
>
> * In response to "neural networks... not convinced should be emphasized so heavily": Indeed, NTK is not the focus of our paper and our results are about Gaussian process regression and kernel ridge regression.
> We have de-emphasized the NTK part and modified the abstract which now reads: "Infinitely wide neural networks can be related to GPR with respect to the neural network GP kernel and the neural tangent kernel, which in several cases is known to have a power-law spectrum."
>
>
> * In response to "Make explicit if $f$ is assumed to be in $L^2(\Omega,\rho)$", we have added a line to reflect that this is the case (second sentence in Section 3).
>
> * "Does $\Phi_0$ in general depend on $f$?" Just as you said, $\Phi_0$ depends on both $f$ and $\{\phi_p\}_{p\geq 1}$. This is reflected in Eq.9.
>
>
> * In response to "The assumption that eigenfunctions are uniformly bounded seems quite restrictive." We have **relaxed our Assumption 6 and now only assume that the infinite norm of the eigenfunction is bounded by a power function.** The same assumption appears, for example, in Valdivia (2018, Hypothesis $H_1$) and is less severe than the assumption of uniformly bounded eigenfunctions as appeared in several other works in the GP literature as pointed out by reviewer nLLV, e.g., e.g., Braun (2006); Chatterji et al. (2019); Vakili et al. (2021).
>
>     E. A. Valdivia (2018), Relative concentration bounds for the spectrum of kernel matrices. arXiv:1812.02108.
>
>     [The other references are listed in the paper].
>
>
> * In response to your question about the unrealizable setting "should the question be about how accurately the posterior approximate $f$ , or how accurately the posterior approximates the projection of $f$ onto the span of the features?".
> Unlike the realizable case ($\mu_0=0$) where the generalization error goes to zero, **we now show that for the unrealizable case (the setting $\mu_0>0$ in Theorems 8, 11 and 12) when the target function is outside the span of the features, the generalization error converges to a constant that depends on $\mu_0$.** These results show that the the error in approximating $f$ is $\sim\mu_0^2$.
>
>
> * In response to your concern about about Remark 10:
> You raised a valid point. We have modified Remark 10 as follows: We can bound the Gaussian random variables $(\omega_p)$ almost surely as $|\omega_p|\leq C\log p$, where $C=\sup_{p\geq 1}{\frac{|\omega_p|}{\log p}}$ is a finite constant. Comparing with the expansion of $f(x)$, we find that $\mu_p=\sqrt{\lambda_p}\omega_p=O(p^{-\alpha/2}\log p)=O(p^{-\alpha/2+\varepsilon})$ where $\varepsilon>0$ is arbitrarily small.
>
>
> * Responding to your question "Can lower bounds on any of your results be established?"
>
>     Yes! **We have improved all our results showing optimality** now. Please see our common response C1.

---

> > ### Author Response · Authors · 2021-11-19
> > **Response to Reviewer nLLV part 2**
> >
> > For the minor comments:
> >
> >  - The works of Widom and Ritter et al are in similar spirit and we now state "The asymptotic rate of decay of the eigenvalues of stationary kernels for input distributions with bounded support is well understood" when referring to these works.
> >
> >  - In regard to your comment "Distributions and densities seem to be conflated slightly in the problem setup," we have now taken care of this wherever applicable.
> >
> >  - We have now added page numbers to book references. Thanks for pointing this out.
> >
> >  - In regard to the comment "For Mercer's theorem to apply you need some assumptions on the kernel/input distribution," we have added the assumption of a continuous kernel (first paragraph in Section 2, P.3). Compact support for the input distribution is not needed because we don't need the uniform convergence of Mercer's expansion. Thanks for pointing this out.
> >
> >
> > We have made significant efforts to address your concerns. In our revision we obtained a series of new results and improvements of our previous results, which in our view substantially strengthened the paper. We hope you will agree with this and consider raising your score to an accept.

---

> > > ### Author Response · Authors · 2021-11-21
> > > **Any further discussion before the end of rebuttal?**
> > >
> > > Dear reviewer,
> > >
> > > We made thorough efforts to address all comments from your initial review. We hope our responses and updated manuscript facilitate your assessment and motivate you to update your recommendation. Kindly do let us know if there is anything else you would like to see improved or clarified before the end of the rebuttal period. We remain attentive to your feedback!

---

> > > > ### Comment · Reviewer_nLLV · 2021-11-22
> > > > **Thank you for the reply**
> > > >
> > > > Hi,
> > > >
> > > > Thank you for the detailed reply to my comments. The proposed changes sound good, and appear to address many of my questions/concerns. As they are relatively substantial and were posted just 1 week-day ago, I am still in the process of reviewing them. I will post questions when I have had time to go through the changes made, and consider revising my score appropriately to reflect these changes.

---

> > > > > ### Comment · Reviewer_nLLV · 2021-11-28
> > > > > **Follow up on revisions**
> > > > >
> > > > > Ok, I think that many of my concerns were well-addressed and the changes improved the paper. I will increase my score to an accept (8) to reflect these improvements. i would encourage the authors to give the appendix a bit more structure (provide summaries of what results are where, give more in depth sketches that build intuition for why lemmas will be useful etc). I have minor comments, but think these can all be addressed easily.
> > > > >
> > > > > Minor comments:
> > > > > - This sentence seems slightly misleading to me: "The most prominent kernels associated with infinite-width NNs are the Neural Network Gaussian Process (NNGP) kernel when only the last layer is trained (Lee et al., 2018; de G. Matthews et al., 2018), and the Neural Tangent Kernel (NTK) when the entire model is trained (Jacot et al., 2018)." I think of NNGP as more about the distributional properties of the stochastic process associated to a certain distribution over weights than a particular training procedure. Please consider rephrasing slightly (or clarify why this phrasing is correct). I do think it is good to mention both of these well-studied limits.
> > > > > - Consider introducing some concept of a loss/risk prior to discussing learning curves. In particular, when you right the learning curve has the form $n^{-beta}$ it isn't clear what quantity this is (e.g classification accuracy, RMSE, log likelihood, any of these?).
> > > > > - In the section on Mercer's theorem, can you justify why $L_k$ maps into $L^2$? I think you mind still need some assumption on the kernel beyond continuity if the domain is not compact (e.g. I can imagine you might run into issues if the domain is all of $\mathbb{R}$ and the kernel grows rapidly away from $0$). I think boundedness would suffice (by Holder's inequality) and since the measure is finite.
> > > > > - It might be worth noting that for stationary kernels on compact subset of $\mathbb{R}^d$, $\alpha$ is implied by certain smoothness assumptions on the kernel (essentially the take-away from Widom 1963).
> > > > > - In equation 14 and 15, it might be more natural to use $K$ instead of $\Phi \Lambda \Phi^T$

---

> > > > > > ### Author Response · Authors · 2021-11-29
> > > > > > **Thank you for your feedback**
> > > > > >
> > > > > > Thank you for your careful evaluation of our revision and your positive feedback. We are glad to read that you found your concerns well-addressed. Of course we will take care of the minor comments suggested in your last response.

---

### Official Review · Reviewer_N1ou · 2021-11-03

**Correctness:** 4
**Technical Novelty And Significance:** 2
**Empirical Novelty And Significance:** Not applicable
**Recommendation:** 6
**Confidence:** 2

**Main Review:**

The paper uses the negative log marginal likelihood and its relationships to Bayesian generalization error to give interesting generalization bounds. The paper largely cites previous theorems for most of its theoretical distributions and under power decay assumption, they use the log likelihood formula to derive bounds. However, it is unclear how novel the contributions are, other than plugging the power decay assumption into the previously derived formula. Furthermore, the relationship to NTK is only briefly touched upon.

Given that the paper is largely theoretical, I believe there should be more emphasis on the technical novelty and the difficulty of extracting an asymptotic expression for the loglikelihood formula given the strong assumptions made on the kernel spectrum. For example, what is the novel theoretical contribution for showing that
? Doesn't it follow from a matrix concentration inequality? If it's an nontrivial application of matrix concentration, can you point out how it is nontrivial. I also saw that you applied a spectral truncation argument. Is this argument necessary and/or novel? Perhaps emphasizing such points in the main sections of the paper would significantly strengthen the paper since it seems like the reader would need to infer the non-triviality from the appendix.

**Update: The authors have significantly strengthened the theoretical understanding of the problem and the intuition/novelty of their techniques.

**Summary Of The Paper:**

The paper summarizes the generalization error of kernel ridge regression and Gaussian processes under a spectral decay assumption.

**Summary Of The Review:**

Since this is largely a theoretical paper, it is unclear if these results are novel since it seems like just plugging the power decay assumption into the log likelihood formula.

---

> ### Author Response · Authors · 2021-11-19
> **Response to Reviewer N1ou**
>
> * In regard to your comment "it is unclear how novel the contributions are, other than plugging the power law decay assumption into the previously derived formula" we offer the following clarification: Kindly note that **it is non-trivial to extract an asymptotic expression for the expected negative log-marginal likelihood** formula from Proposition 3. Further, note that our main results are in Section 3 and they do not follow from simply "plugging" in the power-law decay assumption into the previously derived formulas.
>
>     In the analysis we need to show that $\Phi^T\Phi$ concentrates around $nI$, where $\Phi$ is the empirical eigenfunction matrix. This is a key and non-trivial step in our proof. If one were to assume that $\Phi$ is orthogonal, then the computations could be carried out in terms of a simple regular series. However, in practice $\Phi$ is not orthogonal.
>
> * In regard to your comment "the relationship to the NTK is only briefly touched upon", we clarify that we only leverage the well-known correspondence between infinitely wide neural networks and kernel ridge regression (KRR) w.r.t. the NTK, and our methods can be applied to study the generalization error of infinitely wide neural networks. NTK is not the focus of our paper.

---

> > ### Author Response · Authors · 2021-11-21
> > **Any further discussion before the end of rebuttal?**
> >
> > Dear reviewer,
> >
> > We made thorough efforts to address all comments from your initial review. We hope our responses and updated manuscript facilitate your assessment and motivate you to update your recommendation. Kindly do let us know if there is anything else you would like to see improved or clarified before the end of the rebuttal period. We remain attentive to your feedback!

---

> > > ### Comment · Reviewer_N1ou · 2021-11-22
> > > **Response**
> > >
> > > Thanks for the response. I'm afraid I was unable to fully check the proof details, which is reflected in our confidence score. Furthermore, I think reviewers were not expected to read beyond the main paper section and I think there one can strengthen the paper without requiring the reader to thoroughly understand the appendix.
> > >
> > > Given that the paper is largely theoretical, I believe there should be more emphasis on the technical novelty and the difficulty of extracting an asymptotic expression for the loglikelihood formula given the strong assumptions made on the kernel spectrum. For example, what is the novel theoretical contribution for showing that $\Phi^T\Phi \approx nI$? Doesn't it follow from a matrix concentration inequality? If it's an nontrivial application of matrix concentration, can you point out how it is nontrivial. I also saw that you applied a spectral truncation argument. Is this argument necessary and/or novel? Perhaps emphasizing such points in the main sections of the paper would significantly strengthen the paper since it seems like the reader would need to infer the non-triviality from the appendix.

---

> > > > ### Author Response · Authors · 2021-11-22
> > > > **Technical Novelty of Our Proof Part 2**
> > > >
> > > > * **Assumptions**
> > > > In regard to your comment "given the strong assumptions made on the kernel spectrum":
> > > >
> > > >     Please note that the power-law assumptions we use for the kernel eigenspectrum (Assumption 4) and the eigenexpansion coefficients of the target function (Assumption 5) are standard in the kernel learning literature where they are called resp. the "capacity" and "source" condition. These conditions are related to the effective dimension of the problem and the difficulty of learning the target function. Please refer to the paragraph "Asymptotics of the generalization error of kernel ridge regression (KRR)" in P.2 of our Introduction for pertinent references. Also, see comments by Reviewer e3WT who says: "Studying the asymptotic decay of the generalisation error for power-law spectrum (Assumption 4) and target function decay (Assumption 5) is a classic subject in the Kernel ridge regression (KRR) literature, where these are known as capacity and source conditions."
> > > >
> > > >     More from a practical angle, several recent works empirically justify and rely on a power law assumption for the NTK spectrum. Please refer to the paragraph "Power-law decay of the GP kernel eigenspectrum" in P.2 of our Introduction for pertinent references. Velikanov and Yarotsky (2021) theoretically showed that the power-law asymptotics of the infinite network NTK has the form $\lambda_p \propto p^{-\alpha}$ with $\alpha=1+\tfrac{1}{d}$, where $d$ is the input dimension.
> > > >
> > > >     Finally, our Assumption 6 only assumes that the infinite norm of the eigenfunction is bounded by a power function. The same assumption appears, for example, in Valdivia (2018, Hypothesis $H_1$) and is less severe than the assumption of uniformly bounded eigenfunctions as appeared in several other works in the GP literature as pointed out by reviewer nLLV.
> > > >
> > > >     In summary, our assumptions on the kernel spectrum are frequently encountered in kernel learning and relates to recent advances on neural networks. Furthermore, the generality of our analysis allows us to recover as special cases some results on learning curves for GPR and KRR that were previously obtained under more restricted settings (please see Contributions, P.2, and Remarks 10 and 14).
> > > >
> > > > We have also added new experiments in Appendix A for arc-cosine kernels that lend support to our theory.
> > > >
> > > > Thanks again for your valuable feedback. We hope our response and updated manuscript addresses your concerns and you consider raising your score.

---

> > > > ### Author Response · Authors · 2021-11-22
> > > > **Technical Novelty of Our Proof Part 1**
> > > >
> > > > Thanks for your valuable suggestions. Based on your suggestion, we have updated the manuscript to emphasize the technical novelty of our proof (please see Proof sketch of Theorem 7, P.6 in main text). The updated proof sketch gives a concise and intuitive exposition of the proof. Below is a summary vis-a-vis your questions/comments.
> > > >
> > > > * **Technical novelty**
> > > > For the proof of Theorem 7, first note that $\mathbb{E}_{{\epsilon}}F^0(D_n)=T_1(D_n)+T_2(D_n)$. We highlight three key steps in the proof, viz., approximation, decomposition and concentration, also answering your questions about innovation along the way.
> > > >
> > > >     - **Approximation**: The approximation step builds upon a truncation argument where we analyze the terms $T_1(D_n)$ resp. $T_2(D_n)$ by considering their truncated versions, $T_{1,R}$ resp. $T_{2,R}$ and bound the corresponding residual errors. Concretely, we show that the asymptotics of $T_{1,R}$ resp. $T_{2,R}$ dominates that of the residuals, $|T_{1,R}(D_n)-T_1(D_n)|$ resp. $|T_{2,R}(D_n)-T_2(D_n)|$. Intuitively, the choice of the truncation parameter $R$ is governed by the fact that $\lambda_R =\Theta( R^{-\alpha})=n^{-1+\kappa\alpha}=o(n^{-1})$.
> > > >
> > > >         Hence, in regard to your question: "you applied a spectral truncation argument. Is this argument necessary and/or novel?," the answer is yes, the truncation argument is necessary and novel.
> > > >
> > > >     - **Decomposition**: In this step, we decompose $T_{1,R}$ into (i) a term independent of $\Phi_R$ (the truncated matrix of eigenfunctions evaluated at the data points) and (ii) a series involving $\Phi_R^T\Phi_R-nI_R$, and likewise for $T_{2,R}$. This prepares us for the final step where we employ concentration inequalities.
> > > >
> > > >     - **Concentration**: We draw on tools from matrix concentration to show that the terms independent of $\Phi_R$ (item (i) above) dominate the series involving $\Phi_R^T\Phi_R-nI_R$ (item (ii) above) giving us the desired result.
> > > >
> > > >         In regard to your question: "If it's an nontrivial application of matrix concentration, can you point out how it is nontrivial":
> > > >
> > > >         Note that an ordinary application of the matrix Bernstein inequality to $\Phi_R^T\Phi_R-nI_R$ yields $\|\Phi_R^T\Phi_R-nI\|_2=O(R\sqrt{n})$, which is not sufficient for our purposes, since this would give $O(R\sqrt{n})=o(n)$ only when $\alpha>2$. The key idea in our proof is to instead consider the matrix $\Lambda_R^{1/2}(I+\frac{n}{\sigma^2}\Lambda_R)^{-1/2}\Phi_R^T\Phi_R(I+\frac{n}{\sigma^2}\Lambda_R)^{-1/2}\Lambda_R^{1/2}$ which we can show concentrates around $n\Lambda_R(I+\frac{n}{\sigma^2})^{-1}$. This is a nontrivial consideration which allows us to obtain results that are valid for $\alpha>1$ and thereby cover cases of practical interest, e.g., the NTK of infinitely wide shallow ReLU network (Yarotsky and Velikanov, 2021) and the arc-cosine kernels over high-dimensional hyperspheres (Ronen et al, 2019) that have $\alpha=1+O(\tfrac{1}{d})$, where $d$ is the input dimension.
> > > >
> > > >         [The references above are listed in the paper]

---

> > > > ### Author Response · Authors · 2021-11-29
> > > > **Any questions about our latest response?**
> > > >
> > > > Dear reviewer,
> > > >
> > > > Based on your suggestions, we have rewritten the proof sketch highlighting the three key steps in our proof (viz., approximation, decomposition and concentration) emphasizing how our application of matrix concentration and spectral truncation is novel. Our assumptions are fairly standard in the kernel learning literature and we point to key references supporting this fact. Kindly let us know if you have any questions about the revised proof sketch and our latest response. We remain attentive to your comments and feedback!

---

### Author Response · Authors · 2021-11-19
**Common response: Summary of main updates to the manuscript**

We thank the reviewers for their careful consideration of our manuscript and thoughtful suggestions. We have made significant improvements to our initial results in response to their comments. The main updates are the following, with proofs in the appendix updated accordingly to cover these innovations.

* (C1) We have **improved all our results showing that our previous upper bounds are optimal**. Hence in the revision our results are now expressed in $\Theta$-notation rather than $O$-notation. This responds to the questions about optimality raised by reviewers nLLV and LR4t.

    In the **unrealizable case** (the setting $\mu_0>0$ in Theorems 8, 11 and 12) when the target function is outside the span of the features, we show that the **generalization error converges to a specific constant that depends on $\mu_0$**. This addresses a question by reviewer nLLV.

* (C2) For kernel ridge regression (KRR), we now **distinguish between the variance of the noise ($\sigma_{\mathrm{true}}$) and the variance of the model ($\sigma_{\mathrm{model}}$)**. This allows us to show how the model variance (equivalently, the regularization strength in KRR) influences the mean squared generalization error. In particular, Theorem 12 shows that when $\sigma_{\mathrm{model}}$ is smaller (and within a certain range), the generalization error is lower and this agrees with an earlier result by Cui et al (2021); see Remark 14. In a similar spirit, we also incorporate the noise and model variances for the Bayesian generalization error in Theorem 9. This addresses a comment from reviewer LR4t.

* (C3) In all our theorems, which make high-probability statements, we have added the range of the failure probability $\delta$ depending on $n$, $\alpha$, and $\beta$. This allows us to **quantify precisely the improvements implied by our high-probability bounds compared to, for example, known upper bounds in expectation** (w.r.t. the training inputs) for the generalization error. See Remark 14 for the implications. This addresses a comment by reviewer e3WT.

* (C4) We have **relaxed Assumption 6**. The new assumption is less restrictive than assuming uniformly bounded eigenfunctions, which has been used in several previous works in the GP literature, e.g., Braun (2006); Chatterji et al. (2019); Vakili et al. (2021) as pointed out by reviewer nLLV. This addresses a comment from reviewer nLLV.

* (C5) We have added a **new paragraph titled "Asymptotics of the generalization error of kernel ridge regression (KRR)"** in the Introduction (P.2) that provides a better context for our results in relation to the existing literature on KRR. We clearly emphasize the differences between our contribution and previous works (see Contributions in the Introduction P.2, and Remarks 10 and 14). This addresses comments by reviewer e3WT.

    We have further **improved the exposition by providing more intuition** on how to interpret our bounds (see text below Eqs. 13-16, in Section 3 P.5, and the paragraph following Eq.20 in P.7). This responds to a suggestion from reviewer nLLV.

* (C6) We are including **new experiments in Appendix A for arc-cosine kernels of different orders** and show how the smoothness of the activation function influences the decay rate of the eigenvalues, and thence influences the decay rate of the generalization error. This addresses a comment by reviewer LR4t.

* Finally, we feel obligated to point out that, **contrary to N1ou's review summary, the results we derive cannot be obtained by simply "plugging in" the power-law decay assumption** into the formula for the negative log-marginal likelihood and the generalization error. Had this been the case, then the same logic would apply to prior works in the literature both in the context of Gaussian process regression (e.g. Sollich and Halees, 2002) and kernel ridge regression (e.g., Bordelon et al., 2020; Cui et al., 2021), which is clearly not the case. We provide more details in the individual response.

---

### Comment · Area_Chair_EWiR · 2021-11-28
**Question regarding comparison to KRR, in part. Remark 14**

Dear authors and reviewers,

I tried to parse Remark 14 and to compare it with existing bounds on KRR besides Cui et al.
Here are the results:
- the best choice of $\lambda$ equals those found in other papers, e.g. cited I. Steinwart, D. R. Hush, C. Scovel
- the resulting learning rates also coincide with that paper.

These finding are based upon the "translation" $\alpha = 1/p$
and $2 \beta = 1 + \alpha  \tilde \beta$, where $\tilde \beta$ is the $\beta$ considered in the paper by Steinwart et al. and "translation" is only correct up to some arbitrarily small $\epsilon$ as the source conditions slightly differ. Also, I think, the same rates can be found in the cited Fisher and Steinwart, this time, however, without "cutting" the predictor to some output interval.

Could you comment on this?

---

> ### Author Response · Authors · 2021-11-29
> **Response to Area Chair EWiR**
>
> Thank you for comparing our bounds with those in Refs [1] and [2]:
>
> [1] Steinwart, Hush, and Scovel (2009), Optimal Rates for Regularized Least Squares Regression.
>
> [2] Fischer and Steinwart (2020), Sobolev Norm Learning Rates for Regularized Least-Squares Algorithms.
>
> In the following we offer a comparison of the results in Refs [1,2] and ours. To the best of our understanding of those results, the main differences are both in terms of generality and optimality, which we expand on below.
> We also show how the learning rate in our Theorem 12 matches those in [1,2] for the best choice of the regularization parameter $\lambda$.
>
> Below, we will use $\tilde{\alpha}$ and $\tilde{\beta}$ to represent $\alpha$ and $\beta$ in [2], and $\tilde{\beta}$ to represent $\beta$ in [1], in order to distinguish them from $\alpha$ and $\beta$ used in our paper.
>
> * **Assumptions:** The upper bounds in [1, Corollaries 3 and 6] are based on Assumption (7) with $s=p$, which is true if eigenfunctions are uniformly bounded, and the capacity condition [1, Eq.6] holds for some $p\in(0,1)$ (see [1, Theorem 2]).
>
>     Our Assumption 6 assumes that the infinite norm of the eigenfunction is bounded by a power function and the case when $\tau>0$ is not covered by [1]. Below, we give a concrete example where [1, Assumption 7] is not valid but our Assumption 6 holds:
>
>     Suppose that the capacity condition $\lambda_m=m^{-\alpha}$, $\alpha>2$ holds and the eigenfunctions are normalized Legendre polynomials, i.e. $\phi_m(x)=\sqrt{2m-1}P_{m-1}(x)$ where $P_{m-1}(x)$ is the Legendre polynomial of degree $m-1$ and $x$ is uniformly distributed on $[-1,1]$. These eigenfunctions satisfy our Assumption 6 with $\tau=1/2$ but not [1, Assumption (7)] with $s=p$. To see this, choose $f(x)=\sum_{i=R+1}^\infty\lambda_i\sqrt{2i-1}\phi_i(x)$ for some $m>0$. Since $\phi_i(1)=\sqrt{2i-1}P_{i-1}(1)=\sqrt{2i-1}$, we have $\| f \|_ {\infty} \geq f(1) = \sum_{i=R+1}^\infty (2i-1)\lambda_i=\Omega(R^{2-\alpha})$. It is easy to verify that $\|f\|_ H=(\sum_{i=R+1}^\infty(2i-1)\lambda_i)^{1/2}=O(R^{\frac{2-\alpha}{2}})$ and $\|f\|_ {L_2}=(\sum_{i=R+1}^\infty(2i-1)\lambda_i^2)^{1/2}=O(R^{\frac{2-2\alpha}{2}})$. Then  $\|f\|_ H^{1/\alpha}\|f\|_ {L_2}^{1-1/\alpha}=O(R^{\frac{-2\alpha+3}{2}})$. Since $2-\alpha>\frac{-2\alpha+3}{2}$, $\|f\|_ \infty\geq \|f\|_ H^{1/\alpha}\|f\|_ {L_2}^{1-1/\alpha}$ as $R\to\infty$. So [1, Assumption (7)] with $s=p$ doesn't hold in this case. Legendre polynomials are common when we consider orthonormal basis on the unit interval.
>
>     **Under the assumption of uniformly bounded eigenfunctions (the case $\tau=0$ in our Assumption 6), our Theorem 12 matches the learning rate in [1, Corollary 6]**:
>
>     To see this, note that our Theorem 12 shows $\mathbb{E}_ {\epsilon} M(D_n)=\Theta\bigg(\max\{\sigma_{\mathrm{true}}^2n^{\tfrac{1-\alpha-t}{\alpha}},n^{\tfrac{(1-2\beta)(1-t)}{\alpha}}\}\bigg)$. The best rate is achieved when $\frac{1-\alpha-t}{\alpha}=\frac{(1-2\beta)(1-t)}{\alpha}$, i.e., when $t=1-\frac{\alpha}{2\beta}$.
>
>     Applying our Theorem 12 to the special case of uniformly bounded eigenfunctions (the setting $\tau=0$ in our paper) for $t=1-\frac{\alpha}{2\beta}$, we match the learning rate in [1, Corollary 6] by noting that $\lambda=n^{t-1}=n^{-\frac{\alpha}{2\beta}}$ and using the "translation": $\alpha=1/p$ and $2\beta=1+\alpha\tilde{\beta}$.
> Similarly, applying our Theorem 12 with $t=1-\frac{\alpha}{2\beta}$ recovers the learning rate in [2, Theorem 1(ii)].
>
> * **Optimality:**
>     The results in [1,2] are optimal in a minmax sense. Concretely, [1, Corollary 3] gives an upper bound for the generalization error that is valid for $\lambda\in (0,1]$, and the upper bound for the best possible $\lambda$ matches their minmax lower bound [1, Theorem 9].
>
>     Our asymptotic results (both upper and lower bounds) depend on $\lambda$ and are precise in terms of optimality in the sense that there is no difference (up to constant factors) between our upper and lower bounds on the generalization error $\mathbb{E}_{\epsilon} M(D_n)$ (see Theorem 12).
>
> * **Unrealizable case:** Our methods can deal with the case when the target function is outside the span of the eigenfunctions with positive eigenvalues, i.e., $\mu_0>0$ and show that the generalization error converges to the exact constant $\mu_0^2$. This case is not covered in Refs [1,2].
>
> Thanks again for your comments. Please let us know if have missed anything.

---

### Decision · Program_Chairs · 2022-01-20

**Decision:**

Accept (Poster)

**Comment:**

The title of the paper nicely summarizes the main goal of the paper and the abstract does the same for the achieved results. For this reason I abstain from providing another summary.

The initial reviews were somewhat mixed but during the discussion phase, a lot of questions have been resolved so that actually three reviewers updated (upgraded) their score. Remark 14 certainly needs to be updated according to the discussion in the final few days of the rebuttal phase. In addition, one reviewer pointed to a naive application of Mercer's theorem. This should be addressed as well, either by restricting to compact domains and continuous kernels as suggested by the reviewer, or by considering generalizations as done by e.g. the cited Fischer and Steinwart. Finally, the cited survey by Kanagawa et al also contains some information on learning curves and thus it should be cited more prominently, e.g. around Remark 14.

In any case, this paper is above the acceptance threshold.